# Cellular anatomy of the mouse primary motor cortex

Rodrigo Muñoz-Castañeda[1,21], Brian Zingg[2,3,21], Katherine S. Matho[1,21], Xiaoyin Chen[1,4,21], Quanxin Wang[4,21], Nicholas N. Foster[2,3], Anan Li[5,6], Arun Narasimhan[1], Karla E. Hirokawa[4,7], Bingxing Huo[1], Samik Bannerjee[1], Laura Korobkova[3], Chris Sin Park[8], Young-Gyun Park[9], Michael S. Bienkowski[3,10], Uree Chon[11], Diek W. Wheeler[12], Xiangning Li[5,6], Yun Wang[4], Maitham Naeemi[4], Peng Xie[13], Lijuan Liu[13], Kathleen Kelly[1], Xu An[1,14], Sarojini M. Attili[12], Ian Bowman[2,3], Anastasiia Bludova[1], Ali Cetin[4], Liya Ding[13], Rhonda Drewes[1], Florence D'Orazi[4], Corey Elowsky[1], Stephan Fischer[1], William Galbavy[1], Lei Gao[2,3], Jesse Gillis[1], Peter A. Groblewski[4], Lin Gou[2,3], Joel D. Hahn[15], Joshua T. Hatfield[1,14], Houri Hintiryan[2,3], Junxiang Jason Huang[16], Hideki Kondo[1], Xiuli Kuang[17], Philip Lesnar[4], Xu Li[1], Yaoyao Li[17], Mengkuan Lin[1], Darrick Lo[2,3], Judith Mizrachi[1], Stephanie Mok[4], Philip R. Nicovich[4,7], Ramesh Palaniswamy[1], Jason Palmer[1], Xiaoli Qi[1], Elise Shen[4], Yu-Chi Sun[1], Huizhong W. Tao[16], Wayne Wakemen[4], Yimin Wang[13,18], Shenqin Yao[4], Jing Yuan[5,6], Huiqing Zhan[1], Muye Zhu[2,3], Lydia Ng[4], Li I. Zhang[16], Byung Kook Lim[6,19], Michael Hawrylycz[4], Hui Gong[5,6], James C. Gee[20], Yongsoo Kim[11], Kwanghun Chung[9], X. William Yang[8], Hanchuan Peng[13], Qingming Luo[5,6], Partha P. Mitra[1], Anthony M. Zador[1], Hongkui Zeng[4], Giorgio A. Ascoli[12✉], Z. Josh Huang[1,14✉], Pavel Osten[1✉], Julie A. Harris[4,7✉] & Hong-Wei Dong[2,3✉]

An essential step toward understanding brain function is to establish a structural framework with cellular resolution on which multi-scale datasets spanning molecules, cells, circuits and systems can be integrated and interpreted[1]. Here, as part of the collaborative Brain Initiative Cell Census Network (BICCN), we derive a comprehensive cell type-based anatomical description of one exemplar brain structure, the mouse primary motor cortex, upper limb area (MOp-ul). Using genetic and viral labelling, barcoded anatomy resolved by sequencing, single-neuron reconstruction, whole-brain imaging and cloud-based neuroinformatics tools, we delineated the MOp-ul in 3D and refined its sublaminar organization. We defined around two dozen projection neuron types in the MOp-ul and derived an input–output wiring diagram, which will facilitate future analyses of motor control circuitry across molecular, cellular and system levels. This work provides a roadmap towards a comprehensive cellular-resolution description of mammalian brain architecture.

The brain is an information processing network comprising a set of nodes interconnected with sophisticated wiring patterns. Superimposed on this anatomical infrastructure are genetically encoded molecular machines that mediate cellular processes, shaping the neural circuit dynamics underlying cognition and behaviour. Historically, brain organization has been explored using different techniques at descending levels of granularity: grey matter regions (macroscale), cell types (mesoscale), individual cells (microscale) and synapses (nanoscale)[1]. MRI and classic anatomical tracing have produced macroscale connectomes in human[2] and other mammalian brains[3–5], providing a panoramic—but still coarse—view of organizational principles for further exploration[6]. An essential step toward a comprehensive understanding of brain function is to establish a structural framework with cellular resolution on which multi-scale and multi-modal information spanning molecules, cells, circuits and systems can be registered, integrated, interpreted and mined.

Several recent technical advances together enable large-scale mapping of mammalian brain circuits with cellular resolution.

High-throughput single-cell RNA-sequencing efforts are creating transcriptomic cell-type censuses for multiple brain regions[7]. These data contribute to the development of genetic toolkits enabling reliable experimental access to an increasingly large set of molecularly defined cell types[8]. Continued innovations in volumetric light microscopy enable automated high-resolution imaging of cells and single axons across entire rodent brains. With computational advances in image processing, machine learning and management of large (terabyte) volume image datasets[9], and with the construction of 3D common coordinate framework (CCF) brain atlases that serve as a unified anatomical reference brain for cross-modal data integration[10], new datasets will contribute to revealing general organizational principles of brain architecture at all scales.

Recognizing this emerging opportunity, the BICCN established a multi-laboratory collaboration with the goal of systematically classifying neuron types and mapping multi-scale connectivity in the mouse brain. As a first step, we focused our combined efforts on the MOp-ul. We applied expertise in cell-type-targeted genetic and viral

labelling, high resolution whole-brain imaging, barcoded anatomy resolved by sequencing (BARseq)-based projection mapping[11], complete single-neuron morphological reconstruction, and state-of-the-art neuroinformatic methods for CCF registration. We derived a comprehensive, projection neuron (PN) type-based wiring diagram of the mouse MOp-ul that will facilitate future analyses of motor control infrastructure across molecular, cellular and systems levels. This exemplar brain structure provides a roadmap towards a cellular description of mammalian whole-brain architecture and the multi-scale connectome.

## Results

We established an integrated cross-laboratory anatomical analysis platform comprising myriad technologies, tools, methods, data analyses, visualizations and web-based portals for open access to data and tools[3,4,8,10,12–27] (Extended Data Fig. 1, Methods). Structure abbreviations are defined in Supplementary Table 1 and specific mouse lines in Supplementary Table 2.

### MOp-ul borders and cell types

The spatial location of rodent primary motor cortex (MOp) has been defined by cytoarchitecture, micro- or optogenetic- stimulation[28] and anatomical tracing[29,30], yet discrepancies remain, including between standard 2D and 3D mouse brain reference atlases[10,31–33]. Here, we first defined the MOp-ul borders in 3D using a collaborative workflow with multimodal data co-registered and cloud-visualized[26,27] at full resolution for joint review, delineation and reconciliation (Fig. 1a, Supplementary Video 1; datasets can be viewed at https://viz.neurodata.io/?json_url=https://json.neurodata.io/v1?NGStateID=LwZ24nSZk1JTHw).

MOp-ul shares its lateral border with the primary somatosensory area (SSp); seen in Nissl- and NeuroTrace-stained sections as a transition from larger layer 5 (L5) somas in MOp to smaller somas in the SSp cell-sparse L5a and cell-dense L5b sublayers (Fig. 1b, Extended Data Figs. 2a, b; see also the Allen Reference Atlas[33] (ARA) and http://brain-maps.org). MOp is classically described as agranular cortex, but we identified a 'granular' L4, with densely packed small somas throughout primary (MOp) and secondary (MOs) motor cortex, albeit narrower than in SSp (Fig. 1b, Extended Data Fig. 2b; see also algorithmic analysis of MOp–SSp border, revealing individual variations between animals in Extended Data Fig. 2c, d, Supplementary Information).

Next, we used neuron-type distribution and long-range projection patterns in determining areal delineations[3,10,20,31]. The density of VGluT1 (also known as Slc17a7)-positive neurons corroborated the transition of L4 and L5 at the MOp–SSp border (Fig. 1a, b, Supplementary Video 2), and VGluT3+ neurons highlighted the MOp-ul–MOs medial border (Fig. 1a, b). Lateral and medial borders were further delineated by adeno-associated virus (AAV)-based axonal labelling from SSp upper limb area (SSp-ul) to MOp-ul, and from ventrolateral orbital area (ORBvl) or dorsal retrosplenial area (RSPd) to MOs[3] (Extended Data Fig. 3a). Rostro-caudal borders were defined using AAVretro tracing from the cervical (to delineate upper limb) or lumbar (to delineate lower limb) spinal cord (Fig. 1b, Extended Data Figs. 3b, c, 4, Supplementary Video 1). This revealed two adjacent clusters of cervical spinal cord-projecting neurons: a medial cluster in MOp L5 (projecting to the intermediate and ventral horn) and a lateral cluster underneath SSp L4 (projecting to the dorsal horn) (Fig. 1b, Extended Data Figs. 4, 5i). Finally, the MOp-ul borders were further validated using triple anterograde labelling. Injecting AAV-RFP, *Phaseolus vulgaris* leucoagglutinin (PHAL) and AAV-GFP into MOs, MOp-ul and SSp, respectively, revealed topographically organized, discrete terminal fields in different brain structures (Fig. 1b, Extended Data Fig. 5).

MOp-ul borders were drawn on the CCFv3 average template[10] using Neuroglancer to render a 3D volume aligned with other 3D histological data (Fig. 1c, Extended Data Fig. 2e, Supplementary Video 2). To facilitate integration with existing atlases, we also imported ARA[33] and

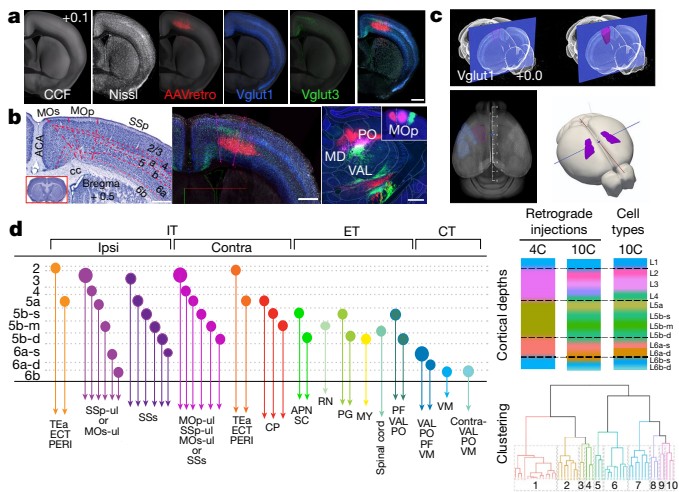

**Fig. 1 | Delineation of the MOp-ul region and its cell-type organization. a**, Brains with different anatomical labelling modalities (Nissl-stained: *n* = 3; AAVretro-labelled cervical spinal projecting neurons: *n* = 2; Cre reporter expression, *n* = 1 for Vglut1 and Vglut3) were co-registered in the CCF average template and viewed in Neuroglancer to facilitate delineation of MOp-ul borders. **b**, MOp-ul delineation based on combinatorial Nissl-stained cytoarchitecture (left) (Extended Data Fig. 2) and regional and laminar distributions of AAVretro labelling and Cre expression (middle). A triple-injection strategy was used to further validate distinctive projections of MOp-ul versus adjacent SSp-ul and MOs (right, *n* = 3 for each injection). AAV-RFP (red), PHAL (pink) and AAV-GFP (green) were injected into the MOs, MOp-ul and SSp, respectively (inset, right), revealing mostly non-overlapping terminal fields in the thalamic nuclei, mediodorsal nucleus (MD), CL, PCN and PO (Extended Data Fig. 5). Scale bars, 500 μm. **c**, The MOp-ul was rendered in 3D within the CCF. **d**, Left, schematic showing classification of cortical projection neuron types based on their laminar positions, projection neuron class (IT, PT and CT), and specific projection targets. Right (top), analysis of the MOp-ul layer organization by hierarchical clustering of soma depth for retrogradely labelled cells and Cre driver data (Extended Data Fig. 3). Bottom, clustering dendrogram based on MOp-ul soma depth grouped every 25 μm. ACA, anterior cingulate area; MY, medulla; RN, red nucleus.

Franklin–Paxinos[32] delineations onto the Allen CCFv3 (Extended Data Fig. 2f, Supplementary Information).

Using the new MOp-ul volume delineation as a region of interest, we precisely mapped cell type distributions for several genetically identified cell populations, for example, glutamatergic (VGluT1+), GABAergic (γ-aminobutyric acid-producing) (GAD2+) neurons, major GABAergic subpopulations, and other Cre driver-based populations[12,20] (Extended Data Fig. 3d).

### Laminar organization of neuron types

The traditional parcellation of cortex into 6 or 8 layers is based largely on cytoarchitecture[34], developmental evidence[35] and long-range projection patterns[36]. Cortical PNs comprise three broad classes: (1) intratelencephalic (IT), primarily targeting cortex and striatum with somas in L2–L6; (2) pyramidal tract (PT) (also known as extratelencephalic (ET)), projecting to lower brainstem and spinal cord with somas in L5; and (3) corticothalamic (CT), projecting to the thalamus with somas in L6[37]. To examine the finer-scale relationship between PNs and soma distribution across layers in MOp-ul, we injected classic retrograde (fluorogold and cholera toxin B subunit (CTB)) and rabies viral tracers into 15 known MOp targets in cortex, contralateral caudoputamen (CP), thalamus, midbrain, pons, medulla and spinal cord (Fig. 1b, Extended Data Figs. 3b, c, 7). Labelled MOp-ul PNs were classified according to soma position and projection target (Fig. 1d, Extended Data Fig. 3b, c, 7), and included 16 types of IT, 7 types of ET and 3 types of CT neurons. These experiments also revealed a more

refined laminar organization than previously appreciated, with the 26 PN subtypes spanning 11 newly delineated layers and sublayers (1, 2, 3, 4, 5a, 5b-superficial, 5b-middle, 5b-deep, 6a-superficial, 6a-deep and 6b) (Fig. 1d). This connectivity-based manual delineation was confirmed computationally with hierarchical clustering on the spatial locations of the retrogradely labelled PN somas (Fig. 1d) and corroborated with Nissl-stained cytoarchitecture and gene expression-based cell type distributions (Extended Data Fig. 6).

Of note, we found several novel IT types: (1) temporal association area (TEa)-projecting neurons in L2 and L5, which generate symmetrical or asymmetrical projections to the two hemispheres; (2) MOs- and SSp-projecting neurons in L4; and (3) ipsilateral projecting neurons in L6b (Extended Data Fig. 7). As these PN types were defined on the basis of single-target retrograde tracing, we validated collateral projections in a subset of types using Cre-dependent, target-defined AAV anterograde tracing (Extended Data Fig. 8a). This method revealed several notable findings (Extended Data Fig. 8b, c): both L5a and L5b IT neurons generate bilateral cortical projections. However, L5a IT neurons preferentially innervate ipsilateral CP, whereas L5b IT neurons generate dense bilateral CP projections. Furthermore, axonal terminals of L5b IT neurons are densely clustered into one specific CP domain[13], whereas those arising from the L5a IT neurons spread diffusely into other CP domains.

Visual inspection of gene or transgene expression by in situ hybridization[12,38,39] also revealed many notable, distinct laminar distribution patterns in MOp (Extended Data Fig. 9).

## Outputs of MOp-ul

Axonal projections from rodent motor cortex have been studied extensively[37,40–43]. However, it is challenging to directly compare these independently generated data, as they exist in different spatial frameworks. We integrated our datasets in CCF to map the output of MOp-ul at regional and cell-type levels. First, we labelled the overall MOp-ul output patterns with PHAL[3,13]. MOp-ul projects to more than 110 targets in brain and spinal cord, with approximately 60 receiving moderate to dense innervation (Extended Data Figs. 5, 10, Supplementary Information). Second, we mapped projections from L2/3, L4, L5 IT, L5 ET and L6 CT PN types with Cre-dependent viral tracers in lines selective for these cell types[4,17] (Fig. 2a, b). Synaptic innervation of targets (versus passing fibres) was also confirmed in a subset of experiments using two alternative viral tracing methods (Extended Data Fig. 11).

We quantified labelled axons in 314 ipsilateral and contralateral grey matter regions in CCFv3[10], creating a weighted connectivity matrix to visualize brain-wide projection patterns (Fig. 2c, Source Data Fig. 2). Outputs from MOp-ul predominantly target isocortex, striatum and thalamus (44.9, 29.0 and 8.1% of total axon density, respectively) with less axon in midbrain, medulla and pons (Extended Data Fig. 13d). Cre-defined projection mapping revealed distinct components of the regional output pathway (Fig. 2c, Extended Data Figs. 12, 13a, 14). Projections in Sepw1-L2/3, Cux2-L2/3, Nr5a1-L4, Scnn1a-L4/5, Plxnd1-L2/3 + L5, and Tlx3-L5 were restricted to isocortex and CP, the defining IT feature. Projections in Sim1-L5 and Fezf2-L5/6 were predominantly subcortical, consistent with the ET classification. Projections in Ntsr1-L6 and Tle4-L6 targeted thalamic nuclei, reflective of CT. Several Cre lines labelled multiple PN classes, for example, IT and ET in Rbp4-L5 (Fig. 2a, c, Extended Data Fig. 12).

We performed unsupervised hierarchical clustering on the basis of connectivity weights in all brain regions and identified four main clusters (Fig. 2c). Cluster 1 comprised all experiments with L5 ET cells, including PHAL, AAV-GFP and Rbp4-L5 IT/ET. Cluster 2 contained L6 CT projections, that is, Ntsr1-L6 and Tle4-L6. Clusters 3 and 4 contained IT PN types: Cux2-L2/3, Tlx3-L5 and Plxnd1-L2/3 + L5 in cluster 3, and Sepw1-L2/3, Nr5a1-L4 and Scnn1a-L4 in cluster 4. Clustering confirmed the visual classification of anterograde tracing into expected major PN types, but notable differences do exist in the relative fraction of total projections per structure between lines in the same cluster (for

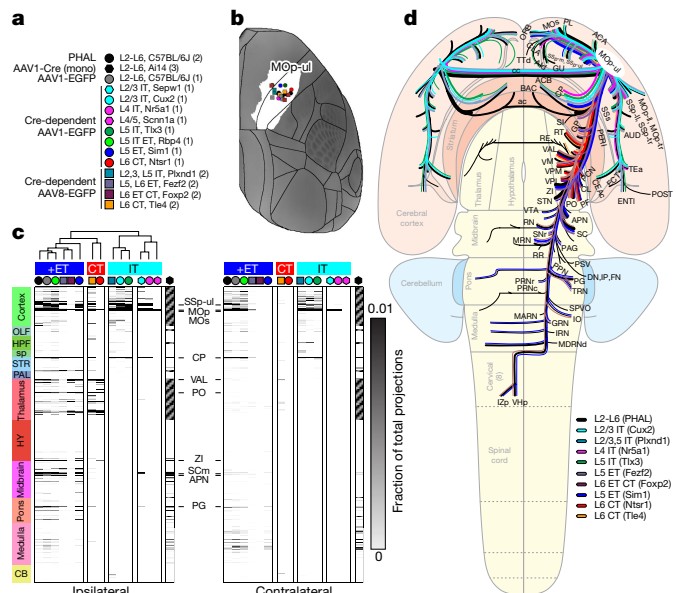

**Fig. 2 | Brain-wide MOp-ul projection patterns by layer and class. a**, Key shows tracer types, mouse lines and layer and projection class selectivity for Cre driver lines used to label axons from MOp neurons. Numbers in brackets represent the number of tracer injection experiments per type. Symbols and colour code are used in **b**–**d**. **b**, Injection sites are plotted on a top-down view of the right cortical hemisphere from CCFv3 with the MOp-ul delineation from Fig. 1 in white. Distance between injection sites is 443.0 ± 185.04 μm (mean ± s.d.). **c**, A directed, weighted connectivity matrix (15 × 628) from MOp to 314 ipsilateral and 314 contralateral targets for each of the fifteen mouse lines or tracers listed in **a**. Each row shows the fraction of the total axon measured from a single experiment or the average when *n* > 1. Rows are ordered by major brain division. For AAV1-Cre monosynaptic tracing, known reciprocally connected regions are coloured grey. We performed hierarchical clustering with Spearman rank correlations and complete linkages, splitting the resulting dendrogram into four clusters. AAV1-Cre was not included in the clustering owing to the many excluded regions. A subset of target regions is indicated. The colour map ranges from 0 to 0.01 and the top of the range is truncated. **d**, Schematic summarizing all major MOp outputs by area, layer and projection class on a whole-brain flat map (Extended Data Fig. 14). ACB, nucleus accumbens; AUD, auditory area; BAC, bed nucleus of the anterior commissure; CB, cerebellum; cc, corpus callosum; CLA, claustrum; DN, dentate nucleus; ENTl, entorhinal area, lateral part; FN, fastigial nucleus; GP, globus pallidus; GRN, gigantocellular reticular nucleus; GU, gustatory areas; HY, hypothalamus; HPF, hippocampal formation; IO, inferior olivary complex; IP, interposed nucleus; IRN, intermediate reticular nucleus; IZp, spinal cord intermediate zone; MARN, magnocellular reticular nucleus; MDRNd, medullary reticular nucleus, dorsal part; MOp-ll, primary motor area, lower limb; MOp-tr, primary motor area, trunk; OLF, olfactory areas; ORB, orbital area; PL, prelimbic area; POST, postsubiculum; PPN, pedunculopontine nucleus; PRNc, pontine reticular nucleus, caudal part; RE, nucleus of reuniens; RR, midbrain reticular nucleus, retrorubral area; SCm, superior colliculus medial zone; sp, spinal cord; SI, substantia innominata; SNr, substantia nigra, reticular part; SPVO, spinal nucleus of the trigeminal, oral part; SSp-ll, primary somatosensory area, lower limb; SSp-m, primary somatosensory area, mouth; SSp-tr, primary somatosensory area, trunk; STN, subthalamic nucleus; TTd, taenia tecta, dorsal; VTA, ventral tegmental area; ZI, zona incerta.

example, Tle4-L6 versus Ntsr1-L6; Extended Data Fig. 13d, left). Our integrated analyses revealed a comprehensive PN type-based output projection map of the MOp-ul (Fig. 2d, Extended Data Fig. 14).

## Inputs to MOp-ul

Next we mapped brain-wide inputs to MOp at region and cell-type levels from three types of tracing experiments (Fig. 3a, b): (1) injection of CTB (Extended Data Figs. 7, 10, 15) in wild-type mice; (2) injection

of Cre-dependent monosynaptic rabies viral tracers in the Cre lines described above plus three interneuron-selective lines (Pvalb, Sst and Vip); and (3) a modified tracing the relationship between input and output (TRIO) strategy combining AAVretro-Cre with monosynaptic rabies viral tracing to reveal inputs to projection target-defined neuron types[44] (Extended Data Fig. 16a). CTB tracing revealed the overall set of input areas projecting to MOp-ul, including somatomotor cortical regions (MOp, SSp, supplemental somatosensory area (SSs) and MOs) and related thalamic nuclei (ventral anterior–lateral complex (VAL), parafascicular nucleus (PF), posterior complex (PO) and ventral medial nucleus (VM)) (Extended Data Figs. 10, 15). Monosynaptic rabies tracing from Cre- and target-defined neurons showed highly similar global input patterns (Extended Data Figs. 13b, 15, 16a). Notably, rabies viral tracing labelled inputs to MOp-ul from pallidal (globus pallidus, external segment (GPe), globus pallidus, internal segment (GPi) and central amygdalar nucleus, capsular part (CEAc)) and other subcortical regions (superior central nucleus raphe (CS) and dorsal raphe (DR)) not seen with CTB (Extended Data Fig. 15).

Labelled inputs to MOp-ul were quantified across the entire brain in each CCFv3 region to create a weighted connectivity matrix (Fig. 3c, Source Data Fig. 3). Input arises mostly from cells in isocortex and thalamus (90.1%, 7.7%, respectively; Extended Data Fig. 13f, pie chart). Consistent with visual observation of highly similar brain-wide input patterns, unsupervised hierarchical clustering revealed only two main clusters (Fig. 3c). The first (larger) cluster comprised CTB and most Cre line rabies tracing datasets. The second cluster comprised all TRIO experiments and one Cre-dependent experiment (Foxp2-L6). The clusters differed significantly in in-degree (average $n$ = 91 versus 30 input regions, $P$ < 0.0001, two-tailed t-test), suggesting that on average a more restricted set of inputs is labelled from target-defined projection classes.

Together, our data suggest that the sets of regions providing input to Cre- and target-defined MOp-ul neuron types are similar, a surprising result given distinct axonal lamination patterns from cortical and thalamic sources[17,45] (Extended Data Fig. 16b). This result is nonetheless consistent with other recent findings that global input patterns mapped with rabies tracer methods are independent of starter cell type[46]. These results do not exclude the possibility of distinct presynaptic neuron types within a source area projecting to specific types within MOp. Notably, all input sources to MOp were also projection targets, indicating prevalent reciprocal areal connections with comparable strengths (Extended Data Fig. 10). In summary, integrated analyses of retrograde tracing experiments revealed a consensus brain-wide input map to MOp-ul (Fig. 3d).

To relate regional inputs and soma layer to single-cell morphology, we compared dendritic arbors of superficial (L2/3/4) and deep (L5) MOp pyramidal cells (Extended Data Fig. 17a–e): L5 neurons have larger and more complex basal trees, whereas superficial neurons have a greater proportion of their dendritic length distal from the soma.

## BARseq projection mapping

Cre driver line and target-defined tracing resolves PNs to subpopulations. These methods do not achieve single-cell resolution and require injections in many animals. BARseq achieves high-throughput projection mapping with cellular resolution using in situ sequencing of RNA barcodes[11]. Using BARseq, we mapped projections from 10,299 MOp neurons to 39 target brain areas (Fig. 4a). Projection patterns were enriched in somas in distinct sublayers, consistent with previous retrograde tracing results and were comparable to those obtained by single-cell tracing (Extended Data Fig. 18a–f, Supplementary Information). The large sample size also revealed additional statistical structure in projections (Supplementary Information, Extended Data Fig. 18g–k).

Hierarchical clustering revealed CT, L5 ET and two subclasses of IT PNs with (IT Str+) or without (IT Str−) projections to the striatum. Consistent with previous reports and with the above tract tracing

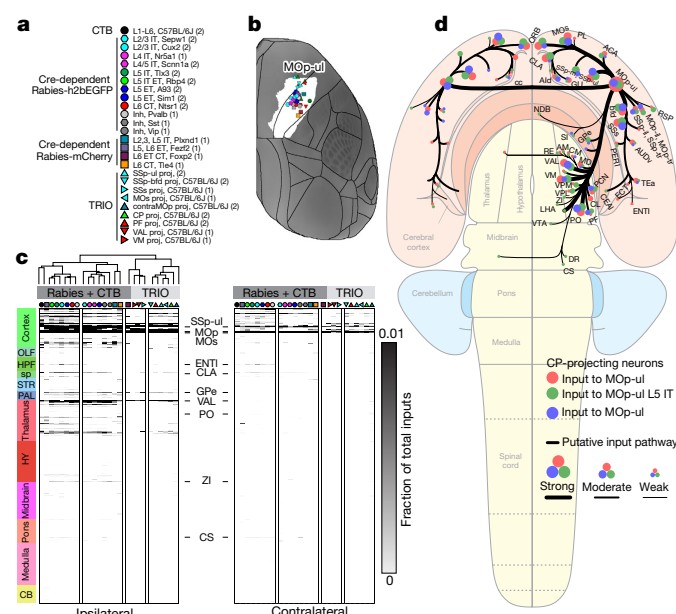

**Fig. 3 | Brain-wide inputs to MOp-ul by layer and class. a**, Key shows tracer types, mouse lines and layer and projection class selectivity for Cre driver lines used to label inputs to MOp neurons. Numbers in brackets represent the number of tracer injection experiments per type. Symbols and colour code are used in **b–d**. **b**, Injection sites are plotted on a top-down view of the right cortical hemisphere from CCFv3 with the MOp-ul delineation from Fig. 1 in white. Distance between injection sites is 622.4 ± 337.01 μm (mean ± s.d.). **c**, A directed, weighted connectivity matrix (26 × 628) to MOp from 314 ipsilateral and 314 contralateral targets for each of the mouse lines or tracers listed in **a**. Each row shows the fraction of the total input signal measured from a single experiment or the average when $n$ > 1. Rows are ordered by major brain division. We performed hierarchical clustering with Spearman rank correlations and complete linkages, splitting the resulting dendrogram into two major clusters (rabies + CTB and TRIO experiments). A subset of input regions is indicated. The colour map ranges from 0 to 0.01 and the top of the range is truncated. **d**, Schematic summarizing major MOp inputs by area (red), layer (L5 IT Tlx3+ neurons, green), and target-defined projection class (CP-projecting neurons, blue) on a whole-brain flat map. The sizes of dots represent relative connectivity strength. AId, agranular insular area, dorsal part; AM, anteromedial nucleus; AUDv, ventral auditory area; bfd, barrel field; CEAl, central amygdalar nucleus, lateral part; CM, central medial nucleus of the thalamus; inh, inhibitory; LHA, lateral hypothalamic area; NDB, diagno band nucleus; proj, projecting; RSP, retrosplenial area; Ssp-bfd, primary somatosensory area, barrel field; VPL, ventral posterolateral nucleus of the thalamus.

results, these four classes occupy distinct laminar positions (Fig. 4b, Extended Data Fig. 19a–c, Supplementary Information). Beyond these classes, further divisions by projection patterns (Methods) resulted in 18 subgroups with distinct laminar distributions (Fig. 4c, Extended Data Fig. 19d–k, Supplementary Information). Notably, each of the 11 sublayers—previously defined by single-target projections—could be uniquely identified by the top two enriched subgroups of BARseq PNs (Fig. 4d), supporting a sublaminar organization of neuron types defined by overall projection patterns.

Differential distribution across layers explains some of the diversity in IT projection patterns, but projections from cells in a sublayer remained highly structured. For example, 72–93% of IT neurons in L3 to L5b-d projecting to contralateral MOs (MOs-contra) also target ipsilateral MOs (MOs-ipsi), whereas only 32–50% of IT neurons without MOs-contra projections target MOs-ipsi (Fig. 4e, Extended Data Fig. 19l). This interdependence between contralateral and ipsilateral projections also generalized to other homotypic pairs of projections (Extended Data Fig. 19l). By contrast, in some cases the relationships between target pairs varied across sublayers. For example, in superficial

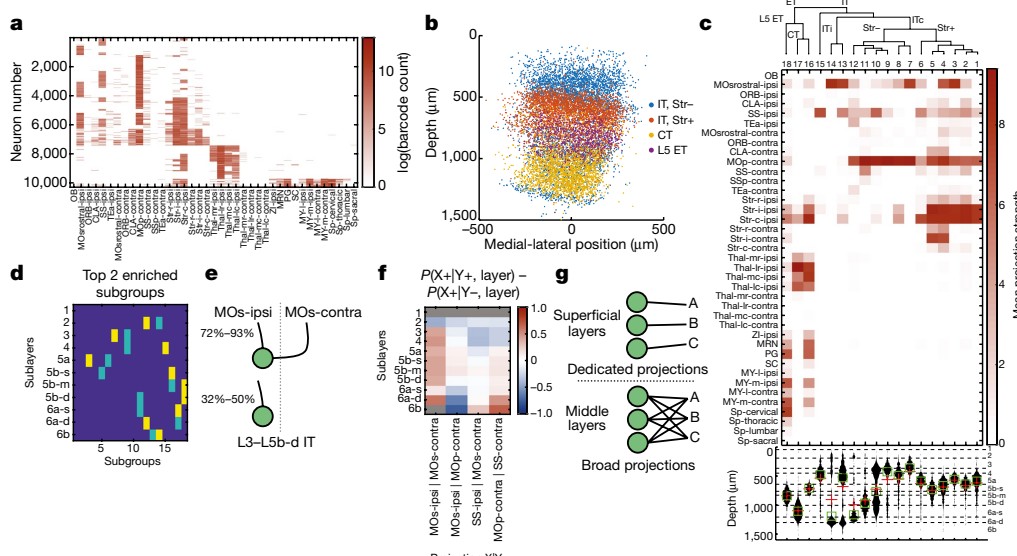

**Fig. 4 | Projection mapping with single-cell resolution using BARseq.**
**a**, log-transformed projection patterns of 10,299 neurons mapped in the motor cortex. Rows indicate single neurons and columns indicate projection areas. See Supplementary Information for a detailed list of dissection areas. Colour bar indicates log of barcode counts. **b**, Scatter plot of soma locations of the mapped neurons in the cortex. The *x*-axis indicates relative medial–lateral positions, and the *y*-axis indicates laminar depth. Neurons are coloured by major classes as indicated. **c**, Mean projection strengths of the indicated subgroups. Rows indicate projection areas and columns indicate subgroups. Top, dendrogram constructed from the distance of mean projection patterns, with major classes and splits indicated. Bottom, histograms of the laminar

distribution of subgroups. Sublayer identities as defined in Fig. 1d are indicated on the right, and sublayer boundaries are indicated by dashed lines. **d**, The most enriched subgroup (yellow) and the second most enriched subgroup (light blue) in each sublayer. **e**, Probabilities of projections to the ipsilateral MOs in IT neurons with (top) or without (bottom) contralateral MOs projections in layers L3 to L5b-d. **f**, The differences in probability for projection X in the indicated sublayer, conditioned on whether the neuron projects to Y. **g**, Cartoon model showing restricted IT projections in superficial layers and broad IT projections in deep layers. Thal, thalamus. OB, main olfactory bulb. Sp, spinal cord. ITc, intratelencephalic neurons with contralateral projections. ITi, intratelencephalic neurons with only ipsilateral projections.

layers (L2 for MOs-ipsi, and L2-4 for ipsilateral SSs (SSs-ipsi)), neurons with MOs-ipsi and SSs-ipsi projections were unlikely to also make contralateral projections to MOp-contra, whereas in the middle layers these ipsilateral projections had no predictive value about the corresponding contralateral projection (Fig. 4f). Similar relationships exist between pairs of contralateral projections (for example, MOp-contra and contralateral somatosensory area (SS-contra); Fig. 4f). These observations suggest that IT neurons in superficial sublayers (L2/3) have more dedicated and selective projections, whereas IT neurons in middle and deep sublayers (L5a, 5b and 6a) have broader projections (Fig. 4g). Therefore, the laminar distribution of neurons not only predicts the areas to which neurons project to, as revealed by retrograde labelling (Fig. 1d), but also affect higher-order statistics—that is, projection selectivity.

## Single-neuron projection patterns

We reconstructed 140 motor cortex PNs across all layers using genetic driver line-based sparse labelling, fluorescence micro-optical sectioning tomography (fMOST) imaging and registration to CCFv3[9]. We augmented this dataset with 121 single neuron reconstructions from the Janelia MouseLight Project[43], and a third set of reconstructions from fMOST images (*n* = 42 cells, 12 of which were previously published[47]), for a total of 303 single neurons. Given the difficulty in obtaining large numbers, we included cells across all of the MOp; 113 of the 303 are within the newly defined MOp-ul borders (Fig. 5a, Extended Data Fig. 20a).

We calculated the fraction of total axon length per brain region, summed across hemispheres, for each neuron (Fig. 5b, Source Data Fig. 5). To test whether single-neuron projection patterns vary across a continuum, we compared the distribution of differences in targets reached between all pairs with a randomized distribution (Extended Data Fig. 20b, c). The shuffled distribution is significantly narrower than the actual distribution, supporting the existence of distinct axon projection patterns at the single-cell level.

Unsupervised hierarchical clustering on the single cell axon and anterograde tracing data from Fig. 2 revealed 13 main clusters (C1–C13; Fig. 5b, c). We annotated clusters as CT, ET or IT on the basis of Cre line tracing data assigned to a cluster and/or brain-wide projection patterns. C1 comprises tracer experiments labelling projections from all layers or that include both IT and ET classes. C2 contains the CT Cre line tracer data and is significantly enriched for somas in L6. The CT cluster was further divided into three subclusters. Neurons in the largest subcluster (C2.1) have collateral projections to ventral posteromedial nucleus of the thalamus (VPM). Details, including specific target weights, can be found in Source Data Fig. 5.

MOp L5 ET neurons in C3–C5 project to subcortical structures with some collaterals in cortex and striatum (Fig. 5b, c, e). C3 and C4 differ in having dense projections to medulla (C3) or thalamus (C4), as previously reported[41]. Within C3, one subcluster (3.2) has stronger collateral projections to the spinal nucleus of the trigeminal, principal sensory nucleus of the trigeminal (PSV) parabrachial nucleus (PB) and facial motor nucleus, which are interconnected and involved in orofacial sensorimotor activities[48]. C3.2 also has stronger projections to medullar reticular nuclei, which mediates skilled forelimb motor tasks through connections with spinal cord[49]. C4 ET neurons terminate in midbrain (that is, midbrain reticular nucleus (MRN), superior colliculus (SC), anterior pretectal nucleus (APN) and periaqueductal grey (PAG)) and pons (that is, pontine grey (PG), tegmental reticular nucleus (TRN) and pontine reticular nucleus (PRNr)), in addition to collateralizing to thalamic nuclei (that is, VAL, VM, PO and PF), and are likely to relate to corticotectal and corticopontine PNs found in L5b-superficial (Fig. 1d). C4 neurons were also divisible into two subclusters, with C4.2 lacking projections to reticular thalamic (RT) and mediodorsal thalamic nuclei.

IT cells and Cre line tracer experiments are in C6–C13. IT clusters are differentiated by: (1) soma layer (enriched for L2/3 in C7, C10 and C11, and L4 in C7 and C13); (2) number of targets per experiment (C8

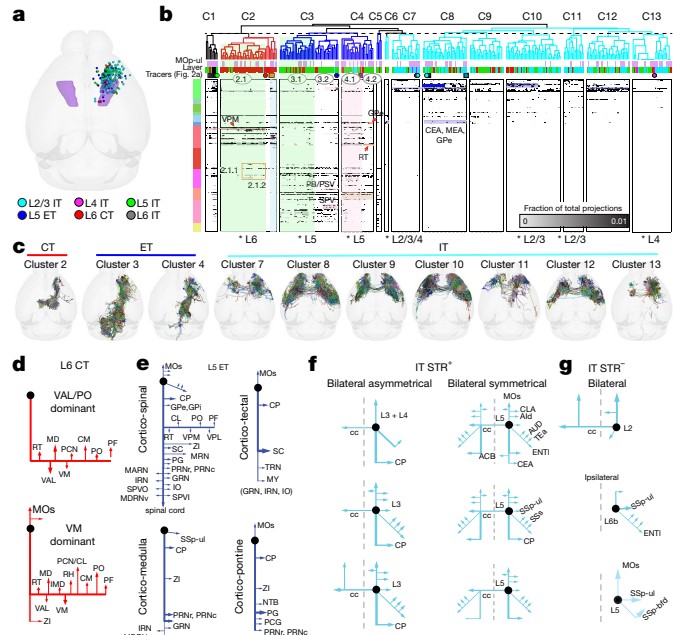

**Fig. 5 | Full morphological reconstructions reveal diverse single-cell projection motifs. a**, Soma locations (*n* = 303) plotted in a top-down view of CCFv3. MOp-ul delineation from Fig. 1 is shown in purple. **b**, Matrix showing the fraction of total axon projections from tracer (following the colour scheme from Fig. 2a) and single-cell reconstruction experiments to each of 314 targets across all major brain divisions. Columns show individual experiments. Rows show target regions ordered by major brain division. Hierarchical clustering and cutting the dendrogram as indicated with the dashed line revealed thirteen clusters. Some subclusters are indicated by the circled numbers. Cells from specific layers were significantly enriched within clusters (Fisher's exact test, two-sided, *P < 0.05). **c**, Top views of all single cells and their axons assigned to cluster 2 (CT), clusters 3 and 4 (ET) and clusters 7 to 13 (IT). Cells are registered to CCFv3, rendered in 3D, overlaid, and randomly coloured (see Extended Data Fig. 20d for single-neuron morphology). **d**–**g**, Schematics of single-cell projection targets following visual inspection and classification of motifs. **d**, Two L6 CT patterns were identified: VAL/PO and VM-dominant projections. **e**, Four L5 ET motifs are shown: cortico-spinal-, cortico-medulla-, cortico-tectal-and cortico-pontine-dominant patterns. **f**, **g**, IT motifs included cells with (+) (**g**) or without (−) (**h**) projections to the striatum (STR). **f**, Bilateral IT motifs include asymmetrical and symmetrical projection patterns. Each schematic shows major projection targets for the cell(s) indicated. cc, corpus callosum; MEA, medial amygdalar nucleus; IMD, intermediodorsal nucleus of the thalamus; MDRNv, medullary reticular nucleus, ventral part; RH, rhomboid nucleus; SPV, spinal nucleus of the trigeminal; SPVI, spinal nucleus of trigeminal nerve, interpolar part.

has significantly more non-zero targets than all other IT clusters; one-way ANOVA and Tukey's post hoc test, *P* < 0.0001); and (3) fraction of axon in specific targets (two-way repeated measures ANOVA, *P* < 0.0001 interaction effect of cluster × target area). For example, we found that C9 has more axonal projections to agranular insular area, dorsal part (AId), presumably via the rostral pathway (Supplementary Information), compared with C7, C8, C12 and C13 (Tukey's post hoc test, *P* < 0.05). Cells in C11 have more axon in medial prefrontal areas (that is, anterior cingulate area, ventral part (ACAv)), compared with C6, C9 and C12 (Tukey's post hoc test, *P* < 0.05). Finally, C12 cells project more extensively to other sensorimotor areas (that is, SSp-ul and SSs) than cells in C6, C9, C11 or C13 (Tukey's post hoc test, *P* < 0.05).

IT cells in C11 and C13 also have fewer axons in CP compared with C8–C10 and C12 (Tukey's post hoc test, *P* < 0.02), similar to IT Str⁻ and IT Str⁺ neurons identified with BARseq. C8 includes many L5 IT cells and has the most extensive collateral projections to other targets, including

some to central amygdalar nucleus (CEA) and GPe. By contrast, C7, C11 and C13, which are enriched for L2/3 and L4 neurons, project to a more limited set of targets, also consistent with BARseq data showing that IT neurons in superficial layers have more 'dedicated' projections.

We estimated the relative proportions of clusters and PN types in MOp by matching single-cell axon projections against the regional patterns from PHAL tracing. This problem is equivalent to a set of constrained, weighted, linear equations that can be solved by standard non-negative least-squares or bounded-variable least-squares optimization[50]. We excluded clusters with fewer than 15 neurons (C1, C5 and C6). Results converged with minimal error (less than 0.5% residual sum of squares) on the following compositions: 32% C2, 40% C4, 12% C8, 7.7% C9, 2.9% C11, 4.9% C12 and less than 1% for C3, C7, C9 and C13, which correspond to 40% ET, 32% CT and 28% IT.

### Diverse PN axon projection motifs

Single-cell analyses also revealed different levels of variability across projections for cells in the same cluster (Fig. 5c, Extended Data Figs. 20d). CT neurons (C2) are most like each other (average Spearman *R* = 0.66) compared with ET (C3–C5: *R* = 0.52, 0.51 and 0.56, respectively) and IT clusters (C6–C12: range 0.54–0.61 and C13: *R* = 0.66). Lower ET and IT correlation coefficients indicate more within-cluster diversity of axon targeting in these PN types.

We examined whether projection variability within a class might be constrained to a set of finer-scale structural motifs (in between 'every neuron is unique' and the projection class level). Among CT neurons, we describe two projection motifs (Fig. 5d): one strongly projecting to VM, the other to VAL and PO; both types also project to other thalamic nuclei, for example, mediodorsal nucleus of thalamus, lateral part (MDl), paracentral nucleus (PCN), central lateral nucleus (CL) and PF. We also observe four ET projection motifs (Fig 5e): (1) cortico-spinal, (2) cortico-medullary, (3) cortico-tectal and (4) cortico-pontine. IT Str⁺ neurons (Fig. 5f) can be further differentiated on the basis of ipsilateral versus bilateral striatal connections. Most ipsilateral-dominant IT Str⁺ cells are in L2/3 or L4 (8 out of 9 cells; Fig. 5f, left) and notably bilaterally asymmetric. L5 IT Str⁺ neurons (*n* = 3; Fig. 5f, right) displayed more bilaterally symmetric projections. Projections from IT Str⁻ cells are either ipsilateral only or had additionally or exclusively contralateral connections (Fig. 5g). IT Str⁻ cells with contralateral projections largely mirrored the projection patterns of their ipsilateral counterparts. These results suggest that the varying single cell axon projections may in part derive from definable finer-scale structural motifs.

### Discussion

Our study integrated data generated by diverse methods for anatomical labelling, imaging and computational analyses to generate a comprehensive overview of brain structure with cell-type resolution for a single mammalian brain region. This achievement includes accurate 3D border delineation, classification of more than two dozen PN types, refined laminar parcellation, anatomical classification of PN types, a multi-scale input–output wiring diagram, around 300 single neuron reconstructions, and approximately 10,000 single neuron projections traced by molecular barcoding.

Our study represents a coherent, multifaceted analysis of neuron types across nested levels of cortical organization (Fig. 6a; Extended Data Fig. 21). The resulting multi-scale input–output wiring diagram provides a high level of structural detail and establishes a foundational framework for determining the functional importance of cell types and circuits (Fig. 6b).

Despite substantial progress in cell-type censuses, a rigorous definition of PN types remains elusive. Some PN types are well aligned with transcriptomic types—for example, two transcriptomic types of TEa–ectorhinal area (ECT)–perirhinal area (PERI)-projecting neurons in L2 and L5 exist with distinguishable asymmetric or symmetric projection

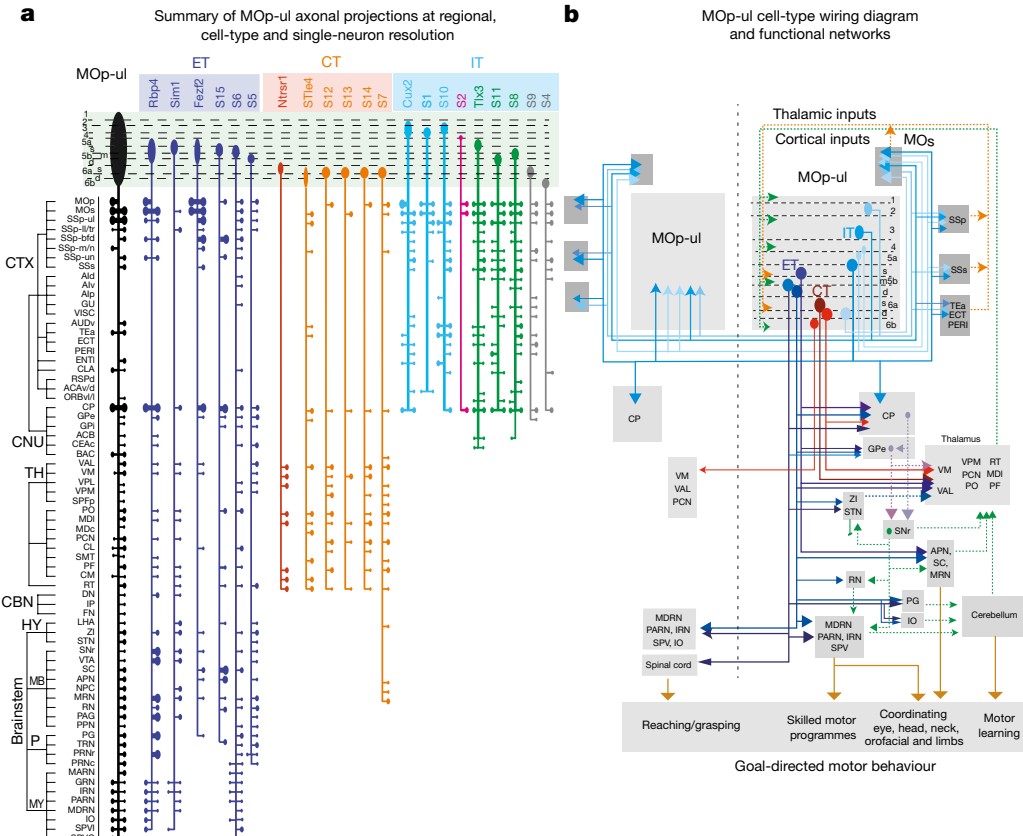

**a** Summary of MOp-ul axonal projections at regional, cell-type and single-neuron resolution

**b** MOp-ul cell-type wiring diagram and functional networks

**Fig. 6 | Cell type wiring diagram of the MOp-ul. a**, Summary of the output connections of multiple MOp-ul cell types (ET, CT and IT) compared to those of the MOp-ul as a whole (left, black). Along each vertical path, the outputs from one Cre line tracing or single cell reconstruction experiment (identified by the prefix S, followed by a number) are summarized. The outputs begin at top with the originating MOp-ul layer(s); branches perpendicular to the main vertical path that end in ovals represent ipsilateral (right) and contralateral (left) sites of termination, identified by the brain division abbreviations at left. Branch thickness and oval size represent relative connection strength. **b**, A summary wiring diagram of MOp-ul cell types and predicted functional roles. A subset of cortical and striatal projection patterns is shown from the diverse MOp-ul IT cell types (six IT cell types in L2–L6b). Three types of CT neurons are shown representing different combinations of thalamic targets and MOp-ul layers of origin. Three of four types of ET neurons are also shown, projecting to subcortical targets involved in different motor functions: (1) cortico-spinal outputs to the cervical spinal cord controlling goal-directed upper limb motor activities, such as reaching and grasping; (2) cortico-medullary projections to

and output from the reticular formation (for example, medullary reticular nucleus (MDRN)) are implicated in task-specific aspects of skilled motor programs[49]; (3) cortico-tectal projections to the SC are implicated in coordinating movements of eye, head, neck and forelimbs during navigation and goal-oriented behaviours (such as defensive and foraging behaviour) and (4) cortico-pontine projections to the pontine grey, which generates mossy fibres to the cerebellum (which is critically involved in associative motor learning). These ET neurons also generate collateral projections to other structures in the motor system, such as GPe, ZI, STN, RN and IO. ACAv/d, anterior cingulate area, ventral and dorsal part; AIp, agranular insular area, posterior part; AIv, agranular insular area, ventral part; CBN, cerebellar nuclei; CNU, cerebral nuclei; CTX, cortex; MB, midbrain; MDc, mediodorsal nucleus of thalamus, capsular part; NPC, nucleus of the posterior commissure; P, pons; ORBvl/l, orbital area, ventrolateral and lateral part; PARN, parvicellular reticular nucleus; SMT, submedial nucleus of the thalamus; SPFp, subparafascicular nucleus, pavicellular part; TH, thalamus.

patterns to their ipsilateral or contralateral targets, among several other examples[7,41,51]. However, mapping between PN types and transcriptome types is not always clear[9,52]. For example, we identified L6 CT VM-projecting neurons that differ from other CT neurons by their location in deep L6a and L6b (Fig. 1d). Spatial transcriptomics[51] also identified several L6 CT clusters distributed across top to bottom of L6; but how these anatomical and molecular types relate to each other remains to be determined. The correspondence between molecularly and anatomically defined PN types will be clarified by future studies and will probably require further method development[53].

Knowledge of evolutionary conservation and divergence of brain structures often yields insights into organizational principles. Previous cross-species comparisons of mammalian brains have largely focused on the macroscale, such as cortical areas and layers, leaving many open questions regarding what is and is not conserved. The joint molecular and anatomic identification of PNs provides a

higher resolution and more robust metric for cross-species translation. Although the primate cortex has more functionally distinct areas and potentially orders of magnitude larger cortical networks than in rodents, a PN-type-resolution analysis may reveal truly conserved core subnetworks and novel species innovations. The MOp provides a good starting point for such comparative studies, given the clearly recognizable conservation and divergence of forelimb structures and motor behaviours from rodents to humans.

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

¹Cold Spring Harbor Laboratory, Cold Spring Harbor, NY, USA. ²UCLA Brain Research and Artificial Intelligence Nexus, Department of Neurobiology, David Geffen School of Medicine, University of California Los Angeles, Los Angeles, CA, USA. ³USC Stevens Neuroimaging and Informatics Institute (INI), Keck School of Medicine of USC, University of Southern California, Los Angeles, CA, USA. ⁴Allen Institute for Brain Science, Seattle, WA, USA. ⁵Britton Chance Center for Biomedical Photonics, Wuhan National Laboratory for Optoelectronics, MoE Key Laboratory for Biomedical Photonics, Huazhong University of Science and Technology, Wuhan, China. ⁶HUST–Suzhou Institute for Brainsmatics, JITRI, Suzhou, China. ⁷Cajal Neuroscience, Seattle, WA, USA. ⁸Center for Neurobehavioral Genetics, Jane and Terry Semel Institute for Neuroscience and Human Behavior, Department of Psychiatry and Biobehavioral Sciences, David Geffen School of Medicine at UCLA, Los Angeles, CA, USA. ⁹Institute for Medical Engineering and Science, Department of Chemical Engineering, Picower Institute for Learning and Memory, Massachusetts Institute of Technology (MIT), Cambridge, MA, USA. ¹⁰Department of Physiology and Neuroscience, Zilkha Neurogenetic Institute, Keck School of Medicine of USC, University of Southern California, Los Angeles, California, USA. ¹¹Department of Neural and Behavioral Sciences, College of Medicine, Penn State University, Hershey, PA, USA. ¹²Center for Neural Informatics, Structures and Plasticity, Bioengineering Department and Krasnow Institute for Advanced Study, George Mason University, Fairfax, VA, USA. ¹³SEU–ALLEN Joint Center, Institute for Brain and Intelligence, Southeast University, Nanjing, China. ¹⁴Department of Neurobiology, Duke University School of Medicine, Durham, NC, USA. ¹⁵Department of Biological Sciences, University of Southern California, Los Angeles, CA, USA. ¹⁶Center for Neural Circuits and Sensory Processing Disorders, Zilkha Neurogenetics Institute (ZNI), Department of Physiology and Neuroscience, Keck School of Medicine, University of Southern California, Los Angeles, CA, USA. ¹⁷School of Optometry and Ophthalmology, Wenzhou Medical University, Wenzhou, China. ¹⁸School of Computer Engineering and Science, Shanghai University, Shanghai, China. ¹⁹Division of Biological Science, Neurobiology section, University of California San Diego, San Diego, CA, USA. ²⁰Department of Radiology, University of Pennsylvania, Philadelphia, PA, USA. ²¹These authors contributed equally: Rodrigo Muñoz-Castañeda, Brian Zingg, Katherine S. Matho, Xiaoyin Chen, Quanxin Wang. ✉e-mail: ascoli@gmu.edu; josh.huang@duke.edu; osten@cshl.edu; jharris@cajalneuro.com; HongWeiD@mednet.ucla.edu

# Methods

## Animal subjects

All animal procedures were performed under Institutional Animal Care and Use Committee (IACUC) approval (Allen Institute for Brain Science (AIBS), Cold Spring Harbor Laboratory (CSHL), University of Southern California (USC), MIT and Huazhong University of Science and Technology in China) in accordance with NIH guidelines. Mice had ad libitum access to food and water and were group-housed within a temperature- (21–22 °C), humidity- (40–51%), and light- (12-h light:dark cycle) controlled room in the vivariums of the institutes listed above. Male and female wild-type C57BL/6J mice at an average age of post-natal day (P)56 were purchased from Jackson Laboratories for histological, multi-fluorescent tract tracing and viral tracing experiments, and single-neuron reconstructions. The mouse lines used at different institutes for specific experiments are described below and listed in Supplementary Table 2.

## Cell-type atlasing

Cell-type atlasing was performed at the laboratory of P.O. (CSHL).

**Brain sample preparation and imaging of cell-type distributions.** Cre-reporter transgenic mice were created by crossing 'knock-in' Cre drivers with reporter mice (CAG-LoxP-STOP-LoxP-H2B-GFP) as described previously[20]. General procedures of brain extraction, histology and imaging methods were described previously[20,21,54]. Whole-brain imaging of Cre reporter lines was achieved using automated whole-brain serial two-photon tomography (STPT). The entire brain was coronally imaged[20,21,54] at an *xy* resolution of 1 μm and *z*-spacing of 50 μm. Whole-brain Neurotrace staining was performed with a modified iDISCO+ protocol[55] (R.M.-C. and P.O., manuscript in preparation).

**STPT cell counting.** Automatic cell counting in MOp-ul was done as previously described[20]. A convolutional neural network was trained using H2B-GFP nuclear signalling. First, we develop an unsupervised detection algorithm for cell detection based on structure tensor and connected components analyses. Results were used to automatically generate 270 random segmented image tiles from 3 different datasets (~1,350 cells), which were used as the ground truth (R.M.-C. and P.O., manuscript in preparation).

**ARA Nissl registration.** Two-dimensional ARA Nissl slices were registered onto the Allen CCF (https://biccn.org/standards/common-coordinate-frameworks-biccn) reference brain. In brief, ARA 2D slices were pre-aligned to a subset of CCF slices spaced 100 μm apart, producing a total of 132 slices as in the ARA (using a custom Python 3.7 script). After 2D alignment, a 3D affine transformation was applied followed by a 3D B-spline transformation (see 'Image registration'; Extended Data Fig. 2).

**Image registration.** Whole-brain 3D datasets were registered to the CCF reference brain. In brief, a 3D affine transformation was calculated first, followed by a 3D B-spline transformation. Similarity was computed using advanced Mattes mutual information metric in the Elastix 2.0 registration toolbox[56]. Two-dimensional datasets pre-registered to the ARA were initially aligned using the output transformations from the original ARA Nissl 2D alignment. Non-pre-registered 2D datasets were initially pre-aligned (see 'ARA Nissl registration' for description; Supplementary Video 1).

**Anatomical feature enhancing.** To improve whole-brain registration, both CCF and image series datasets were pre-processed to enhance intrinsic anatomical features (see below). Anatomical features in the reference brain were initially enhanced (custom Matlab R2018a scripts). Then, a Sobel operator was applied to reduce noise and computational cost during image registration (custom Python 3.7 scripts). Brain image datasets were enhanced following the same process (R.M.-C. and P.O., manuscript in preparation).

**Depth-based cluster analysis.** Cell soma coordinates were grouped every 25 μm from the pia after registration to CCF. Depth-based analysis of MOp organization was performed using unsupervised hierarchical clustering of soma depths distribution on the basis of injection projection patterns or cell-type (Fig. 1f). Proximity was computed using Euclidean distance with complete linkages. All cortical depths were later rearranged based on depth organization and layers were defined by grouping depths by cluster. Thus, layers were defined by adjacent depth belonging to the same cluster.

**High resolution image registration transformation.** After image registration, output transformations were used to generate high resolution registered datasets (custom Matlab R2018a scripts). We automatically generated the displacement field of the initial registration, which was used to compute the high-resolution registration transformations (Supplementary Video 2; R.M.-C. and P.O., manuscript in preparation).

**Cloud-based visualization and delineation with Neuroglancer.** Brains registered at high resolution were converted and stored in a 'precomputed' format in the Google Cloud Platform using Cloud-Volume (https://github.com/seung-lab/cloud-volume). Cloud-based visualization was done using Neurodata's fork (https://viz.neurodata.io/) of Google Neuroglancer WebGL-based viewer[27,57] (https://github.com/google/neuroglancer). Cloud-based delineation of MOp-ul was done using Neuroglancer's annotation tools on the high-resolution registered datasets (Supplementary Video 1).

**MOp-ul 3D rendering.** MOp-ul annotations were exported from Neuroglancer and converted to binary image files using custom scripts (Python 3.7). Cortical layers were delineated on the basis of cell types distribution. For depth-distribution analysis, MOp-ul was divided in 50-μm thickness bins equally spaced between pia surface and white matter. Finally, MOp-ul images were 3D rendered using ParaView (v5.8.1) software[58].

## Multi-fluorescent tracing and cell type-specific input–output viral tracing experiments

**Mouse Connectome Project.** The Mouse Connectome Project was carried out at the laboratory of H.-W.D.

**Tracer injection experiments.** The Mouse Connectome Project uses a variety of combinations of anterograde and retrograde tracers to simultaneously visualize multiple anatomical pathways within the same Nissl-stained mouse brain[3,13]. Triple anterograde tracing experiments involved three separate injections of 2.5% PHAL (Vector Laboratories, catalogue (cat.) no. L-1110, RRID:AB 2336656), and adeno-associated viruses encoding enhanced green fluorescent protein (AAV-GFP; AAV2/1.hSynapsin.EGFP.WPRE.bGH; Penn Vector Core) and tdTomato (AAV1.CAG.tdtomato.WPRE.SV40; Penn Vector Core). Retrograde tracers included CTB Alexa Fluor conjugates 647, 555 and 488 (0.25%; Invitrogen), Fluorogold (FG; 1%; Fluorochrome, LLC), and AAVretro-EF1a-Cre (AAV-retro-Cre; Viral Vector Core; Salk Institute for Biological Studies). Retrograde tracing from the spinal cord (Fig. 1b; Extended Data Fig. 4) was performed with AAVretro-hSyn-GFP-WPRE (Addgene, cat. no. 50465) and AAVretro-hSyn-Cre-WPRE (Addgene, cat. no. 105553) in Ai14 tdTomato Cre-reporter mice (Jackson Laboratories, stock no. 007914, aged 2–3 months). To further establish synaptic connectivity in downstream targets of MOp-ul (Extended Data Fig. 11), AAV-hSyn-mRuby2-sypEGFP (custom design, laboratory of B.K.L.) was used to label axons-of-passage with mRuby2 (red) and presynaptic puncta with EGFP (green). Patterns of synaptic innervation were further

demonstrated in Ai14 mice using injections of self-complementary (sc) AAV1-hSyn-Cre (Vigene Biosciences; $2.8 \times 10^{13}$ GC per ml), which is capable of anterograde transneuronal spread to post-synaptic targets[18,20].

To reveal mono-synaptic inputs to a projection defined neuronal populations (Fig. 3; Extended Data Fig. 16a), we used a modified TRIO strategy[44]. In brief, AAVretro-Cre was injected into a MOp downstream projection target (that is, caudoputamen) and Cre-dependent TVA- and RG-expressing helper virus (AAV8-hSyn-FLEX-TVA-P2A-GFP-2A-oG) and mCherry-expressing G-deleted rabies virus (produced by the laboratory of I. Wickersham at MIT) were injected into the MOp to label the MOp PN population (1st order) and their brain-wide monosynaptic inputs (2nd order).

All injection experiments in this study are listed in Source Data Fig. 2 and Source Data Fig. 3. No statistical methods were used to pre-determine sample sizes, but our sample sizes are similar to those reported in previous publications[3,13]. In most cases, anterograde tracing results are cross-validated by retrograde injections at anterograde axonal terminal fields, and vice versa. The procedures of stereotaxic surgeries, histology and immunohistochemical processing are described in the Supplementary Information.

**Imaging processing and data presentation.** Tissue sections were scanned with an Olympus VS120 slide scanning microscope using a 10× objective. Each tracer was visualized using appropriate fluorescent filters and whole tissue section images were stitched from tiled scanning into VSI image files. An informatics workflow was specifically designed to reliably warp, reconstruct, annotate and analyse the labelled pathways in a high-throughput fashion through our in-house image processing software Connection Lens[13,8], where each section was matched and warped to its corresponding atlas level of the ARA and the labelling was segmented. Threshold parameters were individually adjusted for each case and tracer, resulting in binary image output files for quantitative analysis. Adobe Photoshop was used to correct conspicuous artifacts in the threshold output files that would have spuriously affected the analysis. Results were recorded and output in a spreadsheet for statistical analysis and matrix visualization.

Atlas-registered TIFF image files were converted into JPEG2000 image format, while images with thresholding were aggregated into SVG images. All fluorescently labelled connectivity data are presented through the iConnectome viewer, the iConnectome Map Viewer, and published to the Data Repository Dashboard page, www.MouseConnectome.org. Quantified cell count files and projection matrix also are accessible from www.MouseConnectome.org.

## Allen Institute Mouse Brain Connectivity Atlas Project

**Tracer injections.** Whole-brain axonal projections from MOp-ul were labelled with AAV using the previously established Allen Mouse Brain Connectivity Atlas pipeline. Experimental methods and procedures have been described previously[4,17]. In brief, a pan-neuronal AAV expressing EGFP (AAV2/1.hSynapsin.EGFP.WPRE.bGH, Penn Vector Core, AV-1-PV1696, Addgene ID 105539) was used for stereotaxic injections into wild-type C57BL/6J mice. To label genetically defined populations of neurons, we used a Cre-dependent AAV that expresses EGFP within the cytoplasm of Cre-expressing infected neurons (AAV2/1. pCAG.FLEX.EGFP.WPRE.bGH, Penn Vector Core, AV-1-ALL854, Addgene ID 51502). For retrograde mono-synaptic whole-brain tracing of inputs to Cre-defined cell types in MOp-ul, we used a dual-virus strategy (S.Y. et al., manuscript in preparation and refs. [59,60]). A Cre-dependent rAAV helper virus co-expressing TVA receptor, rabies glycoprotein (G), and tdTomato in the cytoplasm of Cre-expressing infected neurons (AAV1-Syn-DIO-TVA66T-dTom-N2cG) was injected stereotaxically into MOp, followed 21 ± 3 days layer by another injection in the same location of a G-deleted, ASLV type A (EnvA) pseudotyped rabies virus expressing a nuclear GFP reporter (RV.CVS-N2c(deltaG)-H2bEGFP). Information on Cre driver lines is provided in Extended Data Table 2. Detailed

procedures for stereotaxic surgeries and histology are described in the Supplementary Information.

**Imaging and post-acquisition processing.** STPT imaging procedures were previously described[20,21] (TissueCyte 1000, TissueVision). In brief, following AAV tracer injections, brains were imaged at high $xy$ resolution (0.35 μm × 0.35 μm) every 100 μm along the rostrocaudal $z$-axis. Images of rabies tracer-labelled nuclei were also collected every 100 μm, but were imaged at 0.875 μm × 0.875 μm $xy$ resolution. Images underwent quality control and manual annotation of injection sites, followed by signal detection and registration to the CCFv3 through an informatics data pipeline[10,61] (IDP). The IDP manages the processing and organization of the images and quantified data for downstream analyses. The two key algorithms in the IDP are signal detection and image registration. For segmentation, high-threshold edge information was combined with spatial distance-conditioned low-threshold edge results to form candidate signal object sets. The candidate objects were then filtered based on their morphological attributes such as length and area using connected component labelling. In addition, high-intensity pixels near the detected objects were included into the signal pixel set. Detected objects near hyper-intense artifacts occurring in multiple channels were removed. The output is a full-resolution mask that classifies each pixel as either signal or background. An isotropic 3D summary of each brain is constructed by dividing each image series into 10 μm × 10 μm × 10 μm grid voxels. Total signal is computed for each voxel by summing the number of signal-positive pixels in that voxel. Each image stack is registered in a multi-step process using both global affine and local deformable registration to CCFv3 as previously described[10,61].

**Rabies-labelled starter cell counting.** Antibody-stained starter cells were scanned using a 10× objective lens and using a 4-μm step size on a Leica SP8 TCS confocal microscope using appropriately matched fluorescent filters. Images were auto-stitched from tiled scanning into TIFF image files and compiled into maximum intensity projection images for every section of the injection site. A cell-counting algorithm was used to initially identify starter cells from the injection site. Following automated identification of starter cells each section was then manually corrected using ImageJ[62] (v1.53).

Each image containing the injection site was adjusted for brightness and false-positive or false-negative starter cells were corrected using the Cell Counter tool. Starter cells were assigned to cortical layers based on DAPI staining patterns.

**Quantification of whole-brain anterograde projections from MOp-ul.** We generated a weighted connectivity matrix with data obtained from all anterograde tracer experiments[60] for Fig. 2. Experiments and data are provided in Source Data Fig. 2. Segmentation and registration outputs are combined to quantify signal for every voxel in CCFv3. To quantify signal per brain structure, segmentation results are combined for all voxels with the same structure annotation. We defined connection weight in these analyses as the fraction of total axon volume; that is, the axon volume segmented per each brain region divided by the total axon volume across all regions, excluding the injection site (MOp). We note that even with stringent quality control, informatically derived measures of connection weights can include artefacts (false positives), and the AAV-EGFP tracer reports signal from labelled axons, including passing fibres and synaptic terminals. For this reason, all targets ($n$ = 628 total, 314 per hemisphere) were visually inspected for presence of axon terminals, and a binary mask was generated to reflect 'true positives' for these regions. We applied the true positive binary mask to remove true negative connections and regions with only fibres of passage. We compiled a weighted matrix and performed comparative analyses across tracer datasets acquired from multiple laboratories (Allen, Z.J.H. and H.-W.D.). In the case of data from the

Z.J.H. laboratory, integration was straightforward as these experiments were directly registered to CCFv3 as in the Allen pipeline. The H.-W.D. laboratory data were mapped to CCFv3 by matching structure name. As the ontology of the CCFv3 is derived from the ARA, corresponding structures were easily identified for most regions.

**Quantification of whole-brain retrograde inputs to MOp-ul.** We generated a weighted connectivity matrix with data obtained from all retrograde tracer experiments for Fig. 3. Experiments and data are provided in Source Data Fig. 3. The total volume of detected signal was informatically derived for each brain structure in CCFv3, as described above for axon segmentation. In contrast to the heavily manual quality control for axonal projection false positives, we estimated segmentation false positives per CCFv3 structure for the rabies data by quantifying segmentation results from $n = 89–97$ 'blank' brains; that is, brains processed through the imaging and informatics pipeline without rabies-mediated GFP expression. The distribution of false positives per structure was used to set a minimum threshold of six standard deviations from the mean. Any structure not passing this threshold was set to zero. Following this threshold step, the input connection weights were defined as the fraction of fluorescent signal segmented per brain region divided by the total volume above threshold for this set of regions, again excluding the injection site (MOp).

**Clustering analyses based on connection weights.** Unsupervised hierarchical clustering was conducted using the online software, Morpheus, (https://software.broadinstitute.org/morpheus/). Proximity between clusters was computed using complete linkages with Spearman rank correlations as the distance metric. The clustering algorithm works agglomeratively: initially assigning each sample to its own cluster and iteratively merging the most proximal pair of clusters until finally all the clusters have been merged. The software program GraphPad Prism v9 was used for statistical tests.

### Cell distribution and tracing
**Genetic targeting of cortical pyramidal neuron lines to produce gene expression, cell-type-specific input and output whole-brain imaging datasets.** Cell distribution and anatomical tract tracing data were generated as part of the Comprehensive Center for Mouse Brain Cell Atlas in the laboratory of Z.J.H. at CSHL. Experimental methods and procedures have previously been described[8,16,63]. Knock-in mouse lines PlexinD1-2A-CreER, Fezf2-2A-CreER, Tle4-2A-CreER were generated[8]. Foxp2-IRES-Cre was generated by R. Palmiter (University of Washington, Seattle). We crossed CreER drivers (PlexinD1-2A-CreER, Fezf2-2A-CreER, Tle4-2A-CreER) with reporter mice expressing nuclear GFP or tdTomato (R26-CAG-LoxP-STOP-LoxP-H2B-GFP or R26-CAG-LoxP-STOP-LoxP-tdTomato, Ai14) for cell distribution data collection.

For both cell distribution and anterograde tracing analysis, these mice were induced with a 100 mg kg$^{-1}$ dose of tamoxifen (T5648, Sigma) dissolved in corn oil (20 mg ml$^{-1}$), administered by intraperitoneal injection at the appropriate age to enable temporal control of the CreER driver. In the case of the Foxp2-IRES-Cre line, cell distribution data was acquired based on a systemic AAV injection of AAV9-CAG-DIO-EGFP (UNC Viral Core) diluted in PBS ($5 \times 10^{11}$ viral genomes per mouse), injected through the lateral tail vein at 4 weeks of age with 100 µl total volume. Cell distribution datasets from ref. [8] were analysed in the MOp region. Experiments are detailed in ref. [8].

**Tracer injection experiments.** For anterograde tracing, AAVs serotype 8 (UNC Vector Core, Salk Institute for Biological Studies) were delivered by stereotaxic injection. Detailed procedures are described in Supplementary Information. In brief, cell-type specific anterograde tracing was conducted in the mouse knock-in CreER and Cre driver lines. CreER drivers were crossed with the Rosa26-CAG-LSL-Flp mouse converter line such that tamoxifen induction of CreER expression at a given time is converted to constitutive Flp expression for anterograde tracing with a Flp-dependent AAV vector. For anterograde tracing from Foxp2-IRES-Cre driver line, we used a Cre-dependent AAV to express EGFP in labelled axons. Three weeks after injection, mice were perfused with 4% PFA in PBS, brains were dissected out and processed for tissue collection.

For cell-type specific mono-trans-synaptic rabies tracing of inputs, in animals aged approximately 1 month, a Cre-dependent starter virus expressing TVA, EGFP and the rabies glycoprotein was delivered in MOp-ul, followed three weeks later, by the enVA-pseudotyped glycoprotein-deleted rabies virus, all administered with a pulled glass pipette as specified below. In the case of CreER drivers, the starter virus injection was followed by tamoxifen induction two and seven days after injection. Seven to 10 days after injection of the mono-trans-synaptic rabies virus, mice were perfused with 4% PFA in PBS, brains were dissected out and processed for tissue collection. We used the whole-brain STPT (TissueCyte 1000, TissueVision) pipeline to collect whole-brain images as described by the P.O. laboratory[20,21].

**Microscopy imaging of cell-type-specific input mapping.** Imaging from serially mounted sections was performed using 5× objective on a Zeiss Axioimager M2 System equipped with MBF Neurolucida Software (MBF Bioscience). To image starter cells, sections encompassing the injection site were imaged using a 20× objective with a 5-µm step-size on a Zeiss LSM 780 or 710 confocal microscope (CSHL St Giles Advanced Microscopy Center) using matched fluorescent filters. Images were auto-stitched from tiled scanning into TIF image files and compiled into maximum intensity projection images for sections encompassing the injection site. Input cells were manually annotated within the serial sections to extract their position within the dataset. We matched the serial sections to the corresponding sections from CCFv3. Then, we placed fiduciary landmarks on both data and CCFv3 sections for warping conducted using moving least squares in Fiji/ImageJ.

**Cell type specific whole-brain image dataset presentation.** Cell type specific anterograde viral tracing data generated (high resolution STPT images and registration to CCFv3) are available through the Mouse Brain Architecture Cell Type project (http://brainarchitecture.org/cell-type/projection). Cell-type-specific anterograde viral tracing, cell distribution and input tracing image datasets are available through the Brain Image Library (https://www.brainimagelibrary.org/). Cell distribution and anterograde tracing image datasets can also be viewed as image sets registered to the Allen CCF by the P.O. laboratory using Neuroglancer (https://github.com/google/neuroglancer). Links to these various portals can be found in the metadata tabs in Supplementary Tables 3 and 4.

### Dendritic morphology analysis
Dendritic morphology analysis was carried out at the H.-W.D. and X.W.Y. laboratories at UCLA. Several consortium partners in this project contributed two neuronal reconstruction datasets (that is, UCLA, USC and AIBS; Extended Data Figure 17). Both entailed sparse labelling of layer 2–5 pyramidal neurons using similar though distinct methodologies. The UCLA and USC contribution crossed Etv1-CreERT2 (layer 5-specific) and Cux2-CreERT2 (layers 2-4) mice with the Cre-dependent MORF3 (mononucleotide repeat frameshift) genetic sparse-labelling mouse line[64]. The MORF3 reporter mouse expresses a farnesylated V5 spaghetti monster fusion protein[65] from the *Rosa26* locus when both the LoxP flanked transcriptional STOP sequence is removed by Cre and when stochastic-mononucleotide repeat frameshift occurs[66]. After perfusion, the tissue was cut into 500-µm-thick coronal slices, iDISCO+ cleared[55] with a MORF-optimized protocol, stained with rabbit polyclonal anti-V5 antibody (1:500) followed by AlexaFluor 647-conjugated goat anti-rabbit secondary antibody (1:500) and NeuroTrace. Sections

were imaged via a 30× silicone oil immersion lens with 1-μm *z* step on a DragonFly spinning disc confocal microscope (Andor). These tissue generation and processing methods are described in ref. [64]. Composite images of neurons were viewed with Imaris image software, manually reconstructed with Aivia reconstruction software (v.8.8.2, DRVision), and saved in the non-proprietary SWC digital morphology file format[67].

The AIBS contribution crossed Cux2-CreERT2, Fezf2-CreER (layer 5-specific), and Pvalb-T2A-CreERT2 (layer 5) mice with the TIGRE-MORF (Ai166) fluorescent reporter line, which expresses farnesylated EGFP from the TIGRE locus[64]. Following tissue fixation, brains were processed and imaged using the fMOST method. Labelled neurons were reconstructed with Vaa3D software in a semi-automated, semi-user defined fashion[68], using the TeraFly and TeraVR modules enabling a virtual reality reconstruction environment, and reconstructions were saved as SWC files.

Reconstructions from both datasets were analysed concurrently by the H.-W.D. laboratory. Geometric processing of the reconstructions was performed with the Quantitative Imaging Toolkit (http://cabeen.io/qitwiki), allowing us to isolate the basal dendritic tree for analysis, and to render sample visualizations (Extended Data Fig. 17a). The modified SWC files were imported into NeuTube and morphometrics were obtained using L-Measure[69]. Since tissue preparation and data acquisition techniques can have significant effects on certain morphometric properties[70], only measures that are insensitive to these effects were used in the present analyses. These measures were number of primary dendrites, remote bifurcation amplitude and tilt angles, branch order, branch path length, tortuosity, arbor depth, height and width, Euclidian distance, total length, partition asymmetry, path distance, terminal degree and terminal segments length. Data outputs were normalized by dividing all values within each dataset by the mean value of all layer 2–4 neurons for each morphometric. Principal component analysis was run on the data, and the first two components were plotted to create a low-dimension scatter plot of the data (Extended Data Fig. 17b). Wilcoxon signed rank tests were applied to all measures comprising the loadings for these two components, with the comparisons made between superficial (2–4) versus deep (5) layers (Extended Data Fig. 17c); for the comparisons reported here the two datasets (AIBS and USC–UCLA) were not pooled together. A Sholl-like analysis was performed on the reconstructions to assess the distribution of dendritic distance as a function of relative path distance from the soma (Extended Data Fig. 17d). Moreover, we carried out a comparative analysis of persistence diagram vectors[71] of superficial versus deep neurons for both datasets (Extended Data Fig. 17e).

## High-throughput projection mapping at single-cell resolution with BARseq

**BARseq data collection and processing.** BARseq was carried out by the A.M.Z. laboratory at CSHL. Animals injected with Sindbis (see Supplementary Information for details) were sacrificed and dissected as described previously[72] for BARseq (see Supplementary Information, tables 6, 7 for details). Pre-processing of data (see Supplementary Information for details) resulted in 10,299 projection neurons for further analysis.

**Data analysis.** Raw projection barcodes were first normalized by spike-in counts, and further normalized between the two brains so that neurons with non-zero counts in each projection area have the same mean across the two brains. We then performed hierarchical *k*-means clustering on log-transformed and spike-in corrected projection strengths to identify the major classes. However, this clustering did not identify small clusters with distinct laminar positions. To find subclusters with distinct laminar distributions, we used a second clustering method based on binary projection patterns. From a population of neurons, we first split off one subcluster with a particular binary projection to up to three brain areas. For example, a subcluster can

be defined as having projections to the contralateral primary motor cortex, the ipsilateral caudal striatum, but not the caudal medial section of the ipsilateral thalamus. These projections were chosen to maximize the reduction in the entropy of the laminar distribution of neurons. This process was then iterated over the two resulting subclusters, until no subclusters resulted in statistically significant reduction in entropy (*P* < 0.05 without multiple-testing correction). This process resulted in many clusters, some of which may have similar laminar distributions. We then built a dendrogram based on the distance in projection space among the resulting clusters and iteratively combined subclusters similar in laminar distribution. Two subclusters were considered similar in laminae if differences in their laminar distributions were not statistically significant (*P* < 0.05 using rank-sum test with Bonferroni correction) and their median laminar positions were within 200 μm. This process was iterated over each split, starting from ones between the closest leaves or branches. We stopped combining clusters at the level of major classes.

To compare BARseq dataset to single-cell tracing, we randomly down-sampled BARseq dataset to the same sample size as a subset of the single-cell tracing dataset (~160 neurons). We further combined ipsilateral and contralateral cortical areas and combined all samples of the same non-isocortex brain divisions together. This resulted in an axonal resolution that can be compared to the single-cell tracing dataset. We then combined this down-sampled and low resolution BARseq dataset with the traced neurons and analysed the joint dataset. *t*-distributed stochastic neighbour embedding (*t*-SNE) was performed in MATLAB. Clustering was performed using two layers of Louvain community detection[73] in MATLAB (R2018a).

Matching BARseq clusters to single-cell tracing clusters was done using the common axonal resolution, but full-size BARseq dataset using MetaNeighbor[74]. To test the homogeneity of clusters, we down-sampled the datasets with replacement to different sizes (1,000 random samples per cluster size) and calculated the correlation between the down-sampled cluster centroids to the full-data cluster centroids.

Raw bulk sequencing data are deposited at Sequence Read Archive (SRR12247894). Raw in situ sequencing images are deposited at Brain Image Library. Processed projection data and in situ sequencing data are available from Mendeley Data (https://doi.org/10.17632/tmxd-37fnmg.1).

## Single-neuron reconstructions
Single-neuron reconstruction data were produced at the AIBS, Huazhong University of Science and Technology (HUST) and the SEU–AIBS Joint Center.

**Animal subjects.** Male and female transgenic mice at an average age of P56 were used for all experiments (viral tracer and single neuron reconstructions). For the AIBS project, Cre reporter lines are listed in Supplementary Table 2, and include drivers: Gnb4-IRES2-CreERT2, Fezf2-CreER, Cux2-CreERT2, Pvalb-T2A-CreERT2, Sst-Cre, and Cre-dependent EGFP reporters: Ai139 or Ai166[9]. Induction of CreERT2 driver lines was done by administration via oral gavage of tamoxifen (50 mg ml⁻¹ in corn oil) at original (0.2 mg per g body weight) or reduced dose for one day in an adult mouse. The dosage for mice age P7–P15 is 0.04 ml. Mice were transcardially perfused with fixative and brains collected more than 2 weeks after tamoxifen dosing.

**Imaging and post-acquisition processing.** Imaging and post-acquisition processing was carried out at HUST. All tissue preparation has been described previously[75]. Following fixation, each intact brain was rinsed three times (6 h for two washes and 12 h for the third wash) at 4 °C in a 0.01 M PBS solution (Sigma-Aldrich). The brain was subsequently dehydrated via immersion in a graded series of ethanol mixtures (50%, 70% and 95% (vol/vol) ethanol solutions in distilled water) and the absolute ethanol solution 3 times for 2 h each at 4 °C.

After dehydration, the whole brain was impregnated with Lowicryl HM20 Resin Kits (Electron Microscopy Sciences cat. no.14340) by sequential immersions in 50, 75, 100 and 100% embedding medium in ethanol–2 h each for the first three solutions and 72 h for the final solution. Finally, each whole brain was embedded in a gelatin capsule that had been filled with HM20 and polymerized at 50 °C for 24 h.

Whole-brain imaging was performed using a fMOST system. The basic structure of the imaging system is a combination of a wide-field upright epi-fluorescence microscopy with a mechanic sectioning system. This system runs in a wide-field block-face mode but updated to obtain better image contrast and speed and thus enables high throughput imaging of the fluorescent protein-labelled sample (manuscript in preparation). A block-face fluorescence image across the whole coronal plane ($xy$ axes), then the top layer is removed ($z$ axis) with a diamond knife, exposing next layer, and the sample is imaged again, repeating the process. The thickness of each layer is 1.0 µm. In each layer imaging, we used a strip-scanning ($x$ axis) model combined with a montage in the $y$ axis to cover the whole coronal plane[76]. The fluorescence, collected using a microscope objective, passes a bandpass filter and is recorded with a TDI-CCD camera. We repeat these procedures across the whole sample volume to get the required dataset.

The objective used is a 40× water-immersion lens with numerical aperture (NA) 0.8 to provide a designed optical resolution (at 520 nm) of 0.35 µm in the $xy$ axes. The imaging gives a sample voxel of 0.35 × 0.35 × 1.0 µm to provide proper resolution to trace the neural process. The voxel size can be varied upon difference objective. Other imaging parameters for GFP imaging include an excitation wavelength of 488 nm, and emission filter with passing band 510–550 nm.

**Full neuronal morphology reconstruction.** This was carried out at AIBS and the SEU–AIBS joint Center. Vaa3D, an open-source, cross-platform visualization and analysis system, was used to reconstruct neuronal morphologies as described in detail recently[77]. Critical modules were developed and incorporated into Vaa3D for efficient handling of the whole-mouse brain fMOST imaging data, that is, Tera-Fly[77] and TeraVR[24]. TeraFly supports visualization and annotation of multidimensional imaging data with virtually unlimited scales. The reconstructors can flexibly choose to work at a specific region of interest with the desired level of detail. The out-of-core data management of TeraFly allows the software to smoothly deal with terabyte-scale of data even on a portable workstation with normal RAM size. Driven by virtual reality (VR) technologies, TeraVR is an annotation tool for immersive neuron reconstruction that has been proved to be critical for achieving precision and efficiency in morphology data production. It creates stereo visualization for image volumes and reconstructions and offers an intuitive interface for the reconstructors to interact with such data. TeraVR excels at handling various challenging yet constantly encountered data situations during whole-brain reconstruction, such as noisy, complicated or weakly labelled axons.

Trained reconstructors used the Vaa3D suite of tools to complete their reconstructions. Completion was determined typically when all ends had well-labelled, enlarged boutons. A final quality-checking procedure was always performed by at least one more experienced annotator using TeraVR who reviewed the entire reconstruction of a neuron at high magnification, paying special attention to the proximal axonal part or a main axonal trunk of an axon cluster, where axonal collaterals often emerge and branches are more frequently missed due to the local image environment being composed of crowded high contrasting structures. To finalize the reconstruction, an auto-refinement step fit the tracing to the centre of fluorescent signals. The final reconstruction file (SWC) is a single tree without breaks, loops, or multiple branches from a single point.

**Registration of fMOST-imaged brains to Allen CCFv3.** We performed 3D registration of each fMOST image series (that is, the subject) to the CCFv3 average template (that is, the target[10]) using the following steps[9]: (1) fMOST images were down-sampled by 64 × 64 × 16 ($x × y × z$) to roughly match the size of the target brain; (2) 2D stripe-removal was performed using frequency notch filters; (3) approximately 12 matching landmark pairs between subject and target were manually added to ensure correct affine transformation that approximately aligned the orientation and scales; (4) Affine transformation was applied to minimize the sum of squared difference of intensity between target and subject images; (5) intensity was normalized by matching the local average intensity of subject image to that of target image; (6) a candidate list of landmarks across CCF space was generated by grid search (grid size = 16 pixels); and finally (7) our software searched corresponding landmarks in the subject image and performed local alignment. CCF-registered single neuron reconstructions were visualized using Brainrender[78].

**Quantification of whole-brain single-neuron projections from MOp-ul.** We generated a weighted connectivity matrix with data obtained from all single-neuron full morphology reconstruction experiments for Fig. 5. Experiment metadata and data are provided in Source Data Fig. 5. Reconstruction and registration outputs were again combined to quantify axon reconstructed for every CCF voxel, and combined for all voxels within the same CCF structure to generate total axon volume per brain structure for each single reconstructed cell. For Fig. 5, we summed voxels from the same structure across hemispheres to match the data format obtained from MouseLight MOp reconstructions, then calculated the fraction of total signal per structure.

**fMOST data analysis pipeline.** This data analysis was carried out at HUST, resulting in Fig. 5, sample nos. 193377 and 193663.

**Data collection.** PlexinD1-2A-CreER, Fezf2-2A-CreER mice[8] were generated in the laboratory of Z.J.H. and were crossed with Rosa26-loxp-stop-loxp-flpo mice. We used adult double-positive hybrid mice aged 2–3 months for experiments. Each of these mice received injection of 50 nl of flp-dependent pAAV-EF1a-fDIO-TVA-GFP virus (8 × 1012 genome copies per ml; UNC Vector Core) in the MOp. Three days later, the mice were induced intraperitoneally with a small amount of tamoxifen (T5648, Sigma, dissolved in corn oil, diluted at a concentration of 5 mg ml$^{-1}$, and the injection dose per mouse was 10 g ml$^{-1}$), and the virus was expressed in brain for 5 weeks. The whole-brain images were collected using the fMOST system following similar procedures as described above. The objective used was a 20× water-immersion lens with NA 1.0, to provide a designed optical resolution (at 520 nm) of 0.35 µm in the $xy$ plane. The imaging gives a sample voxel of 0.32 × 0.32 × 1.0 µm to provide proper resolution to trace the neural process. The voxel size can be varied with different objectives.

**Data analysis pipeline.** The fMOST datasets have two colour channels. The green channel (excitation wavelength of 488 nm, and emission filter with passing band 510–550 nm) containing fluorescent protein signals from labelled neurons is used to reconstruct neuronal morphology. The red channel (excitation wavelength of 561 nm and emission filter with longpass band of 590 nm) containing propidium-iodide (PI) signal with clear contours of most brain regions, was used to map original images to the Allen CCF space[79]. We have built a data analysis pipeline to perform neuron reconstruction and spatial mapping.

We used GTree software to reconstruct neuronal morphology with human–computer interaction[80]. GTree is an open-source graphical user interface tool, it offers a special error-screening system for the fast localization of submicron errors and integrates some automated algorithms to significantly reduce manual interference. To random access image blocks from brain-wide datasets, the original image (green channel) was pre-formatted to TDat, an efficient 3D image format for terabyte- and petabyte-scale large volume image[71].

GTree has a plugin to import TDat formatted data, and save reconstructions with original position in SWC format. All reconstructions were performed back-to-back by experienced technician and checked by neuroanatomists.

We used BrainsMapi to complete the 3D registration[80]. Specifically, the image of the red channel was down-sampled to an isotropic 10-μm resolution consistent with the CCFv3. We conduct the registration by several key steps including the initial position correction, regional feature extraction, linear and nonlinear transformation and image warping. Among them, a set of anatomically invariant regional features are extracted manually using Amira (version 6.1.1; FEI) and automatically using DeepBrainSeg[81]. Based on these, the unwarping neuron reconstructions can be accurately transformed to the CCFv3.

### Axonal projection analysis

Some axonal projection analyses were carried out at the laboratory of G.A.A. The brain-wide, single-neuron axonal projections from MOp came from three distinct sources: Janelia MouseLight, fMOST processed and reconstructed at the AIBS, and fMOST processed and reconstructed at the SEU-Allen Center in Nanjing. Each reconstruction from all three datasets was provided with a point-by-point reporting of the regions targeted by each neuron. These were the same data analysed in Fig. 5 of the main text. Exclusive-or (XOR) pairwise comparisons were used to quantify the projection differences between two neurons. The targeted regions were then fully shuffled to produce a randomized distribution consistent with the regional projection patterns, corresponding to the 'null' hypothesis of continuous targeting patterns at the single-cell level. The distribution of pairwise XOR distances of the shuffled data was then contrasted with the real pairwise distribution, which enables discernment of how much of the real distribution is accounted for by chance. To this end, given the non-normality of these distributions, we performed a one-tail Levene test[82] to ascertain whether the variance of the experimental distribution was significantly larger than that of the shuffled distribution.

To estimate the relative proportions of the 10 clusters containing 15 or more neurons, we matched their respective single-cell axonal patterns against the regional patterns from PHA-L anterograde tracing across all target regions. Specifically, the problem is equivalent to a set of constrained, weighted, linear equations that can be solved numerically by standard non-negative least-square (NNLS) or bounded-variable least-squares (BVLS) optimization. The NNLS algorithm solves the linear least squares problem[83] $\arg \min_x ||Ax - b||2$ with the constraint $x \geq 0$. The BVLS variant[84] minimizes the same objective function, but subject to explicit boundary conditions. We used the respective R implementations nnls[85] and bvls[86]. Boundary conditions for bvls were 0.01 for lower bound and 1 for upper bound. The results were consistent between the two methods.

### Data collection

Several microscopic methods were used to collect fluorescent imaging data: (1) epifluorescence images were collected with the Olympus VS120 fluorescence microscope running Olympus VS-Desktop v2.9; (2) high-resolution confocal images were captured using an Andor DragonFly 202 spinning disc confocal microscope running Fusion v2.1.0.81 software; (3) lightsheet images were captured with a LifeCanvas lightsheet microscope running SmartSPIM Acquisition Software 2019V3 and oblique light-sheet tomography (OLST version 1) running custom open source software (TissueCyte 1000, TissueVision); (4) 3D fluorescently labelled pathway images were collected using STPT instruments with TissueVision software; (5) single-neuron morphology data were collected using fMOST; (6) BARseq data were collected using an Olympus IX81 microscope with a Crest X-light v2 spinning disc confocal, an 89north LDI 7-channel laser, and a Photometrics Prime BSI camera. Image acquisition was controlled through micro-manager. STPT images at the AIBS were processed using the Allen informatics data pipeline

(IDP), which manages the processing and organization of the images and quantified data for analysis and display in the web application as previously described[4,61]. STPT images at CSH were processed with custom open source OpenSTPT software.

### Ethics oversight

Ethical oversight of experimental procedures was performed by the Institutional Animal Care and Use Committee (IACUC) of the CSHL, USC, Allen Institute, UCLA, UCSD, MIT, Penn State University and the Institutional Ethics Committee of Huazhong University of Science and Technology.

### Reporting summary

Further information on research design is available in the Nature Research Reporting Summary linked to this paper.

## Data availability

All imaging data are available through the archive Brain Imaging Library (https://www.brainimagelibrary.org). Figure-specific datasets are accessible through the Github site (https://doi.org/10.5281/zenodo.5146390). Cell-type-specific anterograde viral tracing data generated (high resolution STPT images and registration to CCFv3) are available through the Mouse Brain Architecture Cell Type project (http://brainarchitecture.org/cell-type/projection). Cell-type-specific anterograde viral tracing, cell distribution and input tracing image datasets are available through the Brain Image Library (https://www.brainimagelibrary.org/). Cell distribution and anterograde tracing image datasets can also be viewed as image sets registered to the Allen CCF by the P.O. laboratory using Neuroglancer (https://github.com/google/neuroglancer). Links to these various portals can be found in the metadata tabs in Source Data Fig. 2 and Source Data Fig. 3. Viral tracing and most anterograde tracing data (including high-resolution STPT images, segmentation, registration to CCFv3, and automated quantification of injection size, location and distribution across brain structures) are available through the Allen Mouse Brain Connectivity Atlas portal (http://connectivity.brain-map.org/). When available, direct links are provided in Source Data Fig. 2 on the metadata tab. For both AAV and transsynaptic rabies viral tracing, we also provide links to CCF-registered data files (http://download.alleninstitute.org/publications/) and to download original images through the Brain Image Library (https://www.brainimagelibrary.org/). These links can be found on the metadata tabs in Supplementary Tables 3, 4. The iConnectome Viewer and iConnectome Map Viewer will be accessible from the data repository dashboard page (http://brain.neurobio.ucla.edu/repository). Triple anterograde and retrograde tracer and viral labelling ARA-registered data are available at the UCLA BRAIN downloads page: http://brain.neurobio.ucla.edu/publications/downloads. Original fMOST image datasets are available to download through the Brain Image Library (https://www.brainimagelibrary.org/). Links to access the final reconstruction files (http://download.alleninstitute.org/publications/, with and without registration to CCF) are also provided in Source Data Fig. 5 on the metadata tab. Source data are provided with this paper.

## Code availability

All code used in this study is available through the Github site https://doi.org/10.5281/zenodo.5146390. iConnectome Viewer and iConnectome Map Viewer are accessible from the data repository page hosted on http://brain.neurobio.ucla.edu/mouseconnectome. The data repository dashboard page is available at http://brain.neurobio.ucla.edu/Dinoskin/page/dashboard. Public code repositories are stored in GitHub (https://github.com/BICCN/AnatomyCompanion). Processing scripts for in situ sequencing images, processed data,

annotated BARseq dissection images and analysis code are available from Mendeley Data (https://doi.org/10.17632/tmxd37fnmg.1). Specialized software used in this study includes: ImageJ 1.53k14 / 26, Morpheus (2021), GraphPad Prism v9, Neurolucida Software (MBF Bioscience), Quantitative Imaging Toolkit (http://cabeen.io/qitwiki), Vaa3D software, TeraVR, TeraFly v4.001, Matlab R2018a, Python 3.7, ParaView 5.8.1, R 4.0.2, Elastix 2.0 and GTree software (https://github.com/GTreeSoftware/GTree).

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

**Acknowledgements** B.Z., H.H., H.-W.D., I.B., L. Gao, L. Gou, L.K., M.S.B., M.Z. and N.N.F. thank K. Cotter, L. Gacia, D. Lo, T. Boesen, C. Cao, M. Becerra, M. Fayzullina and C. Mun for their technical and informatics support. Their work was supported by NIH U01MH114829 (to H.-W.D., G.A.A. and B.K.L.), R01MH094360 (H.-W.D.), U19MH114821 (Z.J.H. and P. Arlotta) and U01MH114831 (J. Ecker/E. Callaway). D.W.W., S.M.A. and G.A.A. gratefully acknowledge the assistance of T. Ferreira in accessing the API for batch downloading the Janelia MouseLight neuron dataset and for providing constructive feedback on the corresponding analysis. Their work was supported by NIH U01MH114829 (to H.-W.D., G.A.A. and B.K.L.), R01NS39600 (to G.A.A.), and R01NS86082 (to G.A.A. and D. Cox). A.C., F.D., H. Zeng, J.A.H., K.E.H., M.N., M.H., P.A.G., P.L., P.R.N., Q.W., S.Y., W.W., and Yun Wang. are grateful to the Transgenic Colony Management, Neurosurgery and Behavior, Lab Animal Services, Molecular Genetics, Imaging, and Histology teams at the Allen Institute for technical support. In particular, they thank V. Wright, M. McGraw, L. Potekhina, L. Kuan and A. Williford from these teams. Their work was supported by the Allen Institute for Brain Science and by NIH grants R01EY023173, U01MH105982 and U19MH114830 to H. Zeng. The authors thank the Allen Institute founder, P. G. Allen, for his vision, encouragement, and support. Y.-C.S., A.M.Z. and X.C. acknowledge members of the MAPseq core facility, including H. Zhan and Y. Li, for facilitating BARseq data production, and W. Wadolowski for technical support. Their work was supported by NIH 5R01NS073129, 5R01DA036913, RF1MH114132, and U01MH109113 to A.M.Z., R01MH113005 and R01LM012736 to J.G., and U19MH114821 to A.M.Z. and J.G.; the Brain Research Foundation (BRF-SIA-2014-03 to A.M.Z.), IARPA MICrONS (D16PC00008 to A.M.Z.), Paul Allen Distinguished Investigator Award (to A.M.Z.), Simons Foundation (350789 to X.C.), Chan Zuckerberg Initiative (2017-0530 ZADOR/ALLEN INST(SVCF) SUB to A.M.Z.), and R. Lourie (to A.M.Z.). Their work was additionally supported by the Assistant Secretary of Defense for Health Affairs endorsed by the Department of Defense through the FY18 PRMRP Discovery Award Program W81XWH1910083 to X.C. Opinions, interpretations, conclusions, and recommendations are those of the authors and are not necessarily endorsed by the U.S. Army. In conducting research using animals, the investigator adheres to the laws of the United States and regulations of the Department of Agriculture. B.Z., J.H., H.W.T. and L.I.Z. were also supported by NIH R01DC008983, RF1MH114112, U01MH116990, and EY019049. J.G. was supported by NIH R01NS096720. J.T.H., K.K., K.S.M., W.G., X.A. and Z.J.H. were supported in part by NIH 5U19MH114821-03 to Z.J.H. P.P.M. was supported by NIH EB022899, MH114824, MH114821, and NS107466, the Mather Foundation, and a Crick-Clay Professorship. X.W.Y. and H.-W.D. were supported by NIH BRAIN Initiative MH106008; X.W.Y., H.-W.D., M.Z. and N.N.F. were supported by NIH BRAIN Initiative MH117079. Y.K. was supported by NIH R01MH116176 and NIH RF1MH12460501. H.P., L.L., P.X., L.D. and Yimin Wang were supported by an Open Science initiative at Southeast University. A.L., Xiangning Li, H.G. and Q.L. were supported by NNSFC 61890953 and 61890954

**Author contributions** Co-corresponding authors: H.-W.D., J.A.H., P.O., Z.J.H. and G.A.A. conceived the project, supervised data generation, conducted data analysis, constructed figures and wrote the manuscript. Co-first authors: R.M.-C., B.Z., K.S.M., X.C. and Q.W. conducted data collection and data analysis, constructed figures and extended data figures, and participated in writing the manuscript. Other co-authors who made significant contributions to data generation, generating extended data figures and Supplementary Information, developing computational tools, as well as project management: N.N.F., A.L., A.N., K.E.H., B.H., S.B., L.K., C.S.P., Y.-G.P., M.S.B., U.C., D.W.W., X.L., Yun Wang, M.N., P.X., L.L. and K.K. Other co-authors who participated in data generations and analysis, developing computational tools, figure generation, morphological reconstructions, as well as manuscript editing (these authors are listed in an alphabetic order): X.A., S.M.A., I.B., A.B., A.C., L.D., R.D., F.D., C.E., S.F., W.G., L. Gao, J.G., P.A.G., L. Gou, J.D.H., J.T.H., H.H., J.J.H., H.K., X.K., P.L., X.L., Y.L., M.L., D.L., J.M., S.M., P.R.N., R.P., J.P., X.Q., E.S., Y.-C.S., H.W.T., W.W., Yimin Wang, S.Y., J.Y., M.Z. and L.N. Other BICCN contributing principal investigators: H. Zeng, A.M.Z., P.P.M., Q.L., H.P., X.W.Y., K.C., Y.K., J.C.G., H.G., M.H., B.K.L. and L.I.Z.

**Competing interests** A.M.Z. is a founder and equity owner of Cajal Neuroscience and a member of its scientific advisory board. J.A.H., K.E.H. and P.R.N. are currently employed by Cajal Neuroscience.

**Additional information**
**Correspondence and requests for materials** should be addressed to Giorgio A. Ascoli, Z. Josh Huang, Pavel Osten, Julie A. Harris or Hong-Wei Dong.

## a — Technologies & Methods

STPT & fMOST  CCF registration  mouse lines  AAV  rabies  BARseq

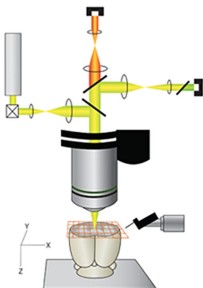
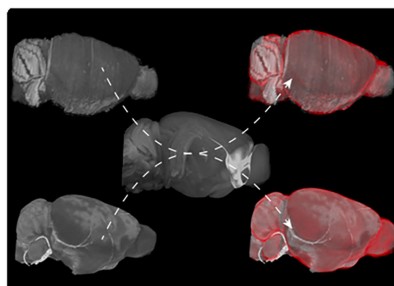
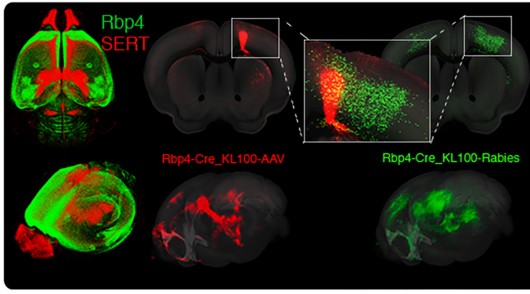
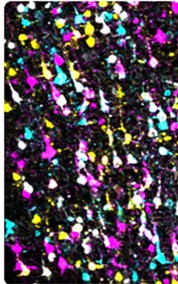

## b — Data Analysis and Visualization

populational analysis  single cell analyses  cloud visualization

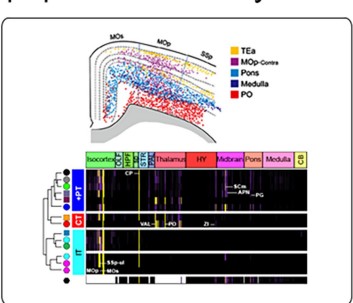
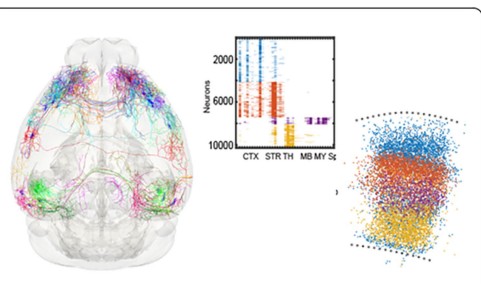
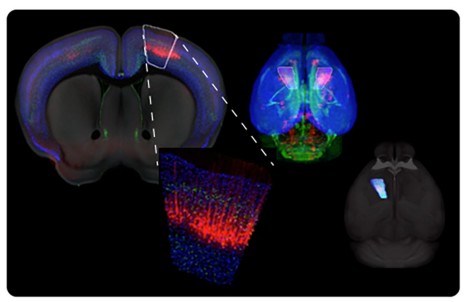

## c — Data Synthesis and BICCN Data Resources

MOP-ul areal  MOP-ul connectome  Data resources

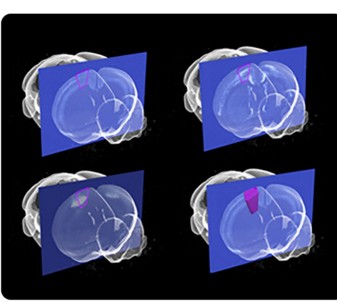
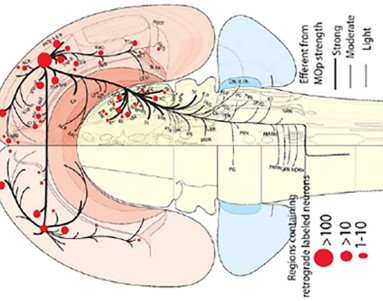
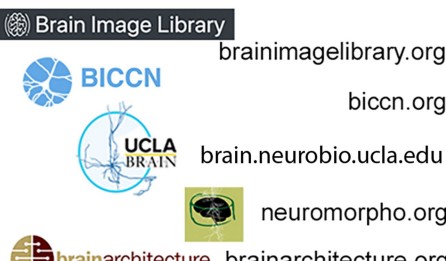

**Extended Data Fig. 1 | Overview of methods, analyses and resources used and generated by the BICCN anatomy group.** Related to Fig. 1. **a**, Whole brain image data were generated primarily by two automated microscopy methods, STPT and fMOST, and registered in the CCF. Data types include cell type distribution data used for MOp-ul regional delineations (e.g., SERT) and transgenic lines used for projection mapping (e.g., Rbp4-Cre_KL100). Outputs and inputs to MOp-ul were mapped with AAV- and rabies virus-based tracing and BARseq methods. **b**, Computational approaches used to analyze co-registered datasets include spatial and regional analyses of population labeling to quantitatively describe layer-specific anterograde and retrograde connections, and analyses of single cell morphology reconstructions and BARseq data to derive single neuron projection patterns. Neuroglancer was used for cloud-based data visualization and collaborative analyses of the CCF registered data at high resolution. **c**, The outcome of these efforts comprise a consensus-based delineation of anatomical borders of the MOp-ul, a detailed description of a cortical layer- and projection neuron type-based wiring diagram, and publicly accessible online data resources. Data and code resources are available at DOI: 10.5281/zenodo.5146390.

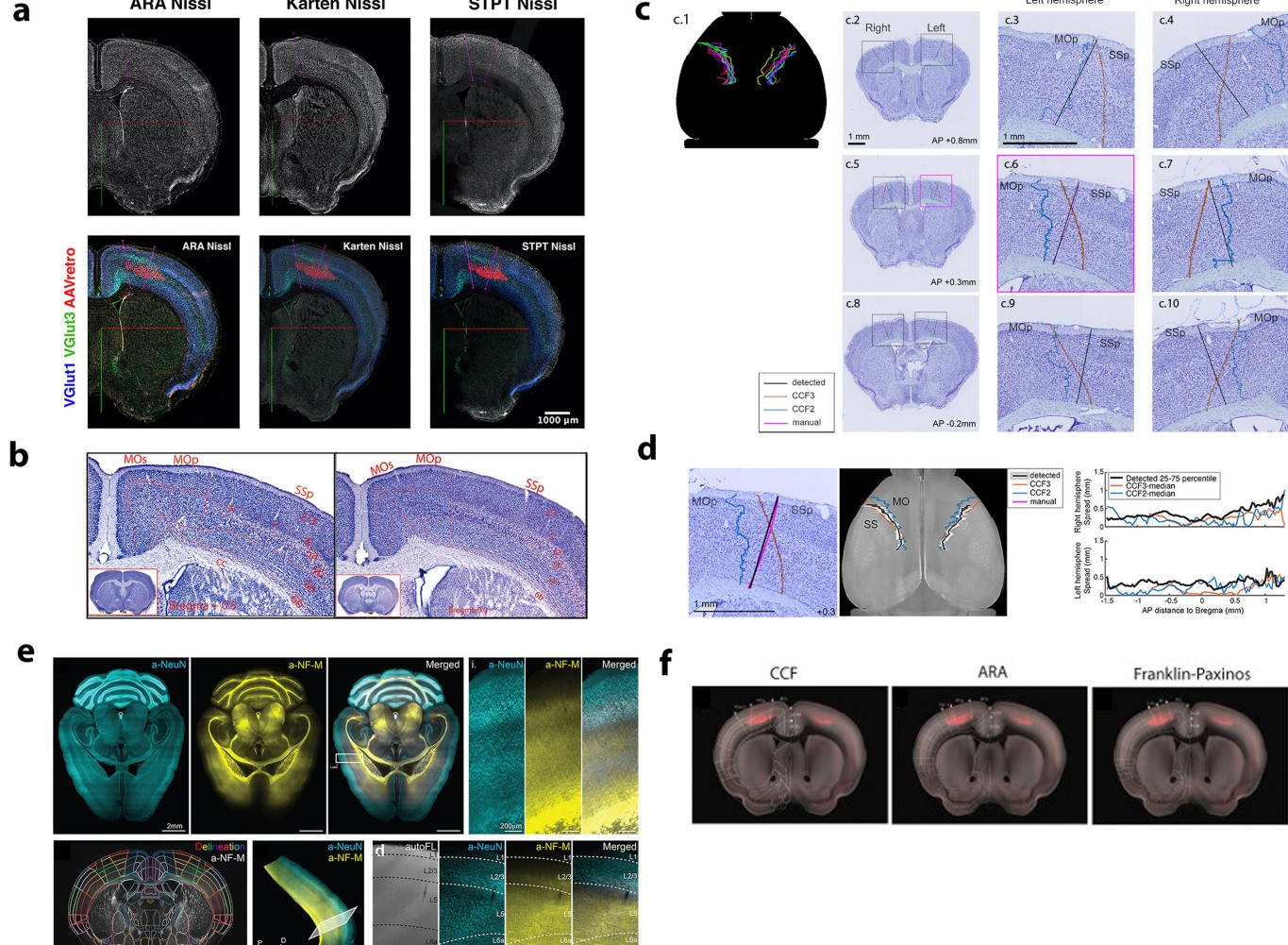

**Extended Data Fig. 2 | Multimodal MOp-ul delineation and validation.**
Related to Fig. 1. **a**, Co-registration of three sets of Nissl staining and other
modality data for the delineation of the MOp-ul. (Upper panel) Cloud-based
visualization in Neuroglancer of a coronal plane (Bregma +0.5) of three
different Nissl brains, ARA, Karten and STPT neurotrace, registered onto CCF
at cellular resolution with MOp-ul annotation. (Lower panel) MOp-ul detail of
the previous brains overlay to the same markers shown in Fig. 1: Vglut1 and
Vglut3 Cre-dependent markers, as well as AAVretro labeled cervical spinal cord
projecting neurons. **b**, Delineations of the MOp based on Nissl-stained
cytoarchitecture. The MOp and its adjacent SSp and MOs were delineated
based on their areal and laminar cytoarchitectonic properties. The SSp is
identifiable with a clearly visible "granular" L4 (gr) consisting of small densely
packed somas, which becomes thinner and less granulated (dysgranular or dg)
towards its medial tip adjacent to the MOp. Contrary to a general belief that
MOp is agranular, we observed a visible thin layer of granular cells that is
continued from SSp throughout the MOp. Finally, we identified a transitional
junction in L2/3 which was much thinner in MOp than in SSp. Digital images of
Nissl-stained histological sections were a gift from Dr. Harvey Karten (http://
brainmaps.org/index.php?action=viewslides&datid=43). Manual annotation
conducted by Dr. Hong-Wei Dong. **c**, MOp-SSp boundary algorithmically
detection based on Nissl cell textures in 10 brains mapped to the CCF with the
borders mapped from CCFv3 (red lines) and CCFv2 (blue lines). **c.1**, Nissl-based
MOp-SSp boundary (at the dorsal surface of the cortex) in 10 brains
co-registered to CCF. Each set of bilateral borders with the same color was
extracted from one brain. **c.2**, A sample Nissl stained section around AP

+0.8mm. **c.3**, A magnified view of the region in left hemisphere shown in **c.2**.
**c.4**, A magnified view of the region in right hemisphere shown in **c.2**. **c.5-c.7**,
Borders for a section around AP +0.3mm. **c.6**, In the left hemisphere of the
section shown in **c.5**, an expert-determined MOp-SSp border was denoted in
magenta. **c.8-c.10**. Borders for a section around AP −0.2mm. **d**, Left:
Algorithmically determined boundary (black), and expert manual annotation
of the MOp-SSp border (magenta) are shown together with boundaries of
reference atlases registered to an individual brain (CCFv2: blue, CCFv3: red).
Middle: Results of the algorithmic detection mapped to the CCF. The black lines
show the median of the detected MOp-SSp boundaries with 25-75 percentile
limits shown in gray. Right: The 25-75 percentile spread as a measure of
dispersion (black lines) plotted together with the distances between the
reference atlases and the median line (see panel c). **e**, (Upper panel) Lightsheet
microscopic images of 3D whole brain histology. 3D delineation of the MOp
layer borders based on whole brain immunostaining with a-NeuN and
a-Neurofilament-M (NF-M) using SHIELD-eFLASH. An optical section (*left*) and
zoom-in view (*right*). (Lower panel) An optical section of the entire brain is
shown and 3D rendering of MOp (A, anterior; P, posterior; D, dorsal; V, ventral;
M, medial; L, lateral); (Right) The cortical layer borders of MOp delineated
based on autofluorescence (black dotted lines) and immunofluorescence
(white dotted lines). **f**, Allen CCF labels, Allen Reference Atlas (ARA) labels,
Franklin-Paxinos labels established in the Allen CCF background images. Red
signals are from retroAAV-Cre injection in spinal cord registered in the CCF. See
details in Supplementary Information for integration of labels from existing
atlases onto the Allen CCF. Acronyms defined in Supplementary Table 1.

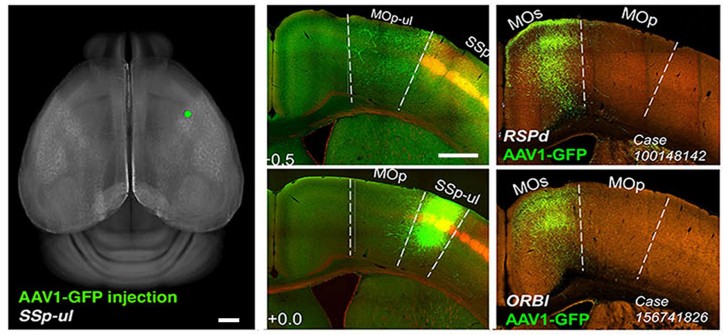

**a**

**b** Layer distribution diagram of retrogradely labeled MOp-ul projection neurons

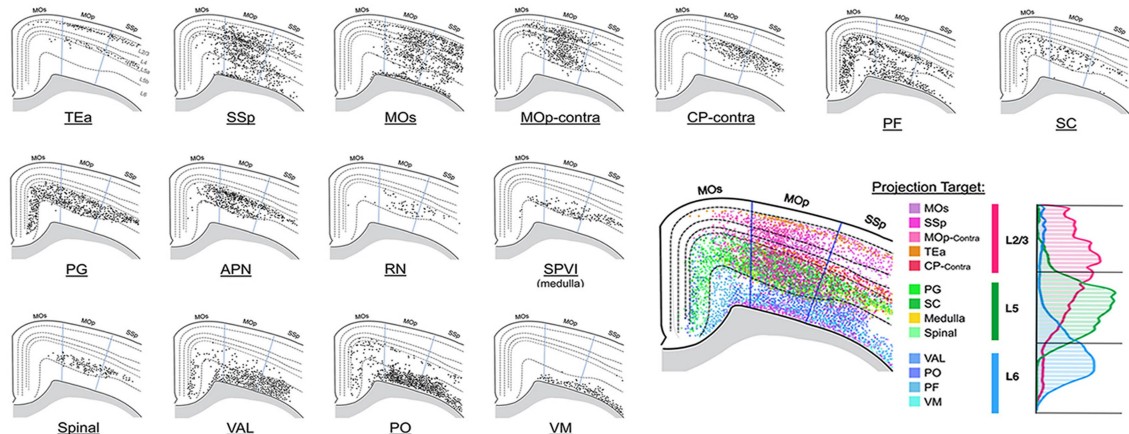

**c** Layer distribution of retrogradely-labeled cell populations in MOp-ul

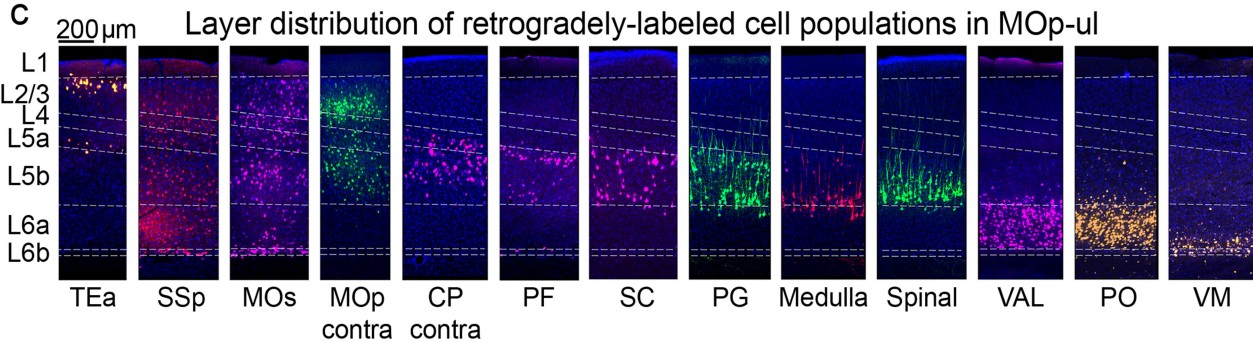

200 μm

TEa | SSp | MOs | MOp contra | CP contra | PF | SC | PG | Medulla | Spinal | VAL | PO | VM

**d** Layer distribution of Cre+ cell populations in MOp-ul

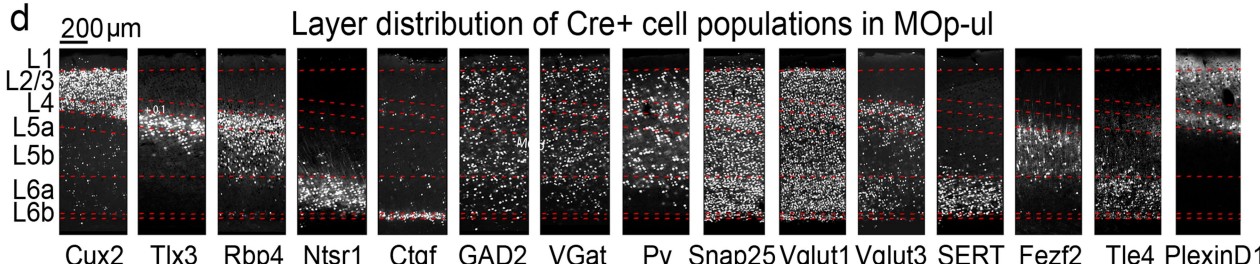

200 μm

Cux2 | Tlx3 | Rbp4 | Ntsr1 | Ctgf | GAD2 | VGat | Pv | Snap25 | Vglut1 | Vglut3 | SERT | Fezf2 | Tle4 | PlexinD1

**Extended Data Fig. 3** | See next page for caption.

**Extended Data Fig. 3 | Connectivity-based MOp parcellation and projection target defined MOp-ul neuron types.** Related to Fig. 1. **a**, Accuracy of the MOp delineation was further validated using three sets of connectivity data: (1) anterograde axonal projections from the SSp-ul to MOp-ul: transgenic mice (Scnn1a-Tg3-Cre driver line crossed with Ai14 tdTomato reporter line) received an injection of cre-dependent GFP-expressing AAV targeted precisely to the SSp upper limb area (left). Targeting restricted to the SSp was confirmed by the presence of tdTomato fluorescence in SSp layer 4. Analysis using Neuroglancer confirmed the existence of a strong monosynaptic projection from the SSp-ul to MOp-ul, therefore, confirming the border of these two adjacent cortical areas (2nd column of images); (2) the MOp medial border with the MOs was identified by the absence of a monosynaptic MOp connection with the dorsal retrosplenial area (RSPd) and ventrolateral orbital area (ORBvl) and but the presence of strong bidirectional connection between the MOs with both the ORBvl and RSPd3 (the 3rd column of images). **b**, Detailed distribution patterns of those retrogradely labeled projection neurons in the MOp. Semi-quantitative analysis shows distinct laminar specificities of different neuron types with distinct projection targets. Please also see Extended Data Fig. 7 for additional retrograde labeling in bilateral MOp. Taken altogether, IT neurons are distributed broadly across L2-6[37] Individual layers contain intermingled IT neurons innervating different targets, and neurons targeting the same structures can be distributed in different layers (also see Fig. 1e; Extended Data Fig. 7). We identified various IT types: 1) two TEa-projecting types: a L2 type that generates an asymmetric projection pattern with denser innervation to the contralateral TEa, and a L5a type with symmetrical projections to bilateral TEa (also see Extended Data Fig. 7); 2) IT neurons that target other ipsilateral somatic sensorimotor areas (e.g. MOs-ul, SSp-ul, MOs and SSs) are intermingled in layers 2, 3, 4, 5a, 5b-middle. These layers also contain dense contralateral MOp-ul PNs and relatively sparse PNs that project to contralateral MOs, SSp and SSs (also see Extended Data Fig. 7a,b); 3) L4 contains many MOs- and SSp-projecting neurons, but far fewer SSs-projecting neurons; 4) L6b IT neurons that project to ipsilateral but not contralateral MOs and SSp (Extended Data Fig. 7b). Cortico-striatal projecting IT neurons are distributed preferentially in L5a and 5b-superficial and -middle (also see Extended Data Figs. 7b). ET neurons are distributed primarily in L5b[37]. Some ET neurons display preferential sublaminar patterns, but other types occur in a smoother gradient across sublayers (also Fig. 1e). Neurons projecting to thalamus (parafascicular nucleus, PF), midbrain (anterior pretectal nucleus, APN; superior colliculus, SC), and hindbrain (pontine nuclei) were preferentially distributed in L5b-superficial and -deep; whereas neurons targeting other regions of the midbrain (red nucleus, RN), the medulla (spinal nucleus of trigeminal nerve interpolar part, SPVI), and cervical spinal cord, were preferentially distributed in L5b-middle and -deep, with the deepest L5b labeling resulting from medulla injections (also see Fig. 1e). Additionally, we identified three classes of L6 CT neurons: (1) L6a neurons that primarily project to the posterior thalamic complex (PO), ventral anterior-lateral thalamic complex (VAL), and PF, as well as the reticular thalamus (RT); (2) VM-projecting neurons in L6a deep sublayer and L6b; and (3) L6b neurons that specifically project to the contralateral thalamic nuclei, such as the PO, VAL and VM (also see Fig. 1e). **c**, Laminar distributions of retrogradely labeled MOp-ul projection neurons after tracer injections into 13 different projection targets (n=2-4 mice per target). TEa projecting neurons are distributed in L2 and L5a; SSp and MOs projecting neurons are distributed throughout L2-6b; contralateral MOp projecting neurons (commissural neurons) are in L3, L5a and L5b; contralateral CP projecting neurons are in L5a and L5b. All of these neurons belong to the intratelencephalic (IT) neuron type. Several PT (pyramidal-tract neurons) or ET (extratelencephalic) type neurons projecting to the PF, SC, pons, medulla, and spinal cord are in L5b, namely the superficial (L5b-s), middle (L5b-m) and deep (L5b-d). Finally, all corticothalamic projecting neurons (CT) are distributed in superficial L6a (VAL- and PO-projecting neurons), as well as deep L6a and L6b (VM-projecting neurons). **d**, Representative examples of laminar distribution of cell populations from a subset of cre-driver lines (Cux2, Tlx3, Rbp4, Ntsr1, Ctgf, GAD2, VGat, Pv, Snap25, Vglut1, Vglut3, SERT, Fezf2, Tle4 and PlexinD1, n=6-8 mice per line). See Extended Data Fig. 6 for the complete cre-driver lines laminar distribution.

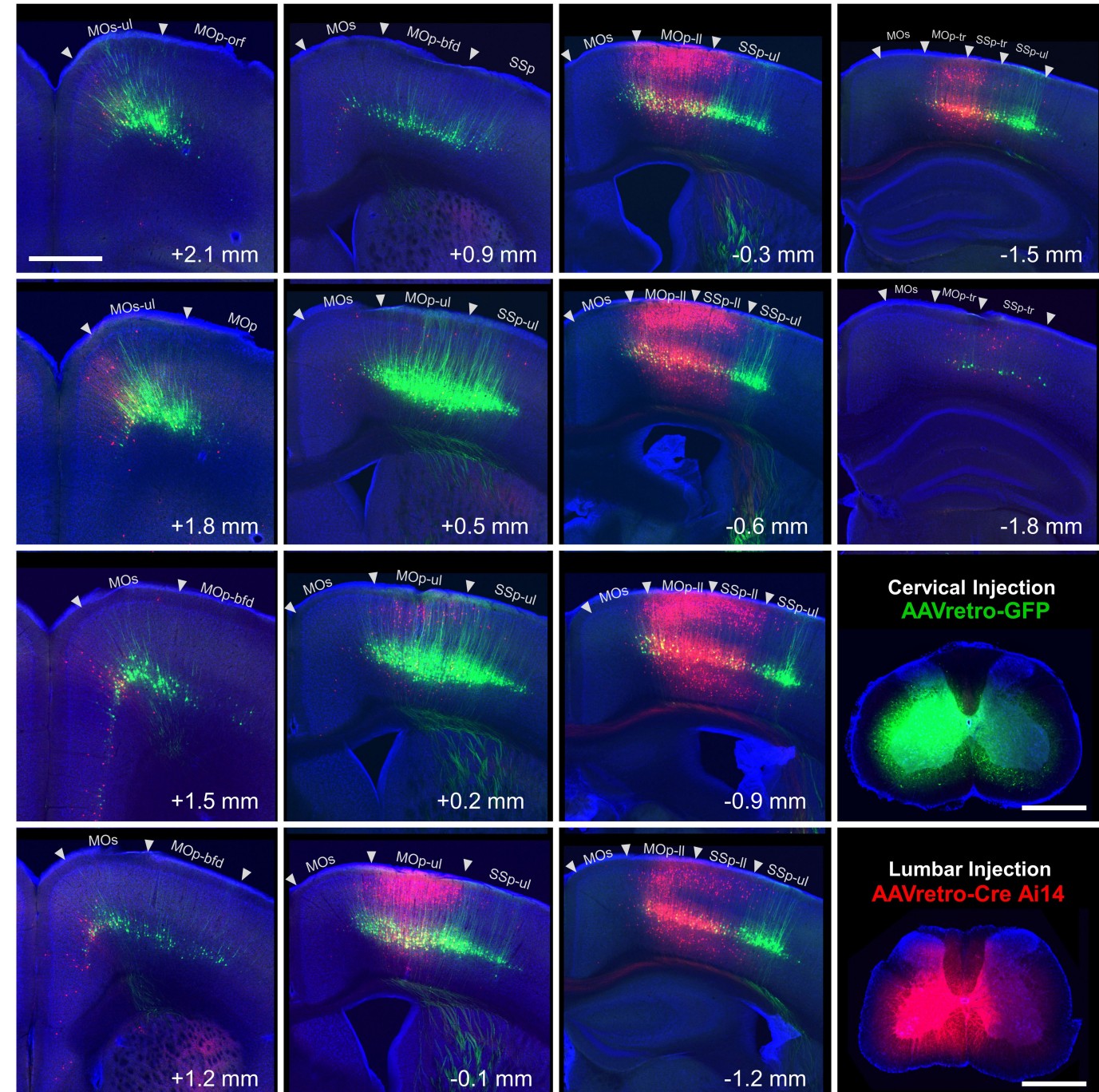

**Extended Data Fig. 4 | Distribution of cervical- and lumbar-projecting corticospinal neurons.** Related to Fig. 1. Panels show retrogradely labeled neurons in secondary motor (MOs), primary motor (MOp), and primary somatosensory (SSp) cortical regions following injections of AAVretro-GFP (green) in cervical spinal cord and AAVretro-Cre (red) in lumbar spinal cord in an Ai14-tdTomato Cre-reporter mouse. Values indicate position of coronal sections relative to bregma. Prominent projections to cervical spinal cord arise from anterior MOs (also known as the rostral forelimb area) and more caudally from MOp between +0.7 to +0.1 mm from bregma (also known as the caudal forelimb area). This latter population serves to define the rostro-caudal extent of MOp upper limb domain (MOp-ul), the focus of this study. The lateral aspect of this labeling extends into primary somatosensory area upper limb domain (SSp-ul) and continues caudally to −1.5 mm posterior to bregma. Upper limb-related somatic sensorimotor areas transition into hindlimb- (MOp-ll and SSp-ll) and trunk- (MOp-tr and SSp-tr) related areas caudal to Bregma −0.1 mm, indicated by the increased presence of lumbar-projecting neurons (red). Injection sites in the contralateral cervical and lumbar spinal cord are shown in the bottom right panels. Scale bars, 500 µm.

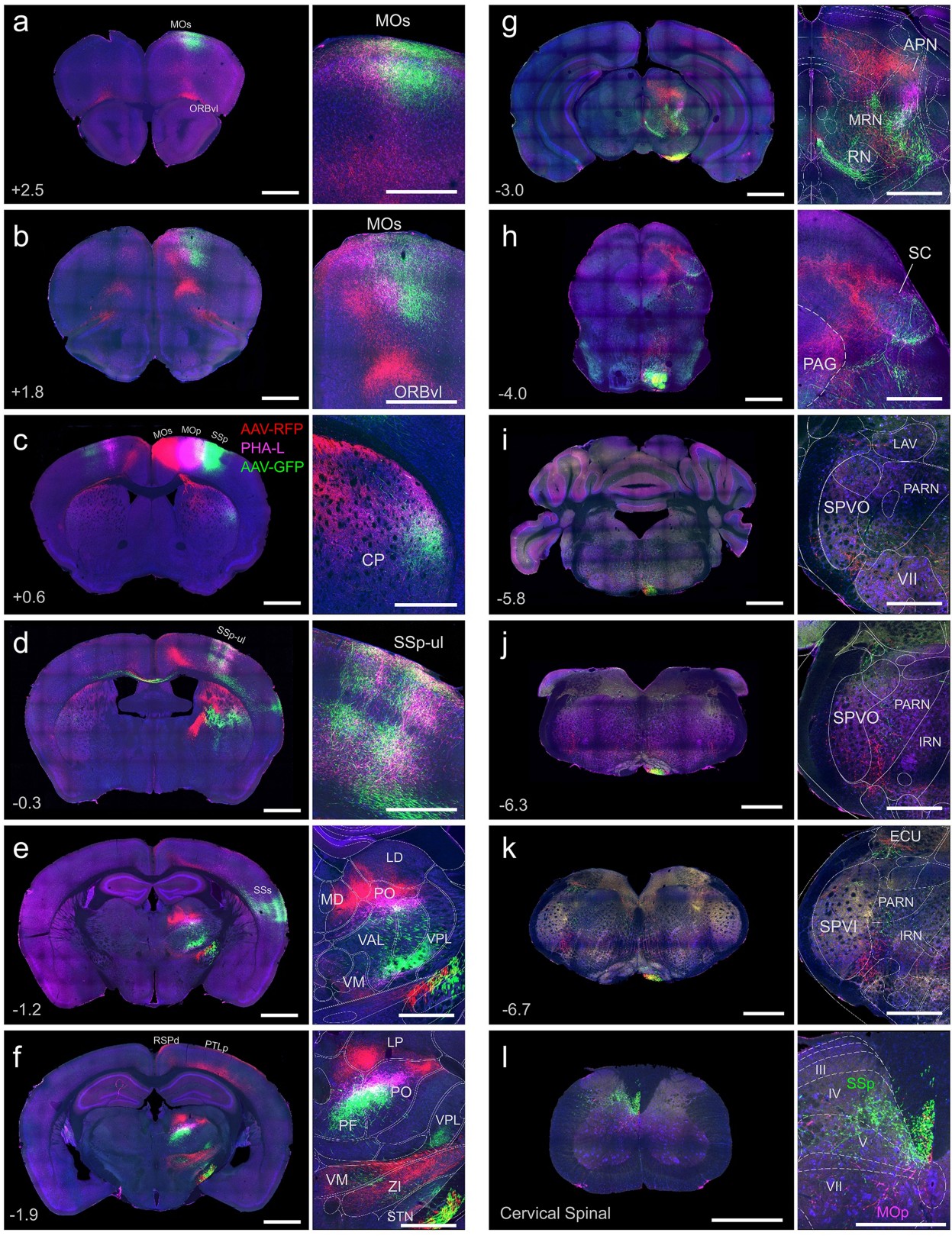

**Extended Data Fig. 5 | Distinct output from the MOs, MOp-ul, and SSp-ul.** Related to Figs. 1, 2. Panels show brain-wide axonal projections (**a**–**l**) following injections of PHA-L (pink) into MOp-ul, AAV-RFP (red) and AAV-GFP (green) into immediately adjacent MOs and SSp-ul, respectively (injection sites shown in panel **c**). MOp-ul and SSp-ul project to similar cortical regions (**a**–**e**), however they differ in their projections to the thalamus and spinal cord, with SSp-ul uniquely innervating VPL (**e**, **f**) and targeting more dorsal layers of spinal cord (**l**), compared with MOp-ul. In addition, the MOs region just medial to

MOp-ul is defined by prominent cortical projections to ORBvl (**a**, **b**), RSPd, and PTLp (**f**), and innervates MD and LP in thalamus (**e**, **f**), but lacks a projection to spinal cord (**l**). Together, these distinct projection profiles demonstrate differences in connectivity that emerge along the medial and lateral borders of MOp-ul. Acronyms are defined in Supplementary Table 1. Scale bars, 1 mm (left panels), 500 μm (right panels, **a**–**l**). Please see Supplementary Information for a detailed description of regional output from MOp-ul mapped using PHAL.

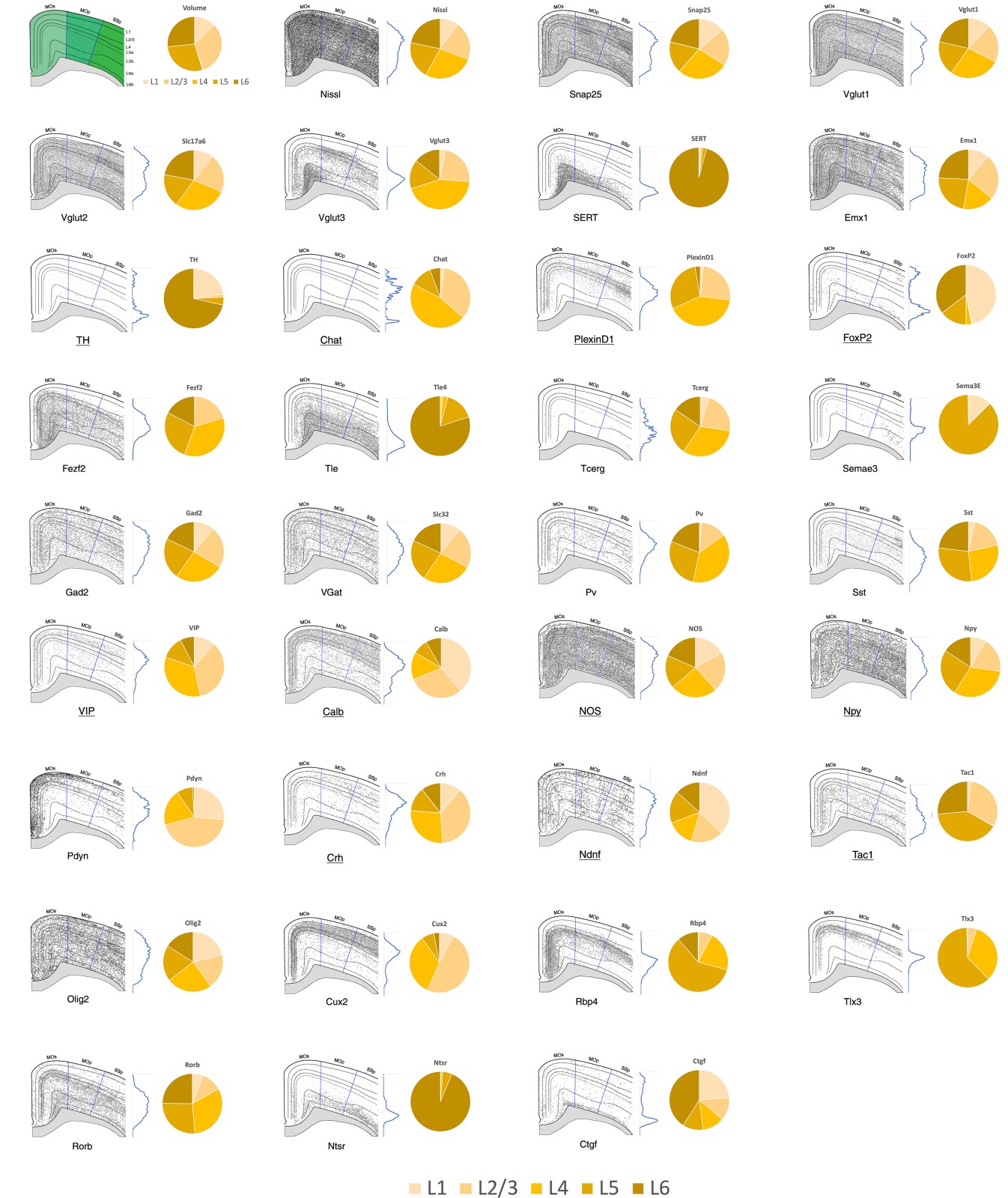

**Extended Data Fig. 6 | Quantitative analysis of Cre expression in newly defined MOp-ul using different cre-driver lines.** Related to Fig. 1. Schematic representation of MOp-ul and cell distribution of Nissl and 33 different cre-driver lines. Quantitative analysis of cortical-depth and layer-based distributions of all different cre-driver lines and Nissl. Cell distributions show specific laminar patterns for all different lines analyzed. Pies represent the percentage of cells in each layer. Interestingly, SERT+ neurons are highly expressed in layer 6 of MOp-ul in contrast to the adjacent SSp area. Similarly, Pdyn+ cells are particularly located in upper layers.

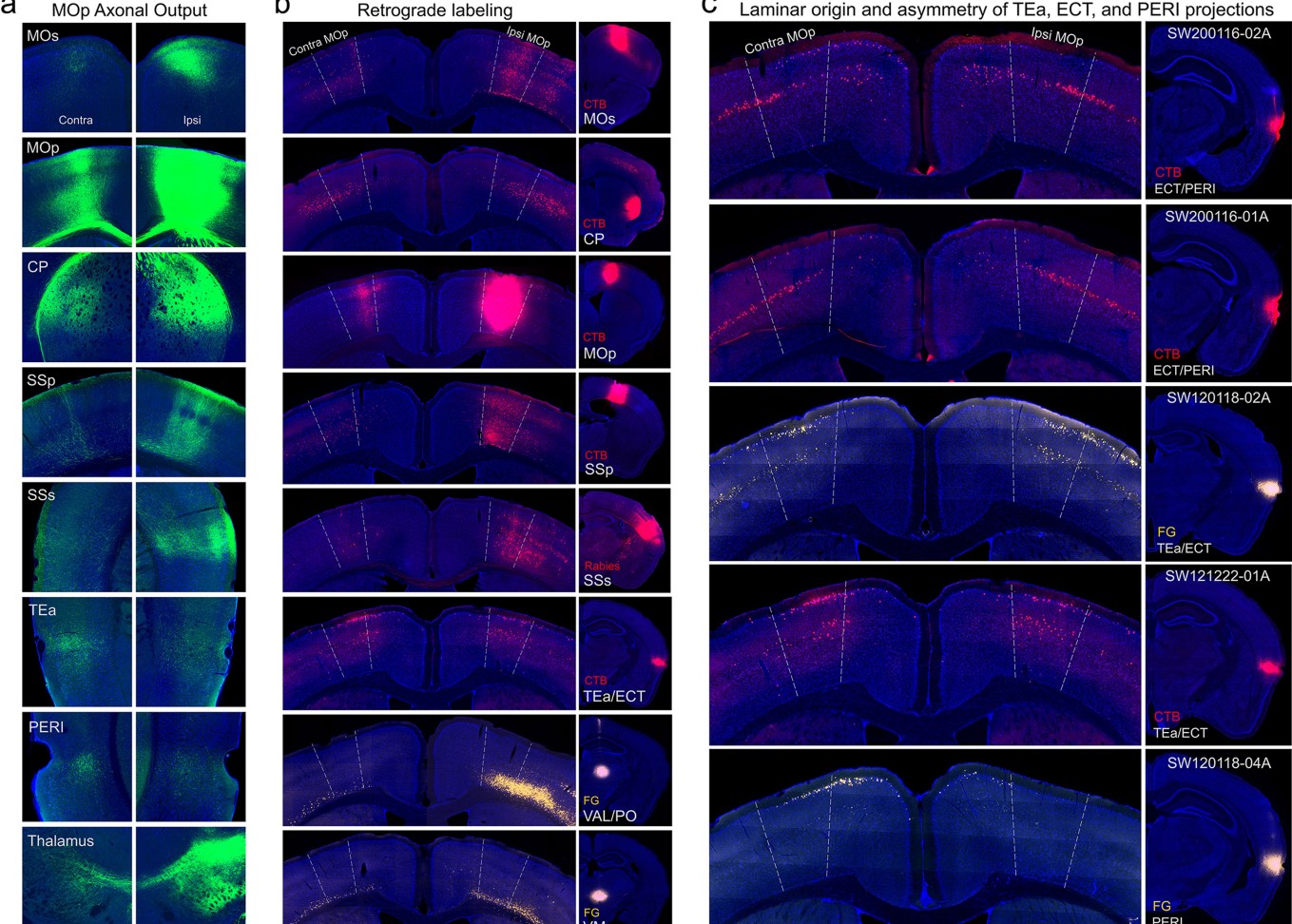

**a** MOp Axonal Output

MOs — Contra / Ipsi

MOp

CP

SSp

SSs

TEa

PERI

Thalamus

**b** Retrograde labeling

Contra MOp / Ipsi MOp

CTB MOs

CTB CP

CTB MOp

CTB SSp

Rabies SSs

CTB TEa/ECT

FG VAL/PO

FG VM

**c** Laminar origin and asymmetry of TEa, ECT, and PERI projections

Contra MOp / Ipsi MOp

SW200116-02A — CTB ECT/PERI

SW200116-01A — CTB ECT/PERI

SW120118-02A — FG TEa/ECT

SW121222-01A — CTB TEa/ECT

SW120118-04A — FG PERI

**Extended Data Fig. 7 | Laminar origin of contralateral MOp projections and distribution of retrogradely labeled TEa, ECT, and/or PERI projecting neurons.** Related to Fig. 1. **a**, Axonal output to contralateral (left column) and ipsilateral (right column) targets following PHA-L injection in MOp-ul (green). Projections are more prominent in ipsilateral targets, except for TEa, ECT, and PERI regions which receive slightly more input in the contralateral hemisphere (second and third panels from the bottom). **b**, Laminar distribution and density of cell body labeling in ipsilateral and contralateral MOp following retrograde tracer injections in each of the targets shown in (**a**). The largest number of cells observed in contralateral MOp arise from injections in striatum (CP), contralateral MOp, and TEa/ECT, mirroring the dense axonal projections to these regions seen in (**a**). Interestingly, contralateral projections to thalamus appear to arise from cells in the deepest part of L6 (bottom two panels). **c**, Panels on the left show the laminar distribution of retrogradely labeled cells in ipsilateral and contralateral MOp following retrograde tracer injection in different parts of TEa, ECT, or PERI (right panels). Cell labeling is most prominent in L5a and upper L2/3 and is seen bilaterally or with a contralateral dominance. Scale bars, 500 μm (**a**, left panels **b**, **c**), 1 mm (right panels **b**, **c**).

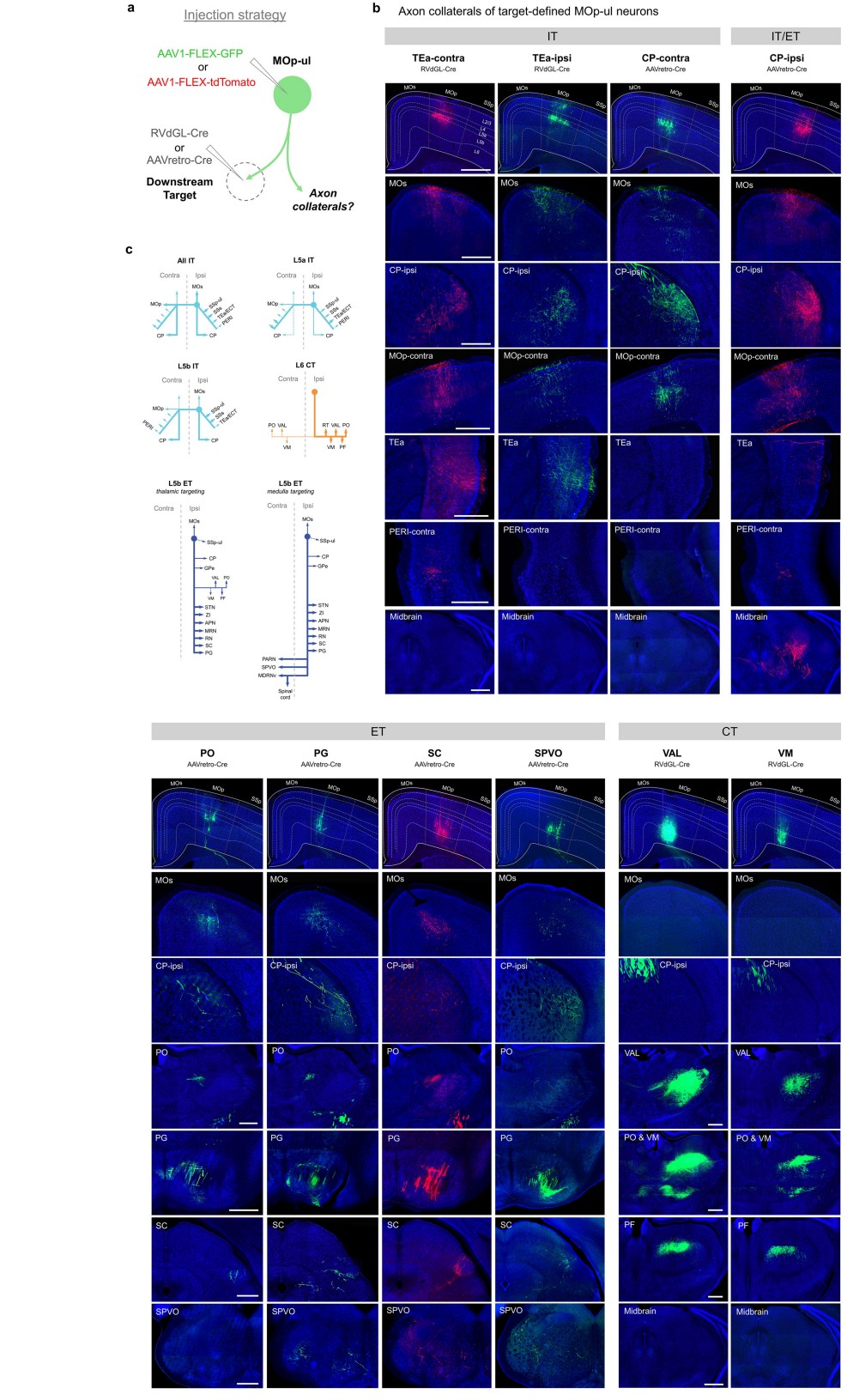

**Extended Data Fig. 8** | See next page for caption.

**Extended Data Fig. 8 | Axon collateral profiles for different target-defined MOp-ul cell populations.** Related to Fig. 1. **a**, Schematic diagram showing injection strategy. A given downstream target of MOp-ul was injected with either AAVretro-Cre or RVdGL-Cre and MOp-ul was injected with either AAV1-CAG-FLEX-GFP or AAV1-CAG-FLEX-tdTomato to Cre-dependently label the axonal output for each target-defined population. **b**, Example images of collateral outputs from different MOp-ul projection neuron types. TEa-projecting neurons (*first two columns*) were found mostly in L5a and collateralized to all cortical targets and striatum, but not to thalamus or brainstem, characteristic of the IT cell class. Interestingly, the striatal projection was predominately ipsilateral, while output to TEa/ECT was bilateral and projections to PERI exhibited a contralateral bias. In contrast, contralateral CP-projecting neurons (*third column*) also exhibited an IT projection profile, however they were found primarily in L5b (perhaps a result of AAVretro viral tropism), and displayed strong bilateral projections to striatum, but very little projection to TEa, ECT, or PERI regions. Ipsilateral CP-projecting neurons

(*fourth column*) exhibited a similar profile, but also included L5b ET neurons, which project to ipsilateral striatum, as well as thalamus and brainstem regions. Target-defined ET neurons (*columns 5-8*) broadly collateralized to all other expected targets of this class, except for thalamic-targeting ET cells (*column 5*, AAVretro-Cre injection in PO labels L5b, but not L6, thalamic projection due to viral tropism) which displayed little or no projection to lateral medulla (e.g. SPVO, last panel in column), while lateral medulla-targeting ET cells (*column 8*, SPVO) showed little or no projection to thalamus (e.g. PO, fourth panel in column). In addition, all target-defined ET cell populations collateralized to ipsilateral MOs (second row). Lastly, L6 VAL or VM-projecting neurons (*columns 9 and 10*) co-targeted all other expected thalamic nuclei (PCN, PO, PF) and the reticular thalamic nucleus (RT). No cortical, striatal, or brainstem collaterals were observed, characteristic of the CT cell class. **c**, Summary of collateral targeting differences for each major cell class. Scale bars, 500 μm (**b**).

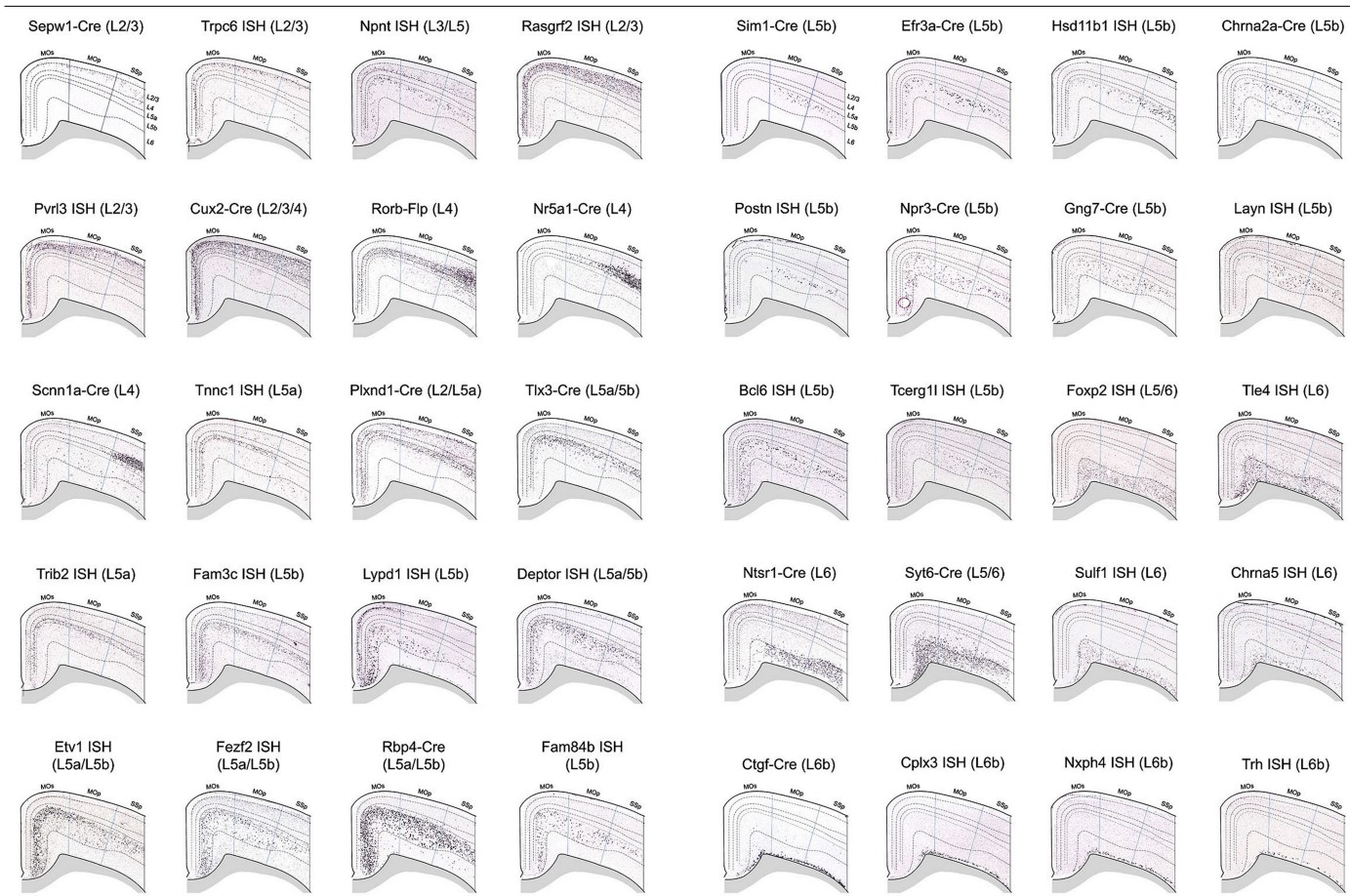

**Extended Data Fig. 9 | Laminar-specific expression of select genes and transgenic mice.** Related to Fig. 1. Panels show *in situ* hybridization (ISH) data for endogenous gene expression in MOp taken from the Allen Gene Expression Atlas (http://mouse.brain-map.org/) or for Cre- or Flp-expression in adult transgenic mice (http://connectivity.brain-map.org/transgenic). Data were manually aligned to a representative coronal atlas section through MOp-ul (+0.5mm bregma) and cortical layers are indicated with dashed lines.

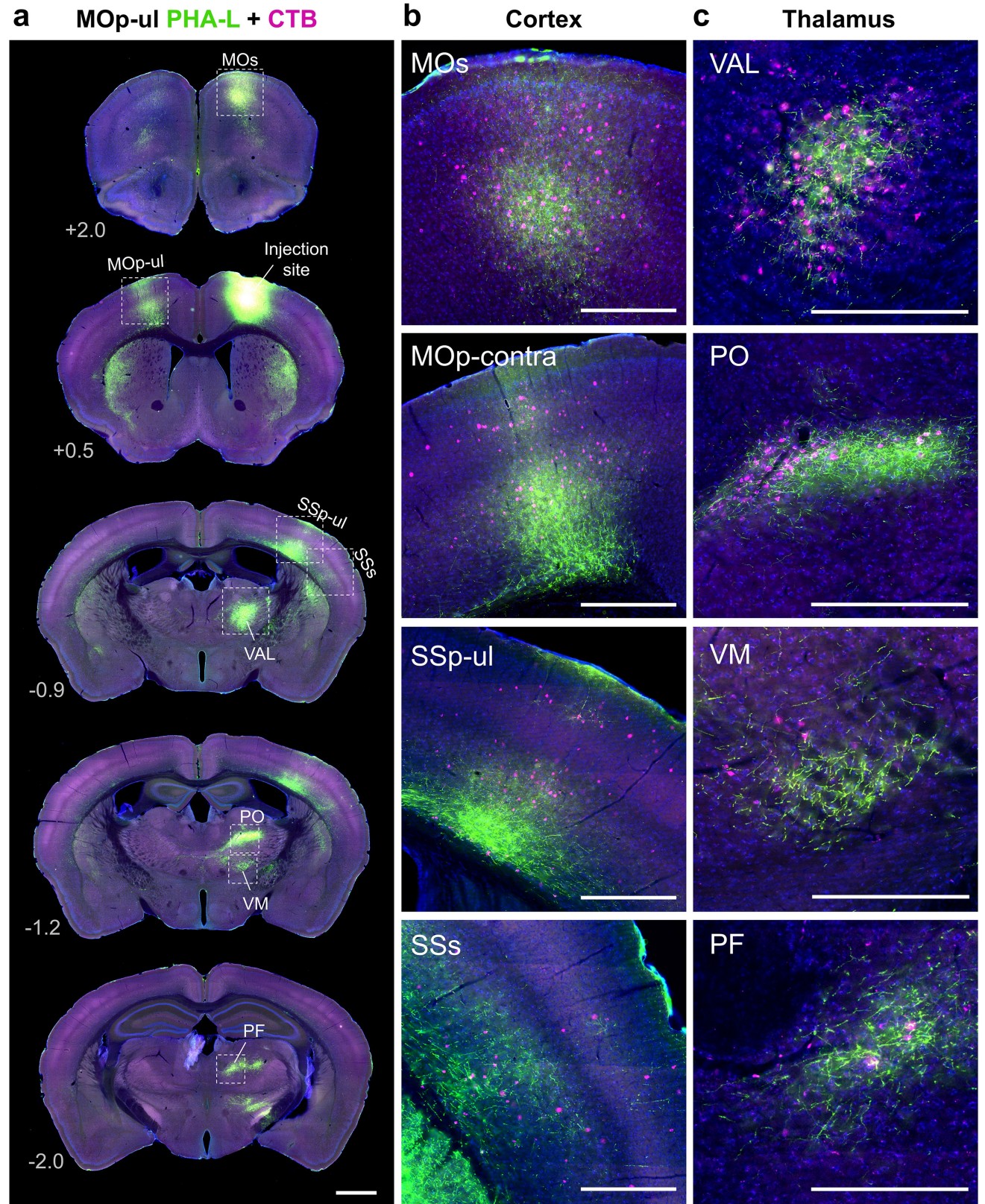

**a** MOp-ul PHA-L + CTB    **b** Cortex    **c** Thalamus

**Extended Data Fig. 10 | Major regions that share reciprocal connections with MOp-ul.** Related to Figs. 1, 2, 3. **a**, Coronal series of sections highlighting brain regions with both retrograde (CTB, pink) and anterograde (PHA-L, green) labeling following co-injection of both tracers into MOp-ul. Values are in mm relative to bregma. **b**, **c**, Higher magnification views (10X) of cortical and thalamic regions shown in (**a**). This data shows that the MOp-ul shares strong reciprocal connections with (1) other somatic sensorimotor areas (panel b), namely the MOs-ul, contralateal MOp-ul, SSp-ul, SSs and TEa (not shown here); and (2) several thalamic nuclei (**c**), such as VAL, PO, VM and PF. Scale bars, 1 mm (**a**), 500 μm (**b**, **c**).

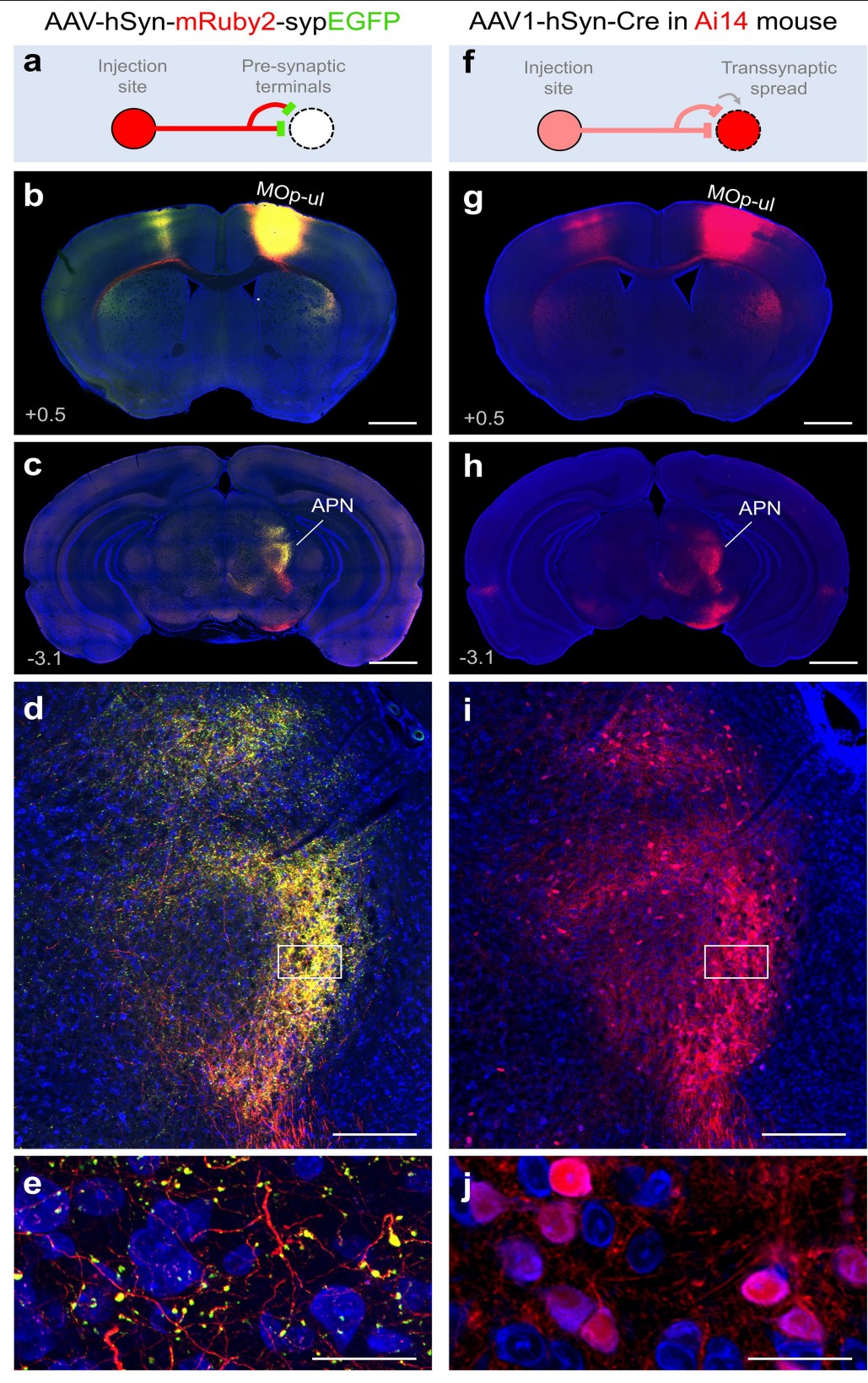

**AAV-hSyn-mRuby2-sypEGFP**

**AAV1-hSyn-Cre in Ai14 mouse**

**Extended Data Fig. 11 | Approaches for further establishing synaptic connectivity in downstream targets of MOp.** Related to Fig. 2. **a**, To distinguish fluorescent labeling of synaptic boutons versus axons of passage in a given MOp target, AAV-hSyn-mRuby2-sypEGFP was injected into MOp-ul and the anterior pretectal nucleus (APN) was examined for the presence of synaptophysin-tagged EGFP+ boutons (green) and mRuby2+ axons (red). **b**, Injection site in MOp-ul. **c**, Labeling of boutons and axons in APN at 4X, 10X (**d**), and 40X magnification (**e**), confirming synaptic innervation of the

downstream structure. **f**, Similarly, AAV1-hSyn-Cre may be used to confirm and quantify synaptic connectivity in a given target region following anterograde transsynaptic spread of the virus to downstream neurons and subsequent expression of tdTomato in Ai14 Cre-reporter mice. **g**, Injection site in MOp-ul. **h**, Post-synaptically labeled tdTomato+ cells (red) in APN at 4X, 10X (**i**), and 40X magnification (**j**) confirming a similar pattern of innervation as shown in (**d**). Blue, fluorescent Nissl stain. Values in mm relative to bregma. Scale bars, 1 mm (**b**, **c**, **g**, **h**), 200 μm (**d**, **i**), 50 μm (**e**, **j**).

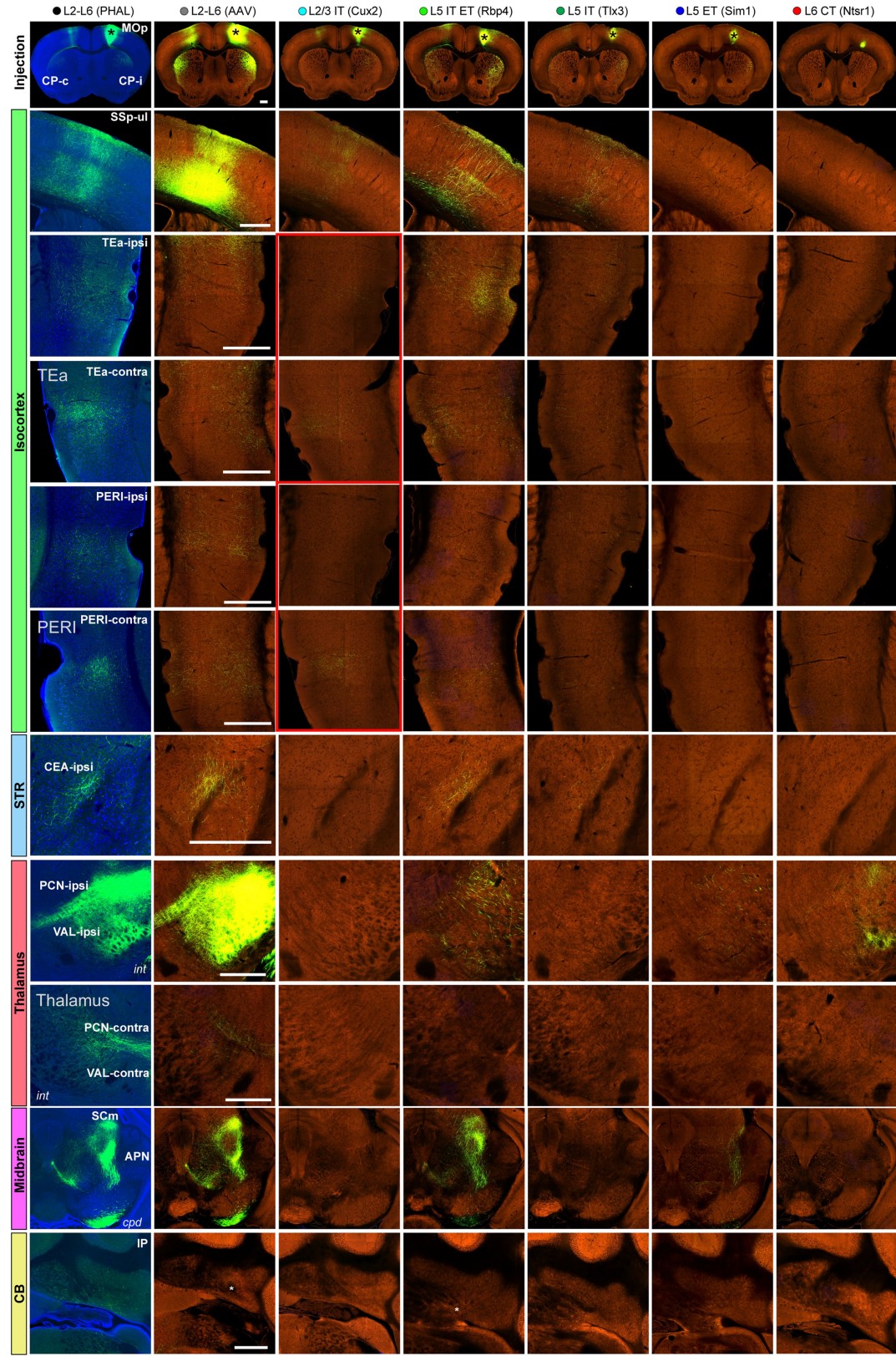

**Extended Data Fig. 12** | See next page for caption.

**Extended Data Fig. 12 | MOp-ul projection patterns to select targets by layer and class.** Related to Fig. 2. Top row, coronal plane images show the approximate center of each tracer injection site into the MOp-ul area (indicated with *) for wild type mice or Cre lines indicated for each column. Labeled axonal projections are also visible in these sections in the ipsilateral and contralateral CP (CPi, CPc). *All other rows*, each row shows coronal images at the level of 10 distinct isocortical and subcortical targets of MOp-ul. Anterograde tracing results are strikingly similar across the conventional tracer, PHAL, and AAV-EGFP injected into MOp of wild type mice (first two columns). For some targets (rows) the layer/class origin of the labeled axons in each target is clear. For example, only Cre lines with IT cells project strongly to SSp-ul (Cux2, Rbp4, Tlx3); few to no axons are present in the L5 PT and L6 CT lines. Of note, the hemisphere asymmetry in projections to TEa and PERI is attributable to L2/3 cells, (red boxes in Cux2 column). MOp-ul projections to CEA originate from L5 IT cells (axons labeled in Rbp4 and Tlx3 lines). Ipsilateral VAL projections are strongest in the L6 CT line (Ntsr1), and none of these Cre lines labeled contralateral thalamic projections, although these axons were labeled in both PHAL and wild type AAV tracing experiments. All midbrain projections arise from L5 PT cells (Rbp4, Sim1), as well as the sparse projection to the deep cerebellar nuclei (IP, in Rbp4 panel only). Number of experiments per line and tracer are listed in Fig. 2a (n=1-2; not all experiments were independently repeated as we previously demonstrated n=1 is a good predictor of connectivity strengths across multiple animals). Scale bars = 1 mm in top row panels, 500 μm in all other rows.

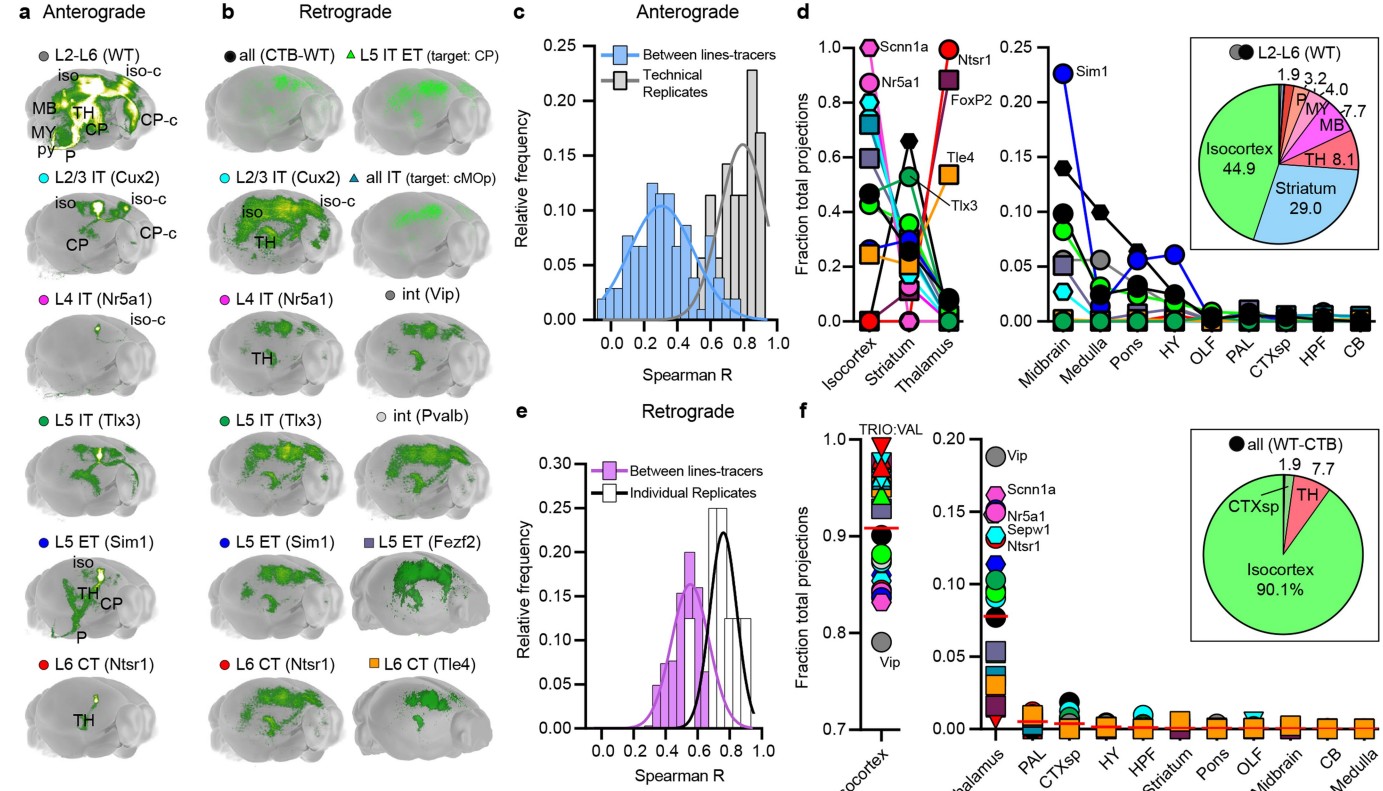

**Extended Data Fig. 13 | Output and input tracing to cell classes in MOp-ul.**
Related to Figs. 2, 3. **a**, **b**, Maximum intensity projections show the brain-wide distribution of anterogradely labeled axons or retrogradely labeled input cells traced from MOp-ul and from distinct layer and class defined by Cre lines. Note the strong similarities in patterns for all retrograde tracing experiments.
**c**, Frequency distributions of Spearman's correlation coefficients (R) from the dataset in Fig. 2c and for Rs measured between individual experimental replicates in MOp. A curve was fit to each distribution (lines). The distribution of Spearman Rs between different line-tracer experiments is normally distributed with weaker correlations than for the replicates (mean = 0.30 v 0.79). **d**, The fraction of total projections is plotted for each line/tracer across 12 major brain divisions. The pie chart inset shows the % of total axons in the PHAL and AAV experiments in WT mice. Most projections from MOp-ul target regions within isocortex, striatum, thalamus and midbrain, with relatively fewer projections to the medulla and pons. The fraction of total projections in each major division reflect the projection class labeled by different Cre lines. For example, L4 IT lines (Scnn1a, Nr5a1) had more axon in isocortex compared

to L2/3 and L5 IT lines (1.0 and 0.87 vs. 0.8, 0.74, 0.72, 0.47; Sepw1–L2/3, Cux2–L2/3, Plxnd1–L2/3+L5, and Tlx3-L5, respectively). In contrast, Tlx3–L5 had larger projections into striatum compared to other IT lines (0.53 vs. 0.17, 0.25, and 0.26; Sepw1–L2/3, Cux2–L2/3, and Plxnd1–L2/3+L5, respectively).
**e**, Frequency distributions of Spearman's correlation coefficients (R) from the dataset in Fig. 3c and for R measured between individual experimental replicates in MOp. A curve was fit to each distribution (lines). Correlations between input patterns for all pairs of lines and tracers (columns in Fig. 3c) were significantly lower than for technical replicates (p=0.02, Kruskal-Wallis test with Dunn's multiple comparison post hoc). The mean of the distribution of Spearman Rs between retrograde tracing experiments is notably closer to the replicate mean (0.55 v 0.76) compared to the anterograde tracing experiments.
**f**, The fraction of total inputs is plotted for each line/tracer across 12 major brain divisions. The pie chart inset shows the % of total inputs from the CTB experiment in WT mice to summarize the total brain-wide distribution across all layers/classes. Most input to MOp-ul is from regions within isocortex, followed by thalamus across all lines/tracers.

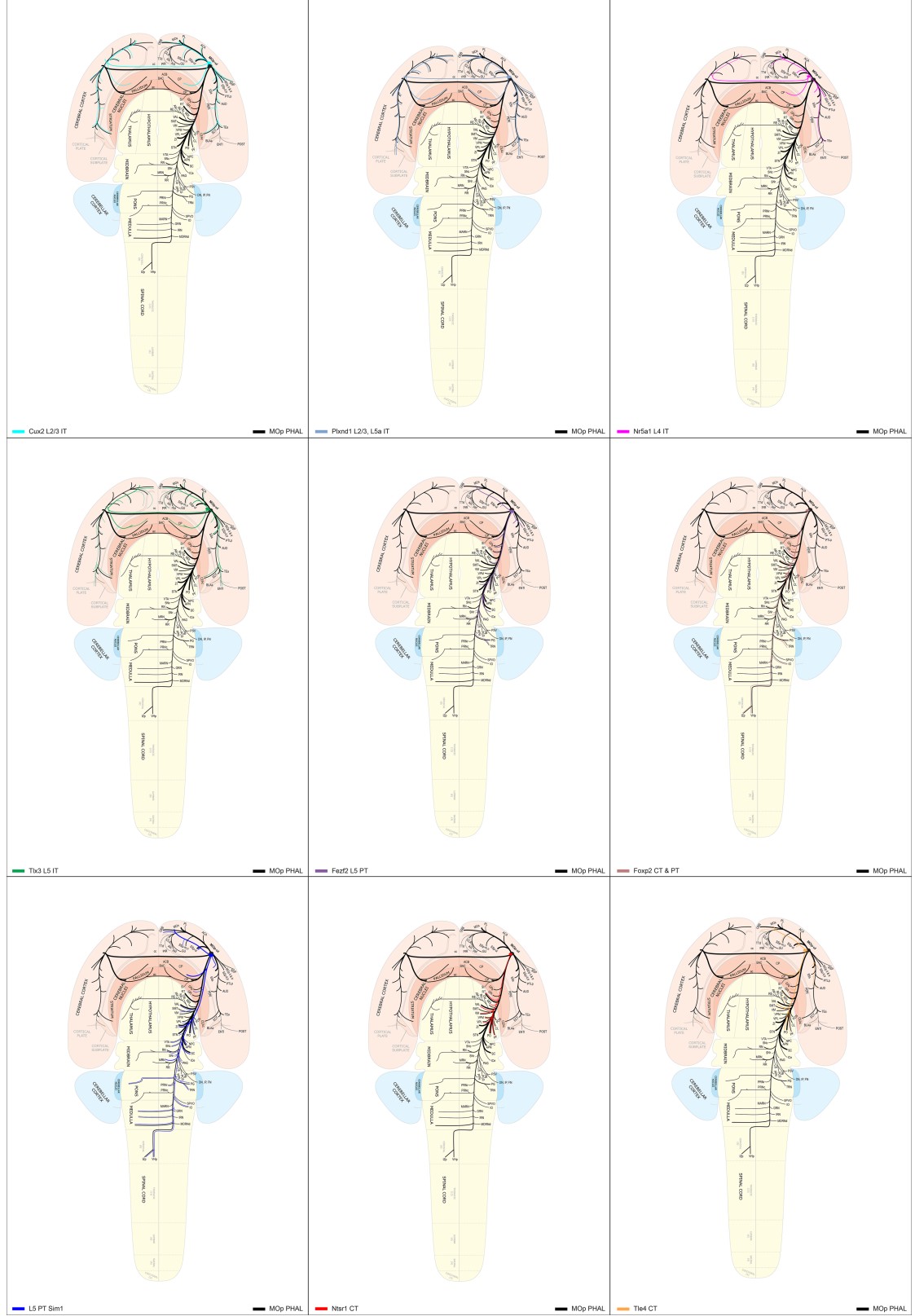

**Extended Data Fig. 14 |** See next page for caption.

**Extended Data Fig. 14 | Schematic summaries of MOp-ul outputs by area (PHAL) and for different cell types revealed with cre-dependent AAV tracing in different cre mouse lines on a whole brain flatmap of the rodent brain5 (also see Swanson, Brainmap 4.0 in http://larrywswanson.com/?page_id=1415).** Related to Fig. 2. These data shows that each of the MOp cell types (L2/3 IT, L4 IT, L5 IT, L5 ET/PT, L6 CT) display a discrete subset of MOp projections. Please note that these results also showed MOp-ul axons targeting several previously unreported areas, e.g., the capsular central amygdalar nucleus (CEAc), bed nucleus of the anterior commissure (BAC), globus pallidus external segment (GPe), contralateral thalamic nuclei (PCN), and cerebellar interposed nucleus (IP; Suppl. Information). Further analyses of the connectivity matrix (Source Data Fig. 2, formatted matrix tab) and images (Extended Data Fig. 12) reveal the predominant PN types constituting new and established MOp-ul output channels (also see Fig. 2d). For example, projections to SSp-ul originate from both L2/3 and L5 IT neurons labeled in the Cux2–L2/3, Rbp4–L5 and Tlx3-L5 populations. Projections to CEAc arise from L5 IT cells (Rbp4-L5 and Tlx3-L5, also see Extended Data Fig. 12), and projections to GPe are primarily from ET cells (Rbp4-L5, Fezf2-L5, Foxp2-L6, Sim1-L5). Anterograde experiments also confirmed (in Cux2-L2/3) the population of L2 neurons projecting contralaterally to TEa, ECT, and PERI identified by retrograde tracing (see Extended Data Fig. 7), and a Tle4-L6 CT projection to contralateral thalamic nuclei. Moreover, the sparse cerebellar projection to the IP nucleus we observed with PHAL, AAV-GFP, and AAV1-Cre anterograde monosynaptic tracing is also labeled in Rbp4-L5, but not other ET lines (Source Data Fig. 2, Extended Data Fig. 12).

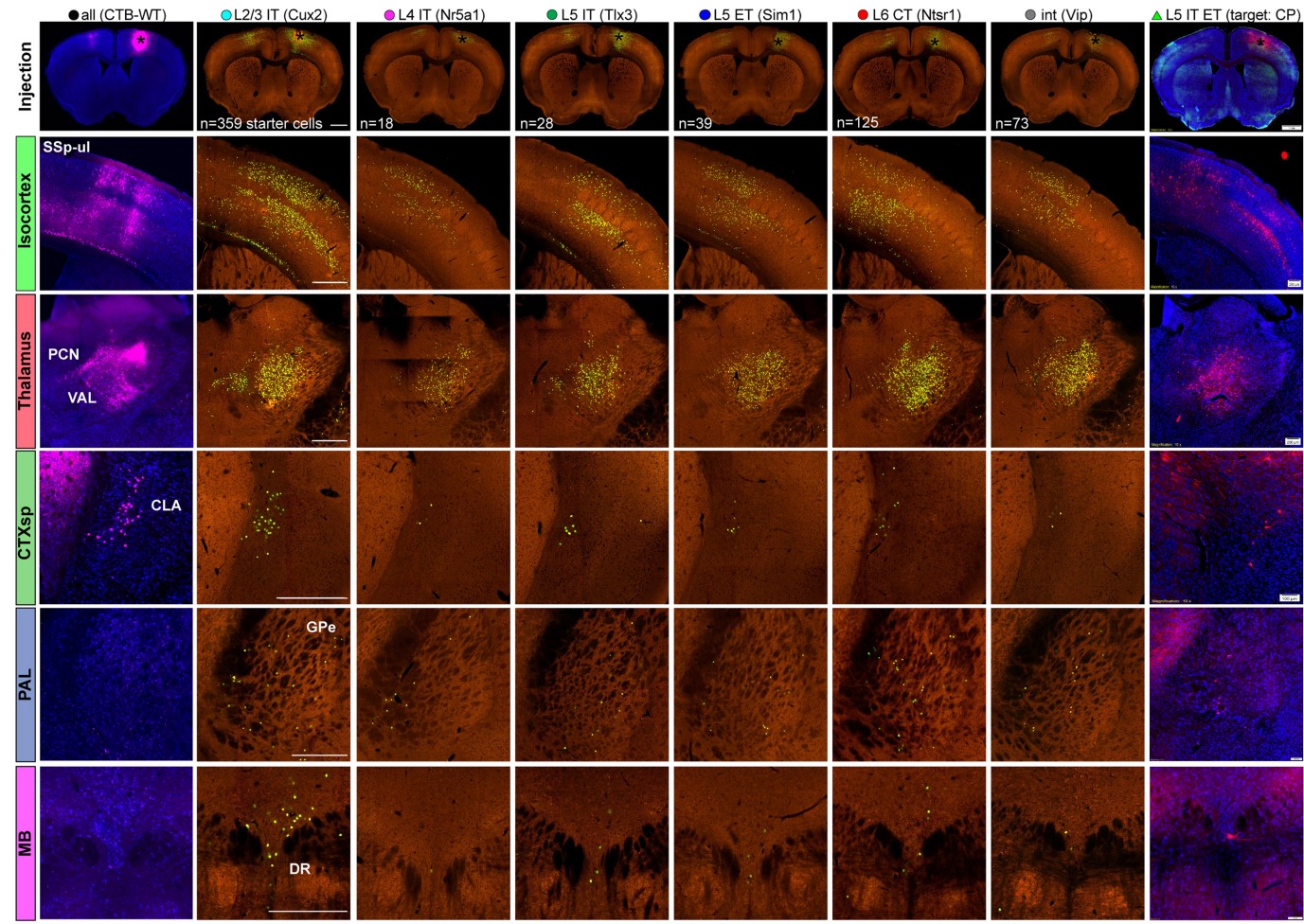

**Extended Data Fig. 15 | Brain-wide input patterns from select sources to MOp-ul by layer and class.** Related to Fig. 3. Top row, coronal plane images show the approximate center of each tracer injection site into the MOp-ul area (indicated with *) for wild type mice or Cre lines indicated for each column. The number of starter cells varies by Cre line for the rabies experiments and is shown in the injection panels. *All other rows*, each row shows coronal images at the level of three distinct source locations with cells that send input to MOp-ul. Number of experiments per line and tracer are listed in Fig. 3a (n=1-2). Scale bars = 1 mm in top row panels, 500 μm in all other rows.

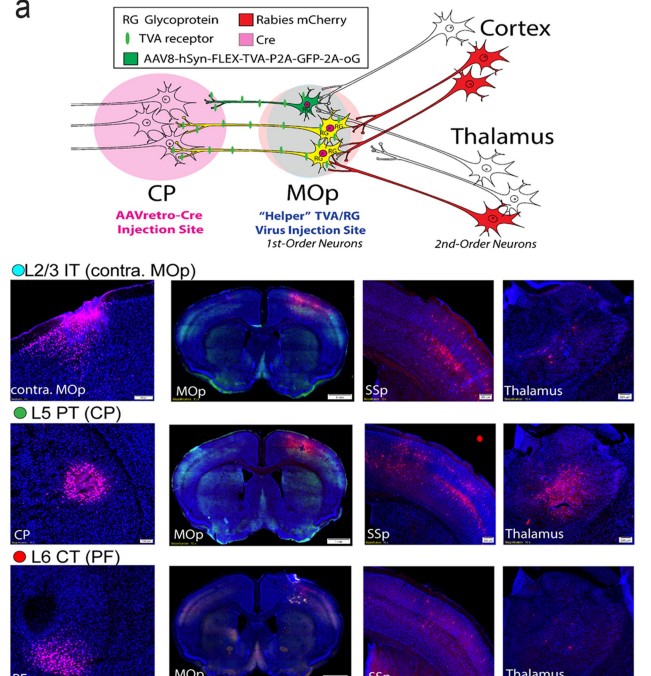

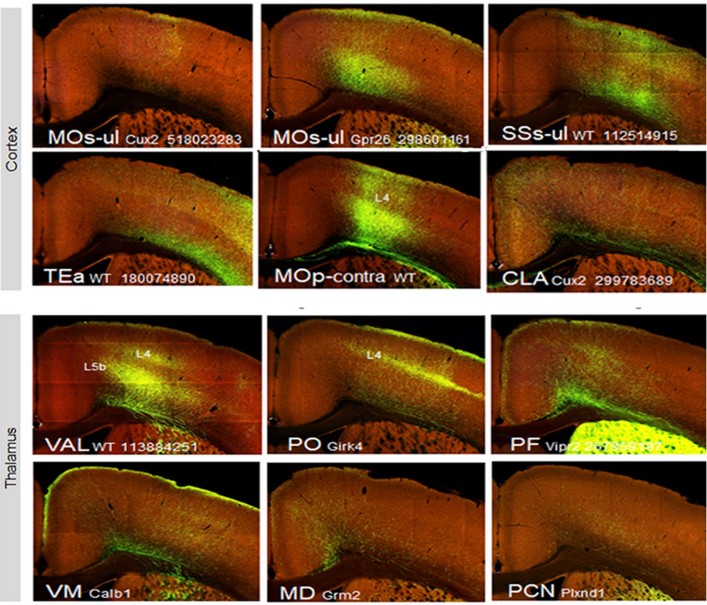

**Extended Data Fig. 16 | Neural inputs to the MOp.** Related to Fig. 3. **a**, TRIO experiments reveal monosynaptic input to projection-defined MOp cell types. (Upper panel) Schematic diagram of TRIO approach. AAVretro-Cre is injected into a downstream target of a MOp projection neuron population (ex. CP) and Cre-dependent, TVA- and RG-expressing helper virus (AAV8-hSyn-FLEX-TVA-P2A-GFP-2A-oG) and mCherry-expressing G-deleted rabies virus are injected into the MOp to label the MOp projection neurons population (1st-order) and their brain-wide monosynaptic inputs (2nd-order). (Lower panel) Example images of three separate TRIO experiments identifying monosynaptic inputs to IT, PT, and CT cell classes within the MOp showing Cre injection sties (left), helper virus and rabies injection sites in MOp (middle), and monosynaptically labeled inputs in the SSp and thalamus (right). **b**, Axonal projections to the MOp-ul arising from different cortical areas and thalamic nuclei display diverse laminar specificities in the MOp-ul. For example, MOs axons are preferentially distributed in L1, L5 & L6; densest TEa axons are primarily distributed in L6b; while axons from SSs and contralateral MOp are distributed diffusely across all layers of MOp-ul. Thalamocortical projections to the MOp-ul more or less follow a rough core/matrix organization described previously for thalamic inputs to primary sensory cortices[37]. In particular, VAL axons generate dense terminals specifically in MOp-ul L4, L5b & L6—a typical "core" type thalamocortical inputs. But, axonal inputs from the PO and PF in the MOp-ul are densely distributed in both L1 (a typical "matrix" type inputs) and L4, thus, a mixture core and matrix pattern. PF axons are further distributed in L6. Inputs from other thalamic nuclei, such as VM, MD, and PCN are diffusely distributed across multiple layers. Based on these results, it is reasonable to anticipate that different PN neuron types (IT, PT, and CT) with their soma and dendritic arbor distributions in different layers may preferentially receive discrete cortical and thalamic inputs at single neuron resolution.

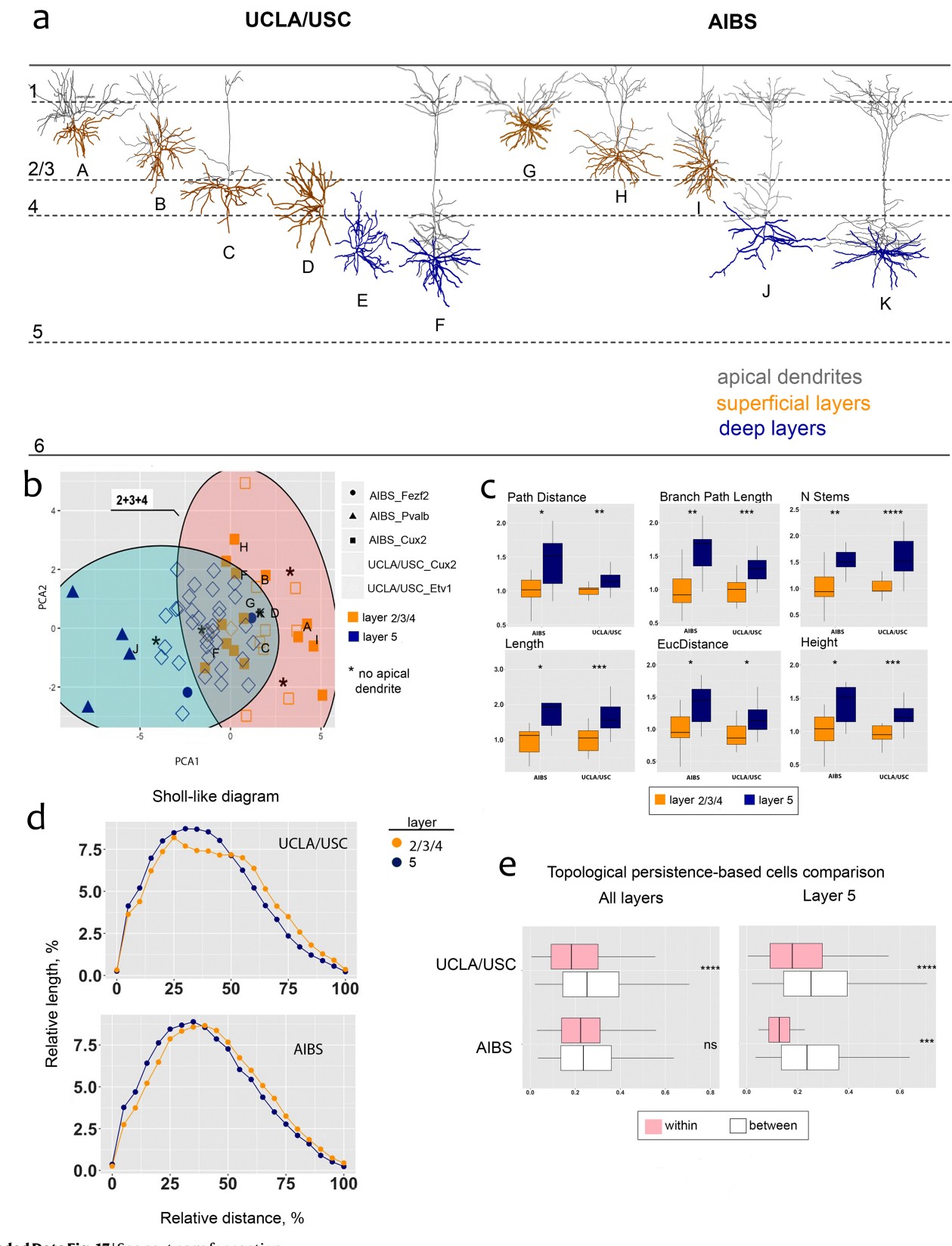

**Extended Data Fig. 17 |** See next page for caption.

**Extended Data Fig. 17 | related to Fig. 3. Local morphometric features of MOp neurons across layers. a**, Examples of reconstructed cells within MOp cortical layers 2/3/4 (orange) and 5 (blue) (see Methods). Note some L4-5 Cux2/Etv1 neurons lack an apical branch. The total neurons reconstructed for each mouse strain are: MORF3 (@UCLA/USC) x Cux2-CreERT2 (n=9) or Etv1-CreERT2 (n=36); TIGRE-MORF (@AIBS) x Cux2-CreERT2 (n=16), Fezf2-CreERT2 (n=3), or Pvalb-Cre (n=4). **b**, Principal component analysis (PCA) shows segregation of MOp layer-specific neurons based on measured morphological features. **c**, Wilcoxon Signed-Rank tests were run (all parameters survived the false discovery rate correction) and group differences between layers 2/3/4 and 5 basal dendritic trees of UCLA/USC (L2-4 [n=11], L5 [n=34]) and AIBS (L2-4 [n=16], L5 [n=7]) cases separately are presented in whisker plots and the degree of their significance is indicated by stars. **d**, Sholl-like analysis comparing basal dendritic patterns in MOp layer 5 and layers 2/3/4 neurons. The distribution of normalized dendritic length is plotted against the relative path distance from the soma. The graph shows that dendrites of neurons within layer 5 of MOp have slightly larger dendritic length closer to the cell body compared to the dendrites of layers 2/3/4 neurons. **e**, Persistence-based neuronal feature vectorization was also applied to summarize pairwise differences between superficial and deep neurons for UCLA/USC and AIBS datasets independently. For whisker plots in c and e, the center line represents the median, box limits show the upper and lower quartiles and the whiskers represent the minimum and maximum values.

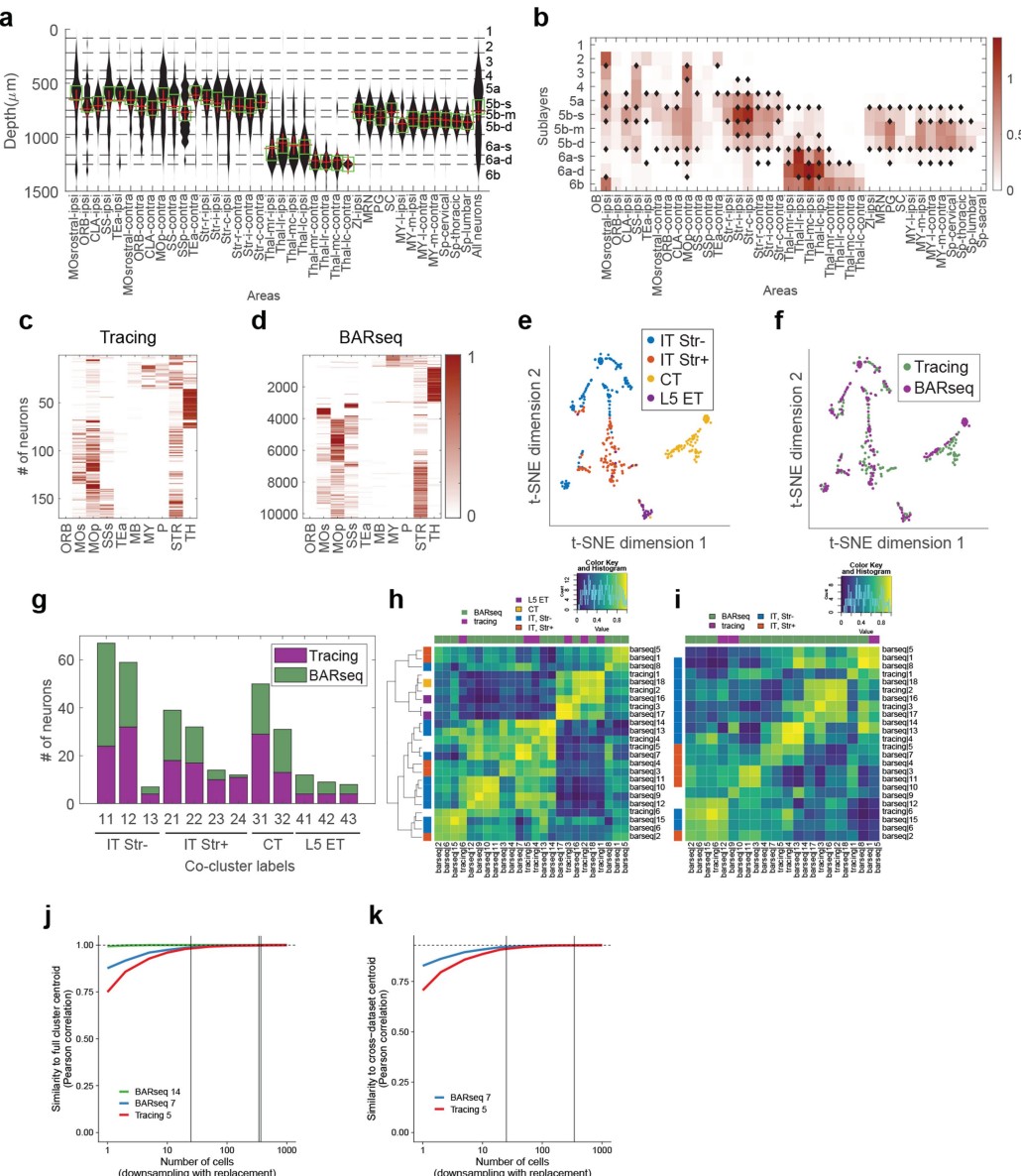

**Extended Data Fig. 18 | related to Fig. 4. BARseq projection mapping in MOp compared to other data modalities. a**, Violin plots of the distribution of neurons with the indicated projections across cortical layers. Red crosses and green squares indicate means and median values, respectively. **b**, Mean normalized projection strengths of neurons in the indicated sublayers. Projection strengths were normalized so that the standard deviation for a projection across all neurons was 0. Black diamonds indicate p < 0.05 for the distribution of the projection strengths in the two adjacent sublayers using two-tailed rank sum test after Holms-Bonferroni correction. p values before correction are shown in Source Data Fig. 2. **c**, **d**, Projection patterns from

single-cell tracing (**c**) and BARseq (**d**) shown at a common resolution achieved by both datasets. **e**, **f**, t-SNE plots of combined BARseq and single-cell tracing datasets color-coded by combined cluster classes (**e**) or by datasets (**f**). **g**, Number of neurons in each combined clusters that belong to each dataset. **h**, **i**, MetaNeighbor analysis of subgroups of all neurons (**h**) or IT neurons (**i**) identified by BARseq and tracing. Higher scores reflect stronger similarity between clusters. **j**, **k**, Similarity of sub-sampled BARseq and tracing neurons of the indicated clusters to their cluster centroids (**j**) or to the centroids of the matching clusters in the other dataset (**k**).

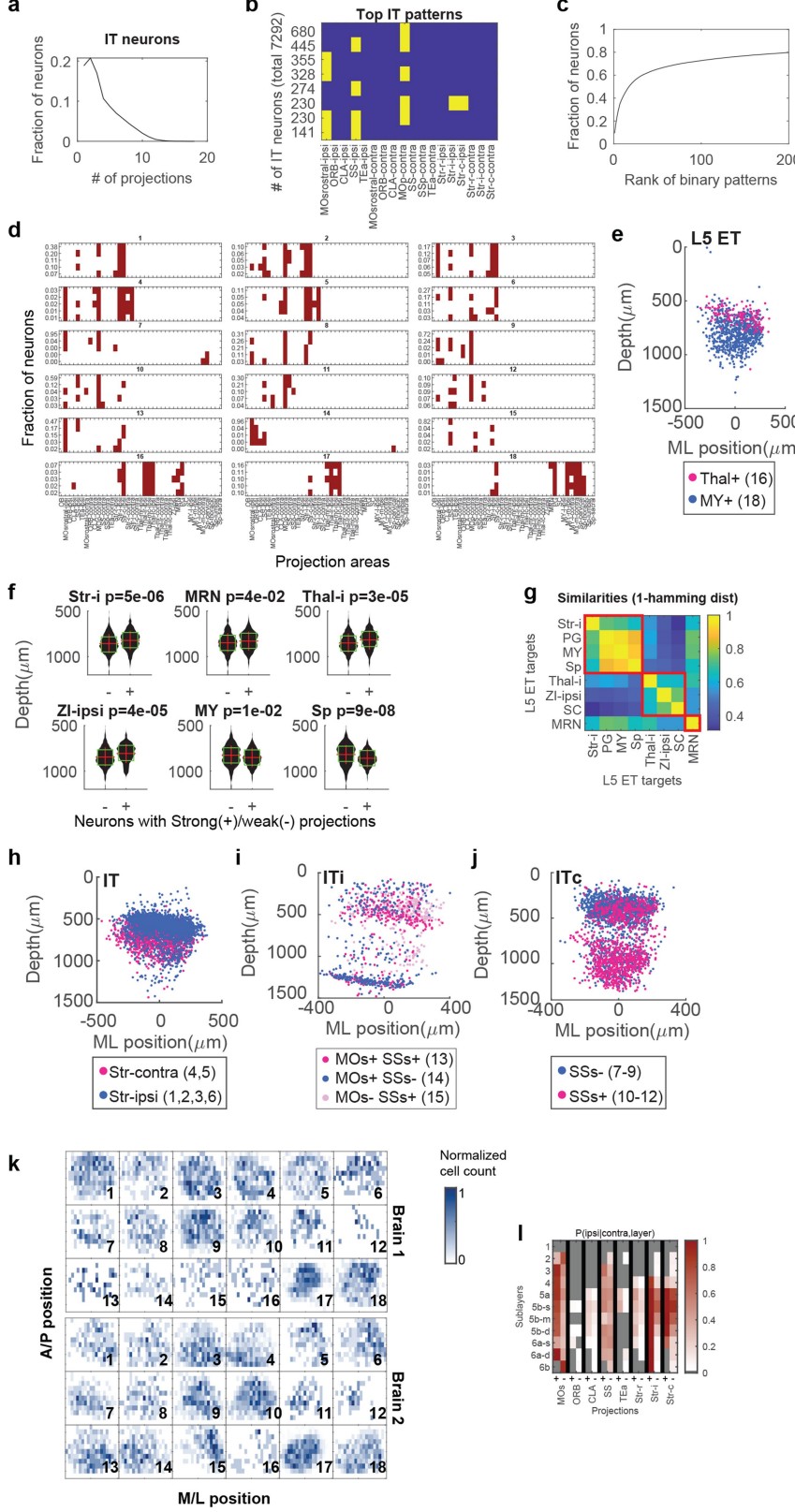

**Extended Data Fig. 19** | See next page for caption.

**Extended Data Fig. 19 | related to Fig. 4. BARseq projection mapping in MOp. a**, Distribution of projection numbers of IT neurons. **b**, Top binary projection patterns of IT neurons. The number of neurons with each pattern is indicated on the left. **c**, Cumulative fractions of IT neurons (y-axis) with the indicated number of binary projection patterns (x-axis). The projection patterns are sorted by their abundances, so the most common patterns are on the left. **d**, The 5 most abundant binary projection patterns in each subgroup. The fractions of neurons are indicated on the left and the subgroup numbers are indicated on top of each graph. **e**, Scatter plot of soma locations of the indicated subgroups in the cortex. X-axes indicate relative medial-lateral positions, and y-axes indicate laminar depth. Group numbers are shown in parentheses. Major classes to which the neurons shown belonged to are indicated above each panel. **f**, laminar distribution of neurons in group 18 with strong (+) or weak (-) projections to the indicated areas. P values using two-tailed rank sum tests after Bonferroni correction are shown on top of each

panel. **g**, similarities between projection targets of L5 ET calculated. The similarity is defined as one minus the hamming distance between two areas based on their binarized co-innervation pattern across neurons of both group 16 and 18. Red squares indicate clusters identified by louvaine community detection. **h**, **i**, **j**, The soma locations of the indicated subgroups of neurons. X-axes indicate relative medial-lateral positions, and y-axes indicate depth. Neurons are colored by subgroups as indicated. Subgroup numbers are shown in parentheses. **k**, Density maps of each subgroup of neurons on the tangential plane. Neurons from the two brains are shown separately to distinguish labeling bias from real biases in distribution. The density maps are normalized so that the highest density is 1 in each plot. Subgroup numbers are indicated on each plot. **l**, The projection probability for the indicated ipsilateral projection (x-axis) conditioned on whether the neuron project to the same contralateral area in the indicated sublayer (y-axis).

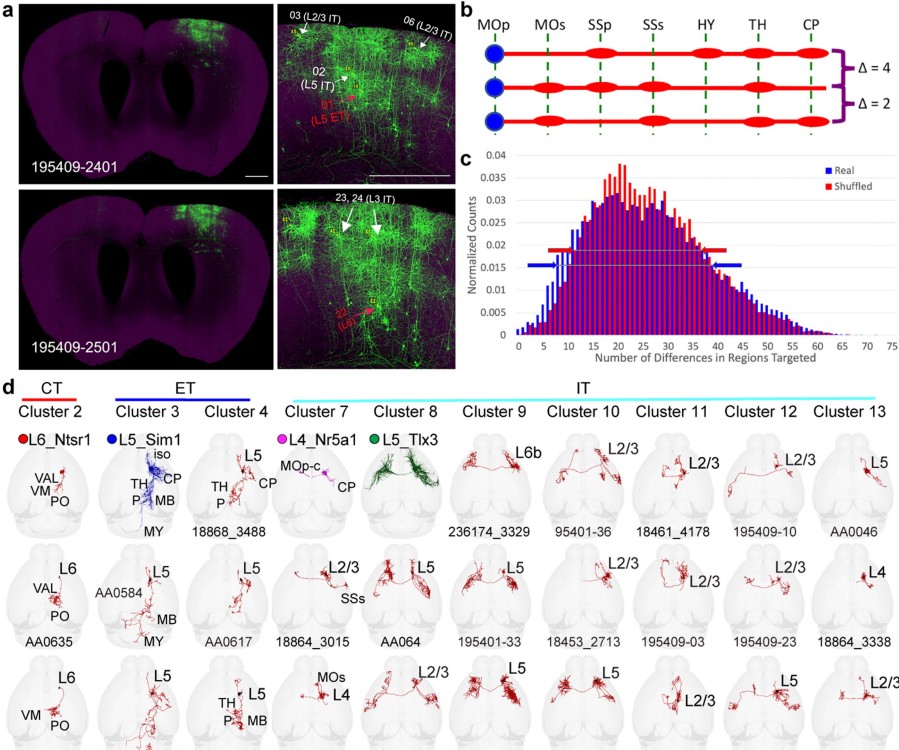

**Extended Data Fig. 20 | related to Fig. 5. Single neuron reconstruction data.**
**a**, Digital images (left panel) show two coronal planes (2401 and 2501) through the viral injection site in one representative case (195409). The sections were counterstained with propidium iodide-stained (PI) cellular nuclei to reveal cytoarchitectonic background and facilitate identification of soma locations of labeled neurons. Four L2/3 IT neurons (#03, 06, 23, 24), one L5 ET (#01) and one L6 (#22) neurons were selected for reconstruction. Scale bars, 500 μm. **b**, **c**, Analyses of projection target patterns for MOp neurons from the MouseLight dataset and schematic of cell type specific networks. **b**, Schematic depiction of the major targets contacted by three MOp cells (identified by their MouseLight name) and the pairwise comparisons to quantify the differences in regions targeted (Δ values to the right). **c**, Histogram of pairwise differences in

regions targeted by MOp neurons ("real") compared to those with randomized targeted regions while normalizing the number of regions invaded by each neuron and the number of neurons invading each region ("shuffled"). The real distribution is broader than the shuffled distribution ($CV_{shuffled}$=0.431, half-height-width$_{shuffled}$=25; $CV_{real}$=0.479, half-height-width$_{real}$=31; half-height widths pairs of horizontal arrows) and the statistical difference is highly significant (1-tailed Levene's test of variances: $p < 10^{-25}$). The left tail reflects similarity between neurons likely in the same projection class; the right to differences between neurons from different classes. **d**, Top views of the CCF show the brain-wide reconstructions rendered in 3D from example Cre line tracer experiments (colored by layer and projection class key) or individual single cells (red) assigned to each cluster.

# MOp-ul regional & cell type networks

## Somatic sensorimotor cortical subnetworks

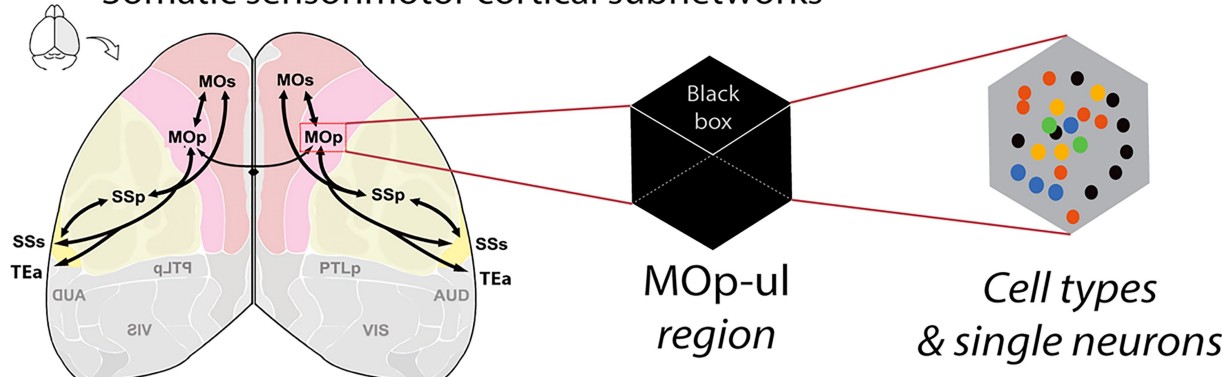

## Single target & *broadcasting* neurons

## *Multi-target* neurons

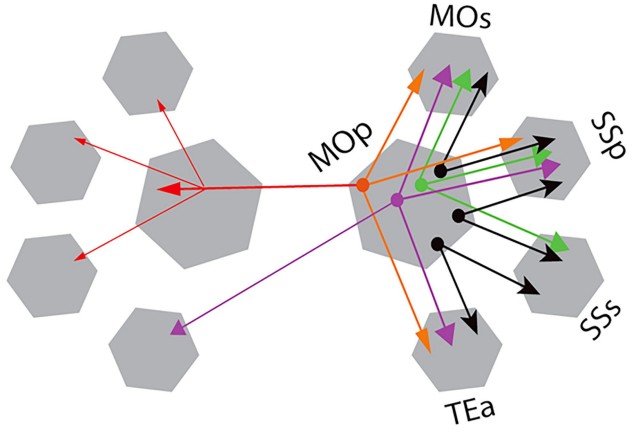

Contralateral   Ipsilateral

Contralateral   Ipsilateral

**Extended Data Fig. 21 | MOp-ul regional & cell type networks.** MOp-ul shares extensive bidirectional connections with other cortical areas, including bilateral projections to MOs, SSp, SSs, TEa, and to the contralateral MOp. Although MOp-ul is considered as a single gray matter region node (macroscale), the more than 15 IT projection neuron types revealed here suggest a very complex network at the mesoscale. Two network schematics show three general categories of IT cell type-specific connections: (1) *single target* types (black arrows, one target, e.g., MOp-ul to MOs); (2) *broadcasting types*, in which one cell type innervates many cortical targets of MOp-ul (red); (3) *multiple combinatorial targets* types, in which one cell type projects variably to several targets. Note that in this model and in the case of multiple models coexisting, each cortical target receives inputs from multiple types of MOp-ul IT neurons.

# Reporting Summary

## Statistics

For all statistical analyses, confirm that the following items are present in the figure legend, table legend, main text, or Methods section.

| n/a | Confirmed | |
|---|---|---|
| ☐ | ☒ | The exact sample size (*n*) for each experimental group/condition, given as a discrete number and unit of measurement |
| ☐ | ☒ | A statement on whether measurements were taken from distinct samples or whether the same sample was measured repeatedly |
| ☐ | ☒ | The statistical test(s) used AND whether they are one- or two-sided *Only common tests should be described solely by name; describe more complex techniques in the Methods section.* |
| ☒ | ☐ | A description of all covariates tested |
| ☐ | ☒ | A description of any assumptions or corrections, such as tests of normality and adjustment for multiple comparisons |
| ☐ | ☒ | A full description of the statistical parameters including central tendency (e.g. means) or other basic estimates (e.g. regression coefficient) AND variation (e.g. standard deviation) or associated estimates of uncertainty (e.g. confidence intervals) |
| ☐ | ☒ | For null hypothesis testing, the test statistic (e.g. *F*, *t*, *r*) with confidence intervals, effect sizes, degrees of freedom and *P* value noted *Give P values as exact values whenever suitable.* |
| ☒ | ☐ | For Bayesian analysis, information on the choice of priors and Markov chain Monte Carlo settings |
| ☒ | ☐ | For hierarchical and complex designs, identification of the appropriate level for tests and full reporting of outcomes |
| ☐ | ☒ | Estimates of effect sizes (e.g. Cohen's *d*, Pearson's *r*), indicating how they were calculated |

*Our web collection on statistics for biologists contains articles on many of the points above.*

## Software and code

Policy information about availability of computer code

| Data collection | Several microscopic methods were used to collect fluorescent imaging data: 1). epifluorescence images were collected with the Olympus VS120 fluorescence microscope running Olympus VS-Desktop v2.9; 2)High resolution confocal images were captured using an Andor DragonFly 202 spinning disk confocal microscope running Fusion v2.1.0.81 software; 3) Lightsheet images were captured with a LifeCanvas lightsheet microscope running SmartSPIM Acquisition Software 2019V3 and oblique Light-sheet tomography (OLST) running custom open source software; 4) 3D fluorescent-labeled pathway images were collected using Serial Two-Photon Tomography (STPT) instruments with TissueVision software 5) single neuron morphology data were collected using fluorescence micro-optical sectioning tomography system (fMOST); 6)BARseq data were collected using an Olympus IX81 microscope with a Crest X-light v2 spinning disk confocal, an 89north LDI 7-channel laser, and a Photometrics Prime BSI camera. Image acquisition was controlled through micro-manager. STPT images at the Allen Institute were processed using the Allen informatics data pipeline (IDP), which manages the processing and organization of the images and quantified data for analysis and display in the web application as previously described (Oh et al., 2014 and Kuan et al. 2015). STPT images at CSH were porcessed with custom open source OpenSTPT software. |
|---|---|
| Data analysis | We used various computational/informatics methods for data analysis, which are all described in detail in Methods section. Novel code will be made publicly accessible through Github or other repositories as indicated in the Methods. For Figure 4, 5, and 7, unsupervised hierarchical clustering was conducted with the online software, Morpheus, (https://software.broadinstitute.org/morpheus/) for algorithms and for visualization of the dendrogram. The software program GraphPad Prism was used for statistical tests and generation of graphs. |

For manuscripts utilizing custom algorithms or software that are central to the research but not yet described in published literature, software must be made available to editors and reviewers. We strongly encourage code deposition in a community repository (e.g. GitHub). See the Nature Portfolio guidelines for submitting code & software for further information.

## Data

Policy information about availability of data

All manuscripts must include a data availability statement. This statement should provide the following information, where applicable:

- Accession codes, unique identifiers, or web links for publicly available datasets
- A description of any restrictions on data availability
- For clinical datasets or third party data, please ensure that the statement adheres to our policy

All the data underlying the results described in this work will be or already are deposited in the Brain Image Library at the Pittsburgh Supercomputing Center and publicly accessible without restrictions or credentials at biccn.org/data through the BICCN Data Inventory hosted by the Brain Cell Data Center at the Allen Institute for Brain Science.

# Field-specific reporting

Please select the one below that is the best fit for your research. If you are not sure, read the appropriate sections before making your selection.

☒ Life sciences ☐ Behavioural & social sciences ☐ Ecological, evolutionary & environmental sciences

For a reference copy of the document with all sections, see nature.com/documents/nr-reporting-summary-flat.pdf

# Life sciences study design

All studies must disclose on these points even when the disclosure is negative.

| | |
|---|---|
| Sample size | The sample sizes for different injection methods with different tracers were specified in Methods sections as described for different laboratories. |
| Data exclusions | The best most representative injections were chosen for the analysis. The others were excluded due to off-targeting of the injection site, missing/damaged tissue, weak tracer labeling of the axons or high background, etc. |
| Replication | This study focuses on characterizing inputs/outputs of the primary motor cortex upper limb area (MOp-ul) using different tracing methods. Each of tracer injections were repeated multiple times in different animals. While the best, most representative cases were chosen for inclusion in the analysis data set, the other injections served as validation cases, demonstrating the replicability and consistency of tracer labeling. |
| Randomization | Randomization is not relevant to the present work since animals were not compared across different conditions |
| Blinding | Traditional blinding was not necessary since animals were not compared across different conditions in this study. |

# Reporting for specific materials, systems and methods

We require information from authors about some types of materials, experimental systems and methods used in many studies. Here, indicate whether each material, system or method listed is relevant to your study. If you are not sure if a list item applies to your research, read the appropriate section before selecting a response.

### Materials & experimental systems

| n/a | Involved in the study |
|---|---|
| ☐ | ☒ Antibodies |
| ☒ | ☐ Eukaryotic cell lines |
| ☒ | ☐ Palaeontology and archaeology |
| ☐ | ☒ Animals and other organisms |
| ☒ | ☐ Human research participants |
| ☒ | ☐ Clinical data |
| ☒ | ☐ Dual use research of concern |

### Methods

| n/a | Involved in the study |
|---|---|
| ☒ | ☐ ChIP-seq |
| ☒ | ☐ Flow cytometry |
| ☒ | ☐ MRI-based neuroimaging |

## Antibodies

| | |
|---|---|
| Antibodies used | [antibody; vendor; catalog number]<br>1. rabbit anti-Phaseolus vulgaris leucoagglutinin antibody; Vector Labs; #AS-2300<br>2. monoclonal mouse anti-Cre recombinase, clone 2D8; Millipore Sigma; #MAB3120<br>3. rabbit anti-dsRed antibody, Rockland Cat# 600-401-379, RRID:AB_2209751<br>4. Mouse anti-Neurofilament-M antibody (monoclonal), Encor biotechnology, #MCA-3H11 |

5. Guinea pig anti-NeuN antibody (Polyclonal), Synaptic systems, #266004

| Validation | Supporting documentation as to the validity of the above antibodies can be found at the following:<br>1. https://antibodyregistry.org/search.php?q=AB_2313686<br>2. https://antibodyregistry.org/search.php?q=AB_2085748 and https://www.emdmillipore.com/US/en/product/Anti-Cre-Recombinase-Antibody-clone-2D8,MM_NF-MAB3120#documentation<br>3.https://antibodyregistry.org/search.php?q=AB_2209751<br>4. https://encorbio.com/product/mca-3h11/<br>5. https://sysy.com/product/266004 |
|---|---|

# Animals and other organisms

Policy information about studies involving animals; ARRIVE guidelines recommended for reporting animal research

| Laboratory animals | Mus musculus, male and female, 2-month old, wild type C57Bl6, Cre driver transgenics and reporters, some obtained from Jackson Laboratories.<br>All animal procedures were performed under Institutional Animal Care and Use Committee (IACUC) approval (Allen Institute for Brain Science, Cold Spring Harbor Laboratory, University of Southern California, MIT, Huazhong University of Science and Technology in China) in accordance with NIH guidelines. Mice had ad libitum access to food and water and were group-housed within a temperature- (21-22°C), humidity- (40-51%), and light- (12hr light/dark cycle) controlled room within the vivariums of the institutes listed above. Male and female wild type C57BL/6J mice at an average age of P56 were purchased from Jackson Laboratories for histological, multi-fluorescent tract tracing and viral tracing experiments, and single neuron reconstructions. The mouse lines used at different institutes for specific experiments are described below and listed in Extended Data Table 2. |
|---|---|
| Wild animals | No wild animals were used in this study |
| Field-collected samples | No field samples were collected for this study. |
| Ethics oversight | Ethical oversight of experimental procedures was performed by the Institutional Animal Care and Use Committee (IACUC) of the Cold Spring Habor Laboratories, University of Southern California, Allen Institute, UCLA, UCSD, MIT, Penn State University, and the Institutional Ethics Committee of Huazhong University of Science and Technology. |

Note that full information on the approval of the study protocol must also be provided in the manuscript.

