## [Peer Review File · Nature]

Manuscript Title: Cellular Anatomy of the Mouse Primary Motor Cortex

Editorial Note

Redactions--Third Party Material

Reviewer Comments & Author Rebuttals

Reviewer Reports on the Initial Version:

Referee #1 (Remarks to the Author):

Muñoz-Castañeda et al report on a project aimed at anatomically classifying and characterizing projection neurons (PN) in motor cortex in terms of how subclasses distribute into layers and sublayers, and connect with other brain centers. The study employs experimental and informatics methods ranging from macro level anterograde and retrograde labeling to micro single-axon analysis and more. I find the most interesting aspects to be those regarding classification of diverse intratelencephalic populations and also the quantitative estimates of the fractions of MOp-fl projections to diverse targets.

However, this seems to constitute mainly a technical exercise to systematically apply multiple complementary tools to generate a large dataset for a particular cortical area in the mouse. The overall scientific advance seems limited. This is especially so when considered in the context of the relevant literature, which is inadequately acknowledged. Nor are all the claims fully convincing. Some of the studies are essentially descriptive impressions and lack rigorous quantitation. The main claim (Abstract) is that they "defined over two dozen MOp-ul projection neuron (PN) types by their anterograde targets; the spatial distribution of their somata defines 11 cortical sublayers, a significant refinement of the classic notion of cortical laminar organization.". But haven't these layers and sublayers already been reasonably well described in motor and sensory cortex? The "six layered" cortex is basically a strawman, as the presence of more than six layers/sublayers is well known as exemplified by the striate cortex of several species. In the Results it turns out that the 25 PNs are only "suggested" by the laminar distribution patterns of retrogradely PNs, shown in Fig 3 as examples but lacking statistics to support assignment of subtypes in the diagram in Fig 3c. The study is presented as if rodent motor cortex is a terra incognita, but cellular anatomy of rodent motor cortex and related areas have been previously described in terms of layers/sublayers and PNs and also their afferents and efferents, from many groups including Alloway, Brecht, Empson, Gerfen, Jones, Kaneko, Kawaguchi, Kita, Nelson, Oberlaender, Shepherd, Svoboda, Yoshida, among others. While the study provides welcome additional information there is little in terms of bona fide new scientific insight or substantive conceptual revisions. As a random example, multiple previous reports show cortical projections to contralateral thalamus, but here this is presented as a "previously unreported" discovery. Also "previously unreported" is the projection to cerebellum, which in this case would indeed be novel but without supporting evidence such as single axon reconstructions must be viewed as tentative, as the concern for artifact is high. The observation that "L5 IT cells have more CP projections compared to L2/3" is also presented as original, but this has long been known. The finding from the monosynaptic tracing of a lack of selectivity is odd, given the well known laminar specificity in cortex of various afferent axons from different sources, and here too raises concerns about the methodology. Thus, the diagram in Fig 8C, is hardly as original and definitive as implied but in fact includes a mix of results either somewhat confirmatory of prior observations or somewhat novel but minor and unconvincing.

Another claim is about defining the boundaries of MOp-fl. But is there real dispute or uncertainty about the border between MOp and SSp? This appears well established in standard atlases, and the supposed identification of the other less distinct borders areas is not particularly convincing. Unless I'm mistaken it also seems that markers (VGluT3+ etc) were selected mainly to match prior demarcations of borders based on other markers, which seems biased. It is also unclear how meaningful the borders defined anatomically here are, and how they compare to the more complex and interesting maps from functional characterizations of forelimb cortex such as cortical

stimulation and recording. A substantial literature in this area is relevant, from groups of Isomura, Jones, Murphy, Petersen, Waters, some of those mentioned above, and more.

The study has numerous minor observations, such as “L5 neurons had larger and more complex basal trees than neurons in superficial layers”, that represent a type of routine neuro morphometry not particularly illuminating and more suited for a specialised journal.

Small sample sizes in terms of animal numbers are a further concern. The numbers of cells and mouse lines are large but often with very low numbers of animals per line, experiment types, and so on, and indeed often just 1 or 2 mice (as listed in the tables in the figures, such as Figure 4a, 5a, etc.).

The statement that “... large scale complete single cell reconstruction provides the ultimate desired resolution to achieve ground truth discovery and an internally consistent classification of anatomic cell types” is compelling. But all the more disappointing that the dataset consists mainly of the 100 or so MOp axons from the MouseLight, with about 50 additional fMOST axons, and the Winnubst et al study already analyzed their MOp axons in considerable detail. Isn't this mostly just reanalysis? How many of the reconstructed cells are even actually within MOp-fl? Many cells seem to lie outside (see Fig .7, images in panels b, c, d, etc showing collections of neurons/axons). Similarly, in some of the other studies, injection sites appear to be conspicuously dispersed such as in Fig 4b and 5b, with some sites outside the borders of MOp-fl, which they went to so much trouble to define.

Another methodological concern is overreliance on Cre lines. By choosing which to study, this introduces a bias. Do any of the lines label any single anatomically-defined PN class or subclass particularly selectively?

Referee #2 (Remarks to the Author):

Review of : Cellular Anatomy of the Mouse Primary Motor Cortex

This manuscript presents the collaborative effort of the Brain Initiative Cell Census Network (BICCN) to provide a comprehensive cell-type description of mouse brain structures. The neuroanatomical techniques used is impressive, spanning classic PHA-L tract tracing, retrograde axonal tracing, targeting of specific neuron subtypes with Cre-drive lines with both anterograde axonal tracing and rabies transsynaptic labeling, single neuron axonal projection labeling, BARcode single cell sequencing and MERfish labeling of neuron populations with all these data sets registered to the Allen Common Coordinate Framework to provide the framework for computational neuroinformatics based analysis. Using the upper limb area of the primary motor cortex as an example the study delivers “ a roadmap towards a cellular description of mammalian brain architecture”.

What is novel about this study is that data from multiple data sets registered to the Allen Common Coordinate Framework (CCF) are used describe the neuroanatomical organization of a single cortical area. The data sets from the different members of the BICCN are extensive and this study provides a blueprint for how such datasets may be used to characterize different elements of cortical organization.

The first step is the non-trivial problem of defining a specific cortical area, the upper limb area of the MOp. Classically, cortical areas have been defined based principally on cytoarchitectural data from Nissl staining. In this study, such data is registered to the CCF and then combined with afferent and efferent labeling registered also registered to the CCF. This approach allows for combining data from multiple animals for each type of labeling to average individual variations.

While this approach appears to clearly define the boundaries of the upper limb of the MOp, several issues should be addressed.

- More convincing would be to apply this approach to areas adjacent MOp areas to establish whether the boundaries between different MOp areas are clear or partially overlapping.
- Would this approach also be applicable to defining specific areas in other types of cortex. For example would this approach be able to identify specific subareas of secondary motor areas, or prefrontal areas (such as distinguishing between for example prelimbic and infralimbic areas. Some areas such as primary motor and somatosensory areas might be able to be subdivided into distinct areas, but other areas may not be so readily parcellated.

The second step is the description of the delineation of the laminar organization of the upper limb of the MOp. For some time it has been recognized that the cytoarchitectural delineation of cortical layers does not adequately characterize the distribution of distinct subtypes of cortical projection neurons or afferents, which underlies cortical functional organization. Confirming many prior studies data presented here show that the distribution of major cortical projection neurons, including intratelencephalic (IT), extra-telencephalic (ET, also referred to as pyramidal tract, PT) and cortico-thalamic (CT) while generally displaying some preferential distribution in layers 2/3 (IT), layer 5 (ET) and layer 6 (CT), is more complicated. Two examples: IT neurons are distributed across all layers and in layer 5 IT and ET neurons are sometimes intermingled. Also, there are many subtypes of ET neurons, based on their projection targets and complicated by the multiple collaterals to different targets individual ET neurons possess (this was first demonstrated at the single cell level by Kita and Kita 2012, which should be referenced). In this study data from using different retrograde tracers, anterograde tracing from Cre-lines expressed in specific IT, ET or CT subtypes and gene expression data are used to map the distribution of projection neuron subtypes in the UL of MOp. While there is nothing ground breaking presented, the detail of the analysis based on extensive data sets is impressive. There are several issues:

- While the classic designation of 6 layered cortex does not adequately characterize the distribution of PN subtypes, subdividing the 6 layers further introduces other issues. First, as presented in their own data, in many cases the boundaries between different sublayers is blurred at best, with considerable intermingling of distinct PN subtypes within the sublayers (particularly in layers 5a,5b,5c). There are many examples, but one that is most obvious is that corticostriatal IT neurons are sometimes described as being confined to layer 5a, but clearly extend and intermingle in layer 5b with various ET PNs. Rather than attempt to define additional sublayers a more accurate characterization is that the distributions of various subtypes are generally distributed in certain laminar patterns, that do not always have clear boundaries.
- Single neuron projection studies have shown that neighboring cortical PN neurons may belong to different subtype classes and that the organization of their relative distribution positions may be shown to be statistically distinct without necessarily being separated by distinct boundaries. (Winnubst et al2019). Such a representation of the ET subtypes seems more accurate than attempting to define sublayers.
- Another question is whether or not the distribution of different ET subtypes (for example) in different sublayers are consistent in different cortical areas. This is a general question regarding the distribution of the many different ET subtypes in different areas. For example it appears that the distribution of medulla and thalamic ET PNs in the UL of MOp is different than that described for ALM (Economo et al. 2018).
- In the Figure 3 caption it is stated that “all corticothalamic neurons (CT) are distributed in layer 6a...”. Why are layer 5 ET neurons that project to the thalamus not considered “cortico-thalamic”.
- For corticostriatal neurons it appears that for this study only those that project contralaterally are considered. Data from single cell axon tracings have shown that individual corticostriatal neurons have very different patterns and distributions of projections such that there are some with very sparse ipsilateral projections with robust contralateral (Winnubst et al., 2019), such that there may be some corticostriatal neurons missed in the data used.
- There is a problem with the “spatial distribution of 40 genes selected from the Allen Brain Gene expression database..(Extended data Fig. 11.)”. Many of those presented are not gene expression data but expression of Cre from the GENSAT Bac-Cre project (Gerfen and Heintz, 2013 and should

be cited as the Cre lines in that paper are used multiple times in this study). First, the lines used are inaccurately identified only by the gene name. For the BAC-Cre lines it is essential to list the actual line, as in Sepw-NP39, Tlx_PL56, Sim1_KJ18, Chrna2_OE25, PlxnD1_OG1, Rbp4_KL100, and Ntsr1_GN220. In all of these cases there are multiple lines for each gene and often the expression patterns differ considerably. For example the Ntsr1-GN220 line expresses in layer 6 of the neocortex, while Ntsr1_GN209 does not express in neocortex at all, but does express in piriform cortex. While the referenced gene may be expressed in the particular PN subtype labeled by Cre expression, it is not necessarily the case.

The data using BARseq and tracing of axonal projections of individual PN cortical neurons provide unprecedented details of the diversity of PN cortical subtypes in terms of the patterns of the multiple targets of individual neurons. The BARseq data identify the major classes of PN cortical neuron subtypes, IT, L5ET and CT subtypes and further subdivide these into at least 18 subgroups on their projection targets. Single neuron axonal projection data also revealed distinct subtypes. Several comments:

- The BARseq data show laminar distributions that correlate generally with that revealed with retrograde tracing data. However, it is not clear whether the patterns of distribution of subtypes of ET PN neurons in sublaminae of layer 5 reported with retrograde tracers are also observed in the BARseq data. The data depicted in Figure 6 may demonstrate that but it is difficult to determine. It does appear from the BARseq data that for L5 ET neurons there are not distinct boundaries between sublaminae.
- The 18 subgroups identified in the BARseq data are not easily discerned from the data depicted in Figure 6. What are the 18 subgroups? While certain subgroups are described in the text a table or some other way of presenting them would be useful to make sense of them.
- Some of the sublaminal descriptions are overstated, for example the distinction of ipsi versus contralateral projecting corticostriatal neurons are described as being in different sublayers, but the data in extended data 19, I shows that there is a general difference, there is a lot of intermingling. More convincing is the select distribution of ipsi only projecting IT neurons in L6, whereas both ipsi and contra projecting IT neurons are intermingled in L5.
- The BARseq data is presented to display the laminar distribution, it would also be informative to display the different subgroupings in the horizontal dimension to see if there is any organization in this plane of neurons with different projection patterns.
- As has been shown from the Janelia MouseLight data, single neuron axon tracing data reveal the complexity of the different PN subtypes based on the diversity of collaterals (particularly those to multiple areas). While the analysis shown as identified 25 subtypes, there are likely other subtypes to be identified as cell projection data increases. Also minor not sure why in figure 9b the different projection patterns are described as models, as the data have shown there are examples of each type.

This study uses multiple types of data sets to provide a comprehensive description of the neuron subtypes in the upper limb area of the MOp. The inclusion of data using classic tract tracing, Cre specific anterograde and trans-synaptic tracing, single cell transcriptomics and axonal projection labeling, all registered into a standard mouse reference framework affirms current concepts of the neuroanatomical organization of the cerebral cortex. While there are not any new organizing principles to emerge from the analysis, the level of detail and the approach used provides a blueprint for addressing fundamental questions. A few examples:

- This study is limited to the description of one subarea of the MOp, would applying this approach to all of the MOp provide a more definitive way of parcellating the MOp and would the same organizational details in the Upper limb area be the same in other areas.
- How does the organization compare between different primary cortical areas and between primary and secondary areas. Specifically, while this approach allows for the clear delineation of the UL area of MOp and would presumably demonstrated similar clear delineation of other specific MOp areas, clear boundaries of subareas of secondary motor areas may or may not emerge. Similarly what about areas more "association" type areas, such as prefrontal and cingulate

cortices.

Determining what elements of the organization of UL MOp are common and which differ between cortical areas will enable developing concepts of how information is organized and processed within cortical circuits to effect behavior. This study provides a roadmap and sets the standard for pursuing such questions.

A suggestion is that the title of the paper include the Brain Initiative Cell Census Network, as this is more about the DataSets and analysis than about the Primary Motor Cortex (also it isn't really about the Primary Motor Cortex but the Upper Limb area).

One additional note is that hopefully the Brain Initiative support of the Cell Census Network intended that data sets produced would be made available to researchers. A major benefit of the Allen Institute is that they not only provide comprehensive data sets but also provide instructions for how to access that data. Providing similar access to all of the data in this study would be beneficial.

Papers that should be cited:

Economo et al. (2018) Distinct descending motor cortex pathways and their roles in movement Nature. 563:79-84.

Gerfen CR, Paletzki R, Heintz N (2013) GENSAT BAC Cre-recombinase driver lines to study the functional organization of cerebral cortical and basal ganglia circuits. Neuron 80:1368-1383.

Kita, T., & Kita, H. (2012). The subthalamic nucleus is one of multiple innervation sites for long-range corticofugal axons: A single-axon tracing study in the rat. The Journal of Neuroscience, 32, 5990_5999.

Author Rebuttals to Initial Comments:

Referee #1 (Remarks to the Author):

Muñoz-Castañeda et al report on a project aimed at anatomically classifying and characterizing projection neurons (PN) in motor cortex in terms of how subclasses distribute into layers and sublayers, and connect with other brain centers. The study employs experimental and informatics methods ranging from macro level anterograde and retrograde labeling to micro single-axon analysis and more. I find the most interesting aspects to be those regarding classification of diverse intratelencephalic populations and also the quantitative estimates of the fractions of MOp-fl projections to diverse targets.

However, this seems to constitute mainly a technical exercise to systematically apply multiple complementary tools to generate a large dataset for a particular cortical area in the mouse. The overall scientific advance seems limited. This is especially so when considered in the context of the relevant literature, which is inadequately acknowledged. Nor are all the claims fully convincing. Some of the studies are essentially descriptive impressions and lack rigorous quantitation. The main claim (Abstract) is that they defined over two dozen MOp-ul projection neuron (PN) types by their anterograde targets; the spatial distribution of their somata defines 11 cortical sublayers, a significant refinement of the classic notion of cortical laminar organization. But haven't these layers and

sublayers already been reasonably well described in motor and sensory cortex? The “six layered” cortex is basically a strawman, as the presence of more than six layers/sublayers is well known as exemplified by the striate cortex of several species.

Response: We thank the reviewer for their thoughtful review which raised several important points. We have endeavored to address their concerns in the revised manuscript, notably by inclusion of extensive new data to strengthen and enhance the rigor of our data and analysis, and by complementary textual revisions that more fully integrate the findings of our study with respect to previous findings from the primary literature. The revisions, elaborated below, clarify and strengthen the scientific advances provided by our novel projection neuron type-based wiring diagram of the primary motor cortex upper limb domain (MOp-ul). A second and broader scientific advance is the first demonstration of a large-scale collaborative and fully data integrated approach to the study of mammalian brain architecture, that provides a roadmap for future studies of other brain areas.

1. Context of our study with respect to other co-submitted BICCN MOp studies

The MOp was selected through mutual agreement by the entire BICCN as an exemplar brain area to explore the potential of collaborative and integrative studies across molecular, electrophysiological and anatomical disciplines to construct a comprehensive cell type-based description of a mammalian brain structure. This effort is synthesized in the BICCN “flagship” paper and described in detail in each BICCN “companion” paper, with most studies applying molecular approaches to describe cell types based on single-cell transcriptomes, DNA methylomes, spatially resolved single-cell transcriptomes, and combined transcriptome, morphological (somatodendritic) and electrophysiological properties. Our companion study provides a key complementary and foundational anatomical analysis to derive a comprehensive cellular resolution input-output wiring diagram that anchors the molecular data within the full MOp framework. In this context, our study is an essential part of a roadmap towards creating a combined molecular and cellular description of mammalian brain architecture, enabling future studies of other brain structures at a similar data depth.

2. Context of our study with respect to MOp literature to date

While MOp has been studied extensively, as with other brain structures to date, a common deficiency is the inability to compare and integrate different datasets within the same spatial framework. This general problem has led to ingrained confusion in structural neuroanatomy. With respect to the MOp, it manifests as (for example) differing delineations of MOp borders (see more below). Here, our collaborative expert consensus-based approach combines state-of-the-art methods for structural neuroanatomy, tracing and viral labeling, imaging and computational data analysis within the mouse brain common coordinate framework (CCF) to generate a model of the MOp that integrates multiple data types and spatial scales. This includes the most comprehensive classification of MOp-ul cortical projection neuron types and their laminar distribution to date, derived from retrograde and anterograde pathway tracing experiments with >300 single neuron morphologies, and precise three-dimensional delineation of the MOp upper limb (MOp-ul) borders, anchoring the resultant input / output connectivity diagram as an integrated model of MOp brain architecture across three spatial scales:

(1) At macroscale: We define for the first time the borders of the MOp-ul in 3D based on cytoarchitectural criteria including data from classic Nissl staining, extensive input / output connectivity data, and cellular genetic Cre-driver expression data (subjected to expert-led as well as machine learning analysis). Importantly, our novel cloud-based Neuroglancer visualization of the co-registered data within CCF at full spatial resolution allowed expert neuroanatomists from different labs to draw and revise borders dynamically in 3D. This expert “crowd-sourcing” overcame the problem of disagreement between competing experts, that is evident in (for example) MOp delineation in different brain atlases, such as that of Paxinos (Franklin and Paxinos, 2019), versus the

Allen Reference Atlas (Dong, 2007), versus Allen CCFv3 (Wang et al., 2020). The combined use of these technologies and approaches sets a new standard for building 3D brain atlases.

(2) At mesoscale: We defined 25 projection neuron types of the MOp based on their stereotyped connectivity, including soma locations and primary projection target regions, across 11 newly defined MOp layer/sublayers. These projection target defined neuron types were further validated by BARseq and single neuron morphology reconstruction data. Together, these data expand substantially a classic interpretation of MOp organization based on IT, PT, CT projection neuron types across 8 cortical sublayers (e.g. Harris and Shepherd, Nat Neurosci. Feb, 2015). We do not claim that our model is final and invariant, but it supports the notion that *classic laminar models of cortical areas may be insufficient and instead could be superseded by integrated layer-gradient based models*.

(3) At microscale: In the initial manuscript submission we reconstructed 53 new single neuron morphologies using the fMOST platform and combined these data with 98 co-registered single neuron morphologies generated by Janelia's MouseLight project. In response to the Reviewer's request for more single neuron data, we have added ~152 newly reconstructed neurons as described in more detail below. Thus the novelty of our study is derived both from the large number of new single cell reconstruction and the integration with the previous datasets co-registered within the mouse brain CCF for a fully integrated analysis, deriving a detailed view of the richness of neuronal projection diversity within major MOp neuron type classes.

In addition to the additional single neuron tracing, we have also applied the emerging technique of high-throughput molecular tracing approach (BARSeq) to characterize single neuron projections from many more MOp neurons (n = 10,299) across 40 brain structures. In the revised manuscript we include a novel analysis comparing these data with the single neuron morphology data, furthering the description of the combinatorial complexity of MOp projection neuron types across target areas.

With respect to previous MOp literature, we thank the Reviewer for bringing to our attention earlier important contributions deserving of acknowledgment, including the following publications: Kita and Kita, 2012; Oswald et al., 2013 (Empson); Economo, et al., 2018; Winnubst et al., 2019; Hooks et al., 2018 (Gerfen), Ueta et al., 2014 (Kawaguchi).

Nevertheless, we would also like to note that the literature on MOp organization in general summarizes across multiple cortical areas, such as general MOp (Kita and Kita, 2012), vibrissal motor cortex (vM1) (i.e., Alloway et al., 2008, 2010; Brecht; Hooks et al., 2018-Gerfen), anterior lateral motor cortex (ALM) (corresponding to the orofacial area, Economo et al., 2018), or forelimb and lower limb (Oswald et al., 2013), with different methods and in different species (rats and mice). While the results in each of these studies are compelling in their own ways, they all include rather small sample sizes from different regions and species, from which it is difficult to draw convincing conclusions. Thus, it is not clear that direct comparison across different functional domains of the MOp is appropriate. For example, in comparison with the MOp-ul, MOp of the mouth domain (partially corresponding to the ALM of Economo et al., 2018) display topographically different terminal fields in their targets, including the dorsal striatum (see Hintiryan et al., 2016, Nature Neurosci) and superior colliculus (Benavidez et al., bioRxiv 2020.03.24.006775). By contrast, our current study generates a comprehensive view of MOp projection neuron types within the same functional domain with diverse data modality which can be directly integrated with molecular and spatial transcriptomics data of the MOp presented in other companion BICCN papers, providing a more holistic view of MOp contrasting with the more fragmented, piecemeal view that can be derived from previous literature.

In summary, the input-output wiring diagram of the MOp-ul described in our study integrates a broad range of technologies, including pathway tracing (classic, viral, and trans-synaptic), molecular barcoding, single neuron reconstruction, 3D whole-brain imaging, and advanced computational analyses and data visualization tools, across macro-, meso- and microscale spatial resolutions,

deriving the most comprehensive view of a brain structure in the mammalian brain to date. Furthermore, we believe that using our study as a roadmap and applying similar approaches across other brain structure will lead to new conceptual models of how information is organized and processed within cortical circuits to drive behavior. Taken altogether, we hope our substantial revisions strengthen and clarify for the Reviewer the advances (both scientific and conceptual) of our study.

3. Scientific justification of our delineation of 11 layer/sublayer of the MOp.

The canonic delineation of 6 cortical layers is mostly based on cytoarchitecture, with more recent description of 8 layers, including deep layers 5a, 5b (distinguished by their cytoarchitectural and long-range connectivity) and 6a, 6b, (L6b is considered as the cortical subplate) and long-range connectivity. The reviewer is correct in pointing out that “the presence of more than six layers/sublayers is well known”, but the key problem is that all previous delineations are largely based on subjective examination of the cytoarchitecture. The key advance in our study is that we delineate the refined layers based on projection neuron types, which are the basic elements of cortical circuit architecture. In the current proposal we have further refined this view to include 11 layers (deep layers 5a, 5b-s, 5b-m, 5b-d, 6a-s, 6a-d, 6b).

To address the reviewer’s concern about the quantitation of our laminar data, we conducted new analyses that are included in the revision. First, we have used hierarchical clustering of the laminar distributions of retrograde labeled cells, Nissl-stained cytoarchitecture, and Cre-expression in the cell type-specific mouse lines to computationally validate the 11-layer delineation initially derived from expert-based analysis of target-specific long-range projection tracing. These results closely recapitulated both the classic 6 layer and the refined 11-layer description based on the level of clustering, hence supporting our laminar delineation. Second, we have added new BARseq analysis to further refine our understanding of the projection patterns of these neuron types across 40 brain structures. BARseq revealed a total of 18 discrete clusters of neurons, which also display distinctive distribution patterns in the newly defined 11 layers/sublayers (Figure 6a,b; Extend data Fig. 21a). Strikingly, each sublayer could be uniquely identified by the top two enriched subgroups of projection neurons (Fig. 6h), which suggest that the 11 sublayers not only are enriched in projections to individual areas, but also reflect laminar organization of neuronal types defined by the overall projection patterns of neurons.

In the Results it turns out that the 25 PNs are only “suggested” by the laminar distribution patterns of retrogradely PNs, shown in Fig 3 as examples but lacking statistics to support assignment of subtypes in the diagram in Fig 3c. The study is presented as if rodent motor cortex is a terra incognita, but **cellular anatomy of rodent motor cortex** and related areas have been previously described in terms of layers/sublayers and PNs and also their afferents and efferents, from many groups including Alloway, Brecht, Empson, Gerfen, Jones, Kaneko, Kawaguchi, Kita, Nelson, Oberlaender, Shepherd, Svoboda, Yoshida, among others. While the study provides welcome additional information **there is little in terms of bona fide new scientific insight or substantive conceptual revisions**. As a random example, multiple previous reports show cortical **projections to contralateral thalamus**, but here this is presented as a “previously unreported” discovery. Also “previously unreported” is the projection to cerebellum, which in this case would indeed be novel but without supporting evidence such as single axon reconstructions must be viewed as tentative, as the concern for artifact is high.

Response: The key distinction from previous studies is the comprehensiveness and coherence of our study that integrates across anatomical levels and scales, using state-of-the-art methods. We defined more than 2 dozen projection neuron types of the MOp based on their stereotyped connectivity, including soma locations and primary projection target regions, across newly defined 11

layer/sublayer MOp organization. To address the Reviewer's concerns, we have conducted the following new analyses.

- We added ~ 150 single neuron reconstructions to increase the total number of single cell morphologies from 151 to 303. Quantitative analysis of single neuron morphology revealed 13 discrete clusters (C2-C13, Figure 7a; Extended Table 5). Each cluster can be further subdivided into multiple smaller clusters with distinctive projection patterns (see section *Classification of projection patterns revealed by complete single-cell reconstruction*). In total, single neuron morphology data support our descriptions of > two dozen projection neuron type in the MOp-ul. Nevertheless, it appears that those reconstructed individual MOp neurons display extremely diverse axonal trajectory patterns and introduce more complexity for cortical neuron classification. We believe that this number (25 PN types) is likely to be an underestimation.
- We performed new BARseq analysis to determine the most common combinatorial patterns of projection targets (i.e., projection patterns of individual neurons to many targets) that are present in each of the newly defined layers. This type of analysis as aforementioned is not possible by single neuron morphology tracings as these are still too laborious to generate sufficiently large numbers of cells.
- We added new cre-dependent AAV anterograde pathway tracing experiments to map collateral projections of different PN types in order to further support and validate the single neuron-based and BARseq-based projection neuron classification.

We believe that these new data and analyses strengthen the scientific conclusions drawn in our study, providing considerable new insights on the complexity of the MOp cell type organization.

The 25 PNs that we reported initially were classified based on their soma locations (i.e., layer 2/3) and projection targets (i.e., TEa) (we changed the wording to “suggested” in this revision). These classifications were further validated and confirmed with other data (BARseq, single neuron morphology, etc). In the revision, we have also conducted quantitative analysis of laminar distributions of these PN neurons in Figure 3.

The observation that “L5 IT cells have more CP projections compared to L2/3” is also presented as original, but this has long been known.

Although it has been known that L5 IT cells have more CP projections compared to L2/3 (which also was described carefully in this study using retrograde tracing, BARseq, and single neuron morphology), we revealed additional novel principles in this study. For example, IT neurons in L2 & 3 have more convergent projections compared to more divergent projections of IT neurons in the middle and deep sublayers (L5a, 5b, and 6a) (Fig. 6q).

The finding from the **monosynaptic tracing of a lack of selectivity is odd**, given the well known laminar specificity in cortex of various afferent axons from different sources, and here too raises concerns about the methodology.

Response: We also initially found surprising the lack of clear qualitative differences in brain-wide input patterns to MOp-ul cells, mapped from Cre driver lines with selectivity for projection neuron classes or with TRIO rabies tracing. However, we were encouraged that these results were not purely methodological artifacts because our results, generated by three different labs (Dong at UCLA, Huang at CSHL and Harris at AIBS), were highly consistent across minor variations in implementing the rabies tracing system. Furthermore, these results are also consistent with findings from recent literature from multiple cortical areas, as referenced previously, “Altogether, these data suggest that input sources to Cre- and target-defined MOp-ul neuron populations are similar, consistent with other recent findings that global input patterns are independent of starter cell type⁴⁷⁻⁴⁹).

References:

Beier, K. T. et al. Topological organization of ventral tegmental area connectivity revealed by viral-genetic dissection of input-output relations. *Cell reports* 26, 159-167. e156 (2019).

Ährlund-Richter, S. et al. A whole-brain atlas of monosynaptic input targeting four different cell types in the medial prefrontal cortex of the mouse. *Nature neuroscience* 22, 657-668 (2019).

Gehrlach, D. A. et al. A whole-brain connectivity map of mouse insular cortex. *bioRxiv*, 2020.2002.2010.941518, doi:10.1101/2020.02.10.941518 (2020).

Even though similar results were observed across multiple labs, it is still important to make sure the lack of selectivity is not due to a technical artifact with Cre-dependent monosynaptic rabies tracing. We expect based on the rabies circuit tracing literature and our viral tools, first, that the ability of the EnvA-pseudotyped rabies virus to infect cells is dependent on the presence of Cre (to mediate Cre-dependent expression of TVA) and, second, that the spread of the glycoprotein-deleted rabies from the starter cell to input cells is dependent on expression of the rabies glycoprotein, provided via AAV to complement the rabies virus. We performed two control experiments at the Allen Institute to demonstrate these assumptions are met. We show that, with our specific AAV helper and rabies virus combination, the spread of the virus outside the injection site depends on (1) the expression of Cre (Reviewer 1 **Fig 1**) and (2) rabies glycoprotein complementation (Reviewer 1 **Fig 2**). Note that **Fig 2** is copied from our previous publication using the same rabies tracing tools (Lo et al., 2019; DOI: [10.1073/pnas.1817503116](https://doi.org/10.1073/pnas.1817503116)). We previously referenced the Lo et al. 2019 paper in the Methods section, but in the revision we expanded the Results text to state these controls were done (“We performed control experiments to show that the spread of rabies-mediated nuclear labeling outside the injection site depends on Cre expression and the supply of rabies glycoprotein (see Fig S1 C,D, in ref 21)”).

Given that the results are not easily explained by a lack of Cre specificity or transcellular spread, it is interesting to consider other biological and technical explanations. For example, while canonical laminar termination patterns are well known, these are generally simplified in ways that highlight the layers with the densest terminations. However, axonal projections that are primarily targeting specific layers can branch and make synapses on cells and their dendrites in the other layers that they traverse (and that would be labeled by different Cre lines). These connections, even if relatively minor, are likely to be captured and revealed through sensitive rabies virus tracing methods.

Finally, although our results suggest a general convergence of inputs to the broad projection cell classes in MOP-ul, they do not rule out the existence of more specific connectivity motifs at the level of more refined cell types than we currently have access to with Cre driver lines or TRIO mapping.

Thus, the diagram in Fig 8C, is hardly as original and definitive as implied but in fact includes a mix of results either somewhat confirmatory of prior observations or somewhat novel but minor and unconvincing.

Response: Figure 8C is a schematic diagram to show a multiscale network model of the MOp-ul summarized based on cell type-specific wiring diagram of the MOp-ul presented, which will provide a conceptual framework for future studies of its cell type-specific functional significance.

Another claim is about defining the boundaries of MOp-fl. But is there real dispute or uncertainty about the border between MOp and SSp? This appears well established in standard atlases, and the supposed identification of the other less distinct borders areas is not particularly convincing. Unless I'm mistaken it also seems that markers (VGluT3+ etc) were selected mainly to match prior demarcations of borders based on other markers, which seems biased. It is also unclear how meaningful the borders defined anatomically here are, and how they compare to the more complex and interesting maps from functional characterizations of forelimb cortex such as cortical stimulation and recording. A substantial literature in this area is relevant, from groups of Isomura, Jones, Murphy, Petersen, Waters, some of those mentioned above, and more.

Response: We appreciate the question regarding the border delineations as it is indeed often assumed that anatomical borders in the mouse brain are well established. However, this is not the case because discrepancies of anatomical delineations are a long-standing issue in neuroanatomy, including cortical parcellation in general and the MOp borders in particular. As shown Figure 3 (below), there is a considerable disagreement in MOp borders as delineated in the three most commonly used mouse

brain atlases – the Franklin and Paxinos (2019), the Allen ARA atlas (Dong, 2008) and the most recent Allen CCFv3 atlas (Wang et al., 2020). This is also clearly described in a recent paper (Chon et al., 2019, Nature Communications). Thus, in order to accurately define 3D anatomical borders for our study, we developed a novel approach based on registering multimodal (Nissl-staining, cell type-specific Cre-driver expression, input/output connectivity data, and whole-brain data (a total of ~30 datasets)) at full resolution (in most cases 1 micron XY-axes with 50 micron Z-axis spacing) on our cloud-based Neuroglancer platform, thus allowing multiple experts at

UCLA, CSHL and AIBS to work collaboratively on these complex data and derive consensus-based anatomical border delineations. In addition to deriving the new brain borders, we believe that this approach sets a new standard for how to construct 3D brain atlases in general.

Reviewer Figure 3: Quantitative comparison of previous MOp delineations. **A)** Overlay of MOp delineations of previous annotations, ARA (blues), Paxinos (green) and CCFv3 (red), at different levels. Displacement in MOp borders are based both on locations and cortical angle definition. **B)** Left panel shows the top view of the newly defined MOp-ul in 3D. Right panel shows the top view overlay of all previous MOp delineations, ARA (blue), Paxinos (green), CCFv3 (yellow) and newly defined MOp-ul (red), identifying mismatches among MOp definitions. Lower panel shows the quantitative

analysis of MOp delineation similarity coefficient (Dice Coefficient), between previous MOp delineations.

The study has numerous minor observations, such as “L5 neurons had larger and more complex basal trees than neurons in superficial layers”, that represent a type of routine neuro morphometry not particularly illuminating and more suited for a specialised journal.

Response: One of the strengths of our study is the comprehensive nature of our MOp-ul description that includes both novel data and concepts on principles of cortical organization as well as more detailed observations. The morphological data referred to by the Reviewer are in fact part of Supplementary data and information and hence not considered a major part of the study.

Small sample sizes in terms of animal numbers are a further concern. The numbers of cells and mouse lines are large but often with very low numbers of animals per line, experiment types, and so on, and indeed often just 1 or 2 mice (as listed in the tables in the figures, such as Figure 4a, 5a, etc.).

Response: The reviewer correctly points out that not all anterograde and retrograde tracer experiments were independently repeated. This is in part because these MOp-ul tracing experiments are part of a much larger effort at the Allen Institute to generate a comprehensive, brain-wide, input/output connectivity diagram from > 200 brain regions. This necessitates striking a balance between the need for broad coverage across Cre lines and source areas with excessive animal use. In support of using a low n per Cre line, we previously showed that $n = 1$ is a strong predictor of anterograde projections across a range of strengths in replicate experiments (Oh et al., 2014) and, more recently, that the correlations between brain-wide projection strengths from experiments at matched locations within the same mouse line are consistent, positive, and significant (Spearman $r > 0.8$, $P < 0.0001$, Extended Data Fig. 1, Harris et al. 2019). Gamanut et al. (2018; <https://doi.org/10.1016/j.neuron.2017.12.037>) also performed extensive analyses of repeated data using retrograde tracer injections in mouse cortex and report surprisingly low variability, particularly for medium to strong connections.

The statement that “... large scale complete single cell reconstruction provides the ultimate desired resolution to achieve ground truth discovery and an internally consistent classification of anatomic cell types” is compelling. But all the more disappointing that the dataset consists mainly of the 100 or so MOp axons from the MouseLight, with about 50 additional fMOST axons, and the Winnubst et al study already analyzed their MOp axons in considerable detail. Isn't this mostly just reanalysis? How many of the reconstructed cells are even actually within MOp-fl? Many cells seem to lie outside (see Fig 7, images in panels b, c, d, etc showing collections of neurons/axons). Similarly, in some of the other studies, injection sites appear to be conspicuously dispersed such as in Fig 4b and 5b, with some sites outside the borders of MOp-fl, which they went to so much trouble to define.

Response: Despite the recent technological advances in imaging methods and the growing use of machine learning for neuronal reconstructions, the study of complete single neuron morphology remains a difficult and fairly low-throughput effort. Nevertheless, we believe we have now significantly extended both the depth/coverage in the single cell data and related analyses in the revised manuscript. We want to point out that we also complemented the single neuron tracing data with the much higher-throughput BARseq method to derive additional information on the combinatorial nature of the single MOp-ul neuron projections.

Since the first submission, we doubled the number of single cell morphological reconstructions, adding ~150 reconstructed neurons for a new total of 303 MOp cells (97 neurons in L2-4, 127 in L5 and 79 in L6), among which $n= 113$ are located in the newly defined MOp-ul.

The Janelia MouseLight project accounts for ~ 1/3 of all the cells included in our revised analyses ($n=121$). Their motor cortex dataset is biased toward deep layer neurons (100/121 cells are in L5 or L6), with relatively few superficial layer IT neurons represented. We complemented this dataset by reconstructing relatively more superficial neurons within the BICCN labs. Notably, all groups contributing reconstructions provided some cells from all layers and projection types, but not with

sufficient numbers to make meaningful observations on the diverse projection patterns. We confirmed that reconstructed cells obtained from different groups were distributed across all clusters (i.e., no cluster was driven by an individual lab). Thus, we argue that rather than characterizing our efforts as “just reanalysis”, it should instead be interpreted as a major strength of the paper that we demonstrated the successful integration of newly generated data with publicly available datasets in the CCF. (Please see section *Classification of projection patterns revealed by complete single-cell reconstruction*, where we have provided much more comprehensive analysis of different PN types).

In our revision, we have also conducted several new analyses to further improve the description of the included single neuron morphology data:

1. We provide a detailed view of all the soma locations in the cortex to show more clearly which cells are located in MOp-ul specifically, and which are in other domains of MOp (Figure 7a).
2. We provide a high resolution “Nissl”-like (DAPI) background for reconstructed neurons to enhance the visualization of the precise anatomical locations in the MOp; (a new Extended Data Figure 22).
3. As discussed above, we performed further comparisons of single neuron axonal projection motifs with BARseq and cell type-specific axonal tracing data to characterize the most “common” collateral projection patterns across the different layers.

Another methodological concern is overreliance on Cre lines. By choosing which to study, this introduces a bias. Do any of the lines label any single anatomically-defined PN class or subclass particularly selectively?

Response: It is a fair point that much of the work in this paper relies on Cre driver lines to genetically access and label populations of neurons. However, we selected these lines based on a combination of direct experience in our labs or the GENSAT project, validation studies, and robust and extensive characterization of anatomical expression patterns and projection mapping (Gong et al., 2007; Gerfen et al., 2013, Harris et al. 2014, Kim et al., 2015; Harris et al., 2019; Matho et al. 2021). We agree with the reviewer that each Cre line will have its own biases, but argue that is a driving reason to then perform experiments across many different Cre lines, with their already established selectivity biases, to achieve broader coverage of different cell populations and reduce sampling bias. Note that we also performed pathway tracing experiments and single cell reconstructions in wild type mice (i.e. in addition to the Cre driver lines).

Although most Cre driver lines we selected here generally label broad classes of projection neurons (as extensively characterized in Harris et al., 2019 by layer and/or IT, ET, CT type and replicated here), some appear to also label a mix of transcriptomically defined cell types (Tasic et al. 2016). Our own data suggests that, of the ~8 Cre lines used for single cell tracing, there is still some diversity within the individual projection motifs of MOp cells from each line (represented by cells being assigned to different clusters), but the broader projection class identities per Cre line (IT, PT, CT) are consistent at the single cell level.

To be more explicit about which mouse lines we used, we also added a table with information about the construction of the line, originating lab, and original references (new Extended Data Table 2).

Referee #2 (Remarks to the Author):

This manuscript presents the collaborative effort of the Brain Initiative Cell Census Network (BICCN) to provide a comprehensive cell-type description of mouse brain structures. The neuroanatomical techniques used is impressive, spanning classic PHA-L tract tracing, retrograde axonal tracing, targeting of specific neuron subtypes with Cre-drive lines with both anterograde axonal tracing and rabies transsynaptic labeling, single neuron axonal projection labeling, BARcode single cell sequencing and MERfish labeling of neuron populations with all these data sets registered to the Allen Common Coordinate Framework to provide the framework for computational neuroinformatics based analysis. Using the upper limb area of the primary motor cortex as an example the study delivers “a roadmap towards a cellular description of mammalian brain architecture”.

What is novel about this study is that data from multiple data sets registered to the Allen Common Coordinate Framework (CCF) are used to describe the neuroanatomical organization of a single cortical area. The data sets from the different members of the BICCN are extensive and this study provides a blueprint for how such datasets may be used to characterize different elements of cortical organization.

The first step is the non-trivial problem of defining a specific cortical area, the upper limb area of the MOp. Classically, cortical areas have been defined based principally on cytoarchitectural data from Nissl staining. In this study, such data is registered to the CCF and then combined with afferent and efferent labeling registered also registered to the CCF. This approach allows for combining data from multiple animals for each type of labeling to average individual variations.

We thank the Reviewer for his/her positive comments about the novelty and importance of this study.

While this approach appears to clearly define the boundaries of the upper limb of the MOp, several issues should be addressed.

- More convincing would be to apply this approach to areas adjacent MOp areas to establish whether the boundaries between different MOp areas are clear or partially overlapping.

Response: While this is out of the scope of the current study, we agree that the same approach can be applied to delineate the borders of other MOp domains (i.e., MOp mouth, MOp-lower limb, MOp-trunk, and MOp whisker), determining whether boundaries between adjacent MOp domains are clear or partially overlapping as well as to borders of other cortical and subcortical areas. Such efforts will be part of follow up studies by the BICCN anatomy group.

- Would this approach also be applicable to defining specific areas in other types of cortex. For example would this approach be able to identify specific subareas of secondary motor areas, or prefrontal areas (such as distinguishing between for example prelimbic and infralimbic areas. Some areas such as primary motor and somatosensory areas might be able to be subdivided into distinct areas, but other areas may not be so readily parcellated.

Response: We agree that the same approach – collaborations across multiple research sites pursuing multimodal data co-registered in CCF and displayed online using the Neuroglancer – can be applied to spatially refining other cortical areas. For example, the Dong lab has recently applied similar multimodal approach to defining the border between the prelimbic and infralimbic areas according to their afferent and efferent connections with the basolateral and lateral amygdalar nuclei (Hintiryan et al., bioRxiv 807743; doi: <https://doi.org/10.1101/807743>; *Nature Communications, in press*) (see below). To follow up on this study, projection neuron types of the prelimbic and infralimbic areas need to be carefully classified following the same approach described in the current study.

The second step is the description of the delineation of the laminar organization of the upper limb of the MOp. For some time it has been recognized that the cytoarchitectural delineation of cortical layers does not adequately characterize the distribution of distinct subtypes of cortical projection neurons or afferents, which underlies cortical functional organization. Confirming many prior studies data presented here show that the distribution of major cortical projection neurons, including intratelencephalic (IT), extra-telencephalic (ET, also referred to as pyramidal tract, PT) and cortico-thalamic (CT) while generally displaying some preferential distribution in layers 2/3 (IT), layer 5 (ET) and layer 6 (CT), is more complicated. Two examples: IT neurons are distributed across all layers and in layer 5 IT and ET neurons are sometimes intermingled. Also, there are many subtypes of ET neurons, based on their projection targets and complicated by the multiple collaterals to different targets individual ET neurons possess (this was first demonstrated at the single cell level by Kita and Kita 2012, which should be referenced). In this study data from using different retrograde tracers, anterograde tracing from Cre-lines expressed in specific IT, ET or CT subtypes and gene expression data are used to map the distribution of projection neuron subtypes in the UL of MOp. While there is nothing ground breaking presented, the detail of the analysis based on extensive data sets is impressive. There are several issues:

Response: We agree with the reviewer that the classic 5- or 6- layer or even 8-layer (with 5a, 5b and 6a, 6b delineations) does not capture the complexity of projection neuron type distribution across cortex. In the current study we extend this description to include 11 layers (adding deep layers 5a, 5b-s, 5b-m, 5b-d, and 6a-s, 6a-d, 6b) and in the revised manuscript we include additional data and analysis to support this MOp organization, including hierarchical dendrogram clustering of the laminar distributions of retrogradely labeled cells, Nissl-stained cytoarchitecture, and Cre-expression in the cell type-specific mouse lines, and new BARseq analysis to further refine our data on the projection patterns of these neuron types across approximately 40 brain structures surveyed by this assay.

Importantly, rather than proposing that the 11-layer classification is the new doctrine, we would like to propose that laminar cortical organization comprises a considerable projection neuron type diversity distributed in a gradient across the classically defined cortical layers, which we expect will continue to be further refined by additional single cell morphology studies combined with spatial transcriptomic studies (please see Zhang et al., BioRxiv, 2020.06.04.105700. <https://doi.org/10.1101/2020.06.04.105700>).

We thank the Reviewer for pointing out to us the omission of the Kita and Kita, 2012 citation and we have corrected this in the revised manuscript.

- While the classic designation of 6 layered cortex does not adequately characterize the distribution of PN subtypes, subdividing the 6 layers further introduces other issues. First, as presented in their own data, in many cases the boundaries between different sublayers is blurred at best, with considerable intermingling of distinct PN subtypes within the sublayers (particularly in layers 5a,5b,5c). There are many examples, but one that is most obvious is that corticoatrial IT neurons are sometimes described as being confined to layer 5a, but clearly extend and intermingle in layer 5b with various ET PNs. Rather than attempt to define additional sublayers a more accurate characterization is that the distributions of various subtypes are generally distributed in certain laminar patterns, that do not always have clear boundaries.

Response: This is a very important suggestion. In the current manuscript the described 11-layer pattern is supported by both anatomy expert-based analysis of the data and unbiased computational analysis of the different cytoarchitectural and projection type and cell type distribution. But as described above, we do share the Reviewer's view that the PN distribution across layers can be perhaps better described as more or less continuous gradients across the cortical depth than clearly defined layers and we include this view in the new discussion of the paper. We expect that this will be further resolved by increasing the numbers of complete single neuron morphologies as well as orthogonal studies, such as the spatial transcriptomic MERFISH studies.

- Single neuron projection studies have shown that neighboring cortical PN neurons may belong to different subtype classes and that the organization of their relative distribution positions may be shown to be statistically distinct without necessarily being separated by distinct boundaries. (Winnubst et al 2019). Such a representation of the ET subtypes seems more accurate than attempting to define sublayers.

Response: We agree with the Reviewer that neighboring cortical PN neurons may belong to different subtype classes. Yet, it appears that two major subclasses of ET neurons, medulla-projecting versus thalamic projecting, are in relatively distinguished sublayers, as shown in this study and previous report (Economo, et al., 2018). This result is also consistent with MERFISH data (bioRxiv 2020.10.19.343129; bioRxiv 2020.06.04.105700), in which 5 different clusters of L5 ET neurons were recognized. While cells belonging to the L5_ET_1-4 clusters co-occupy the upper L5, L5_ET_5 cells are distinctly located in lower L5. Hence, we would like to suggest that more single cell morphology, MERFISH, BARseq and other types will be needed to further define the cell types that are in fact represented in distinct layers and those that are more broadly dispersed across layers.

- Another question is whether or not the distribution of different ET subtypes (for example) in different sublayers are consistent in different cortical areas. This is a general question regarding the distribution of the many different ET subtypes in different areas. For example, it appears that the distribution of medulla and thalamic ET PNs in the UL of MOp is different than that described for ALM (Economo et al. 2018).

Response: Based on our carefully comparison of the distributions of two subtypes of ET projection neurons (medulla projecting versus thalamus projecting) in the MOp-ul (as shown in this study) and compared that to ALM (Economo et al., 2018), we believe that they show a similar laminar preference in both cortical areas: medulla-projecting neurons are in deep L5b; whereas thalamus-projecting neurons are in superficial L5b.

- In the Figure 3 caption it is stated that “all corticothalamic neurons (CT) are distributed in layer 6a...”. Why are layer 5 ET neurons that project to the thalamus not considered “cortico-thalamic”.

Response: The definition of a “CT projection neuron” typically denotes that it resides in L6 and exclusively projects to the thalamus. As shown in single neuron morphology reconstruction data, some of the CT neurons generate short collateral projections to other cortical areas, but rarely to the brainstem. As shown in Figure 7b, one subgroup of C2.1 (2.1.2, Fig. 7b) also have short collateral projections in the zona incerta and MRN. These collateral axons are presumably a direct extension of cortico-thalamic projections observed in our anterograde tract tracing experiments. In contrast, L5

thalamic-projection ET neurons are defined based on their ultimate projection targets in the brainstem, but with collateral projections to the thalamus. These neurons usually display more complex axonal trajectory patterns (see Figure 7b).

- For corticostriatal neurons it appears that for this study only those that project contralaterally are considered. Data from single cell axon tracings have shown that individual corticostriatal neurons have very different patterns and distributions of projections such that there are some with very sparse ipsilateral projections with robust contralateral (Winnubst et al., 2019), such that there may be some corticostriatal neurons missed in the data used.

Response: In Figure 7e-f, we showed IT neurons with the ipsilateral and bi-lateral projections to the striatum (CP). We also identified some neurons with weak ipsilateral striatal projection but robust contralateral projections from our newly reconstructed data set. Our retrograde data demonstrating the laminar distribution of CP-projecting cells is analyzed for contralateral CP injections to help distinguish IT cells from ET cells, which have an exclusively ipsilateral CP projection. Incidentally, retrograde patterns in ipsi MOp are the same as contra MOp, suggesting IT->CP cells distribute equally throughout L5a/L5b, regardless of contra or ipsi CP injection, and they therefore overlap ET->ipsi-CP cells, which distribute only in L5b of ipsi MOp (based on Cre-dependent anterograde tracing of L5b neurons).

In our BARseq data, we were able to sample projections to both the ipsilateral and contralateral CP in IT neurons because IT and ET neurons could be distinguished by differences in other collateral projections. BARseq data showed that IT neurons projecting to either the ipsilateral or contralateral CP were largely in L5a/L5b. IT neurons projecting to only the ipsilateral CP were more restricted to L5a whereas contralateral CP-projecting IT neurons spanned a slightly larger range of laminar positions. Therefore, although there are ipsilateral CP-projecting IT neurons that could not be detected by injections of retrograde tracers into the contralateral CP, most of these neurons reside in the same laminar positions as contralateral CP-projecting IT neurons.

- There is a problem with the “spatial distribution of 40 genes selected from the Allen Brain Gene expression database(Extended data Fig. 11.)”. Many of those presented are not gene expression data but expression of Cre from the GENSAT Bac-Cre project (Gerfen and Heintz, 2013 and should be cited as the Cre lines in that paper are used multiple times in this study). First, the lines used are inaccurately identified only by the gene name. For the BAC-Cre lines it is essential to list the actual line, as in Sepw-NP39, Tlx_PL56, Sim1_KJ18, Chrna2_OE25, PlxnD1_OG1, Rbp4_KL100, and Ntsr1_GN220. In all of these cases there are multiple lines for each gene and often the expression patterns differ considerably. For example the Ntsr1-GN220 line expresses in layer 6 of the neocortex, while Ntsr1_GN209 does not express in neocortex at all, but does express in piriform cortex. While the referenced gene may be expressed in the particular PN subtype labeled by Cre expression, it is not necessarily the case.

Response: The Reviewer is correct that the distribution data in the Suppl Fig 11 included both native gene expression data and expression pattern of Cre lines using different gene promoters and among those the expression patterns of the transgenic Cre (BAC) lines are specific for some BAC lines and not see with other lines using the same BAC construct. While such transgenic lines, when properly validated, can still serve as useful tools for layer-specific labeling, we agree that the distinction between native gene and Cre transgenic *in situs* in the Suppl Fig 11 can be made better and we've updated this in the revised manuscript. We have also added Extended Table 2 to list all Cre lines including those originating from the Gensat project.

The data using BARseq and tracing of axonal projections of individual PN cortical neurons provide unprecedented details of the diversity of PN cortical subtypes in terms of the patterns of the multiple targets of individual neurons. The BARseq data identify the major classes of PN cortical neuron subtypes, IT, L5ET and CT subtypes and further subdivide these into at least 18 subgroups on their

projection targets. Single neuron axonal projection data also revealed distinct subtypes. Several comments:

- The BARseq data show laminar distributions that correlate generally with that revealed with retrograde tracing data. However, it is not clear whether the patterns of distribution of subtypes of ET PN neurons in sublaminae of layer 5 reported with retrograde tracers are also observed in the BARseq data. The data depicted in Figure 6 may demonstrate that but it is difficult to determine. It does appear from the BARseq data that for L5 ET neurons there are not distinct boundaries between sublaminae.

Response: We have included a laminar distribution plot of individual projections from the BARseq dataset to allow a better comparison of the laminar distribution of individual projections in the BARseq data (ED Fig. 21a). This plot showed that indeed the laminar distribution of the BARseq dataset corresponds well to those obtained from retrograde tracers. For example, consistent with the differential laminar distribution of ET neurons projecting to different targets, BARseq also showed distinct sublayers in L5 enriched in projections to the medulla (MY-ipsi), Spinal cord (Sp), Pontine gray (PG), and superior colliculus (SC).

As a further validation of the retrograde tracing analysis using BARseq, we were able to define the same 11 sublayers in the BARseq dataset, and showed that many projections were differentially distributed across adjacent sublayers (Fig. 6b). We further showed that each sublayer was most highly enriched in a unique set of two projection subgroups defined by the overall projection pattern of each neuron (Fig. 6h). These analyses do not show that the sublayer boundaries are distinct (in this regard we agree with the reviewer), but further demonstrates that the 11 sublayers are differentially enriched in different types of projection neurons.

In addition, we now also include new analysis on the BARseq dataset that identified systematic differences in collateral projection patterns for IT neurons in superficial and deep layers: IT neurons in superficial layers have more dedicated projections, whereas those in the middle sublayers project more broadly. This difference in the higher-order structures of projections suggests that IT neurons in superficial and deep layers provide distinct network structures that may allow different modes of communication in the cortex, a hypothesis consistent with other studies (e.g. Whitesell et al., 2021). This hypothesis implies potential functional and evolutionary consequences that can be explored and tested in future studies.

- The 18 subgroups identified in the BARseq data are not easily discerned from the data depicted in Figure 6. What are the 18 subgroups? While certain subgroups are described in the text a table or some other way of presenting them would be useful to make sense of them.

Response: We thank the Reviewer for pointing out to us that the data presentation was not sufficiently clear. In the revised manuscript we now include the top projection patterns for each of the 18 subgroups in ED Fig. 21k.

- Some of the sublaminal descriptions are overstated, for example the distinction of ipsi versus contralateral projecting corticostriatal neurons are described as being in different sublayers, but the data in extended data 19, 1 shows that there is a general difference, there is a lot of intermingling. More convincing is the select distribution of ipsi only projecting IT neurons in L6, whereas both ipsi and contra projecting IT neurons are intermingled in L5.

Response: We have changed the text describing the ipsi- and contralateral striatal projections to reflect the intermingling nature of the distributions.

- The BARseq data is presented to display the laminar distribution, it would also be informative to display the different subgroupings in the horizontal dimension to see if there is any organization in this plane of neurons with different projection patterns.

Response: We have included the distribution of the 18 subgroups in the tangential plane. Although we see some patterns in individual brains, these patterns were not consistent across the two brains. We attributed these differences to labeling biases across the two animals (Extended Data Fig. 21n).

- As has been shown from the *Janelia MouseLight* data, single neuron axon tracing data reveal the complexity of the different PN subtypes based on the diversity of collaterals (particularly those to multiple areas). While the analysis shown as identified 25 subtypes, there are likely other subtypes to be identified as cell projection data increases. Also minor not sure why in figure 9b the different projection patterns are described as models, as the data have shown there are examples of each type.

Response: We agree with the Reviewer that additional single neuron morphology data will likely reveal additional PN subtypes. We have removed the word “models” in the description of the single neuron complexity.

This study uses multiple types of data sets to provide a comprehensive description of the neuron subtypes in the upper limb area of the MOp. The inclusion of data using classic tract tracing, Cre specific anterograde and trans-synaptic tracing, single cell transcriptomics and axonal projection labeling, all registered into a standard mouse reference framework affirms current concepts of the neuroanatomical organization of the cerebral cortex. While there are not any new organizing principles to emerge from the analysis, the level of detail and the approach used provides a blueprint for addressing fundamental questions. A few examples:

Response: We appreciate positive comments from the Reviewer. Nevertheless, this study did reveal some novel organizational principles of MOp-UL PN types. For example, for the first time we clearly distinguished three types of the CT class, the VLA/PO projecting neurons in L6a, VM-projecting neurons in the deep sublayer of L6a and L6b, and contralateral thalamic projecting neurons in L6b. For the L5 ET class, we confirmed that the vast majority of thalamus-projecting neurons terminate in the midbrain and/or pontine, thus, are composed of cortico-tectal (SC, APN), cortico-rubral (RN) and cortico-pontine projecting neurons. One surprising finding is that MRN-projecting neurons appears to be distinguished from other L5 ET types as shown in BARseq data (Figure 6k). Our single neuron morphology data revealed that, while all medulla-projecting neurons share many common targets in the midbrain, pontine, and medulla, one particular subgroup (Cluster 3.2 in Figure 7b) generates denser projections to the brain structures (SPV, PSV, PB and MDRN) directly involving the coordination of orofacial and forelimb movements (Moore et al., 2014).

The IT class displays the most heterogeneity. Based on integrative analysis of retrograde labeling, BARseq, and single neuron reconstructions, we first catalog all IT neurons into two subclasses, STR- (or cortico-cortical), and STR+ (or cortico-cortical/striatal) neurons, following the same logic of Shepherd (2013). These subdivisions have an important functional significance, since cortico-striatal projection is the initiate step of transferring cortical information to the entire cortico-basal ganglia-thalamic loop and downstream motor system (Shepherd, 2013; Hintiryan et al., 2016; Foster et al., 2020). STR- IT neurons are mostly located in L2/3, 4, 6b with preferable ipsilateral projections to the frontal and insular areas (cluster 11, 13 in Figure 7b). In contrast, STR+ (cortico-cortical/striatal) projection neurons are mostly distributed in L5a and L5b-s (as well as fewer in L6 and L2/3) (Figure 3a,b) and generate extremely diverse axonal trajectories, mostly symmetric, to many more targets in other cortical areas and striatum (shown in Figure6, 7g).

- This study is limited to the description of one subarea of the MOp, would applying this approach to all of the MOp provide a more definitive way of parcellating the MOp and would the same organizational details in the Upper limb area be the same in other areas.

Response: While outside the scope of the current study, we agree that similar in-depth analysis of other MOp functional areas is important in order to fully understand MOp organization in relation to function.

- How does the organization compare between different primary cortical areas and between primary and secondary areas. Specifically, while this approach allows for the clear delineation of the UL area of MOp and would presumably demonstrated similar clear delineation of other specific MOp areas, clear boundaries of subareas of secondary motor areas may or may not emerge. Similarly, what about areas more “association” type areas, such as prefrontal and cingulate cortices.

Determining what elements of the organization of UL MOp are common and which differ between cortical areas will enable developing concepts of how information is organized and processed within cortical circuits to effect behavior. This study provides a roadmap and sets the standard for pursuing such questions.

Response: We agree that these are important questions and the current study can be considered as a good starting point to systematically catalog projection cell types across the entire cortex using integrative anatomical approaches. We can use the same technologies (multi-fluorescent tract tracing, BARseq, single neuron morphology, etc.) to define the borders of the other functional domains (i.e., mouth, lower limb, or trunk) of the MOp and MOs (secondary motor areas), as well as those higher order association areas (such as prefrontal and cingulate cortices). Many of those laminar specific genes display their expressions across the entire neocortex (i.e., *Cux2* in L2/3, *Etv1* in L5). Therefore, the same cre-lines can be also applied to characterize cell type-specific projections in other cortical areas.

A suggestion is that the title of the paper include the Brain Initiative Cell Census Network, as this is more about the DataSets and analysis than about the Primary Motor Cortex (also it isn't really about the Primary Motor Cortex but the Upper Limb area).

One additional note is that hopefully the Brain Initiative support of the Cell Census Network intended that data sets produced would be made available to researchers. A major benefit of the Allen Institute is that they not only provide comprehensive data sets but also provide instructions for how to access that data. Providing similar access to all of the data in this study would be beneficial.

Response: All anatomical data generated by BICCN are indeed publicly available through the BICCN data portal at Brain Image Library (BIL) and the Pittsburg Supercomputing Center (PSC).

Papers that should be cited:

Economo et al. (2018) Distinct descending motor cortex pathways and their roles in movement
Nature. 563:79-84.

Gerfen CR, Paletzki R, Heintz N (2013) GENSAT BAC Cre-recombinase driver lines to study the functional organization of cerebral cortical and basal ganglia circuits. *Neuron* 80:1368-1383.

Kita, T., & Kita, H. (2012). The subthalamic nucleus is one of multiple innervation sites for long-range corticofugal axons: A single-axon tracing study in the rat. *The Journal of Neuroscience*, 32, 5990_5999.

Response: We thank the Reviewer for noting these omissions. Each of the publications they mention is now cited appropriately in our revised manuscript.

Reviewer Reports on the First Revision:

Referee #2 (Remarks to the Author):

Review of Revision of: Cellular Anatomy of the Mouse Primary Motor Cortex

The extensive revisions made to the original submission of the Cellular Anatomy of the Mouse Primary Motor Cortex for the most part adequately address my concerns. In some cases there was additional clarification of descriptions of the findings and addition of references to prior work. In other cases the authors have justified how they have presented their findings and conclusions. While this reviewer (and presumably others) may not agree with all of their opinions, they have articulated their opinions clearly enough that will add to the academic discourse of the organization of the cortex.

An example is their “demarcation of 11 cortical sublayers”, which by being put into the Abstract becomes one of the major conclusions of the study. As another reviewer points out the 6 layer cortical model is used as a “straw man” as it has been known and described often that the 6 layer model is inadequate. This study suggests that the demarcation of 11 sublayers refines the classic six layer cortical organization. A problem with this construct is that defining “layers” implies boundaries and it is clear from many prior studies and data in this study that there are not “sharp borders” between the identified sublayers. An alternative description of the organization is that this and many other prior studies

- have identified many distinct subtypes of cortical projection neurons based on their projection patterns
- the spatial distribution of major classes of these subtypes conform generally with the classic 6 layer cortex (layer 2/3: intracortical projections, layer 5: subcortical projections and layer 6: thalamic projections)
- Subtypes of these major classes display spatial distribution patterns within identifiable zones of the 6 major layers (zones having more indistinct boundaries)

The problem with defining “sublayers” is that it perpetuates, doesn’t refine, the problem with the concept of the cortex being organized in layers with distinct boundaries. The “intermingling” of some newly identified cortical projection subtypes that are classified as being in different of the newly defined 11 sublayers may actually be due to there being a functional organization that does not conform to the 11 sublayer concept. There are also certainly cortical projection subtypes that have not been identified in this study. Another issue is that these 11 sublayers are identified in primary motor cortex, there are likely other projection subtypes in other cortical areas. These are the types of arguments that should be instructive in discussing the present study in terms of the organizing principles of the cortex. The data presented in this study, being very detailed, provides a basis for such discussion.

Specific comment:

In the Methods Section for the Animal Subjects used for Data Projection of the Allen Institute: Vrial Tracer Experiments. (line 1630) The sentence (line1642-1643): “Transgene expression patterns in many Cre driver lines used in this study were previously described 18.19” Should also include reference 33 (Gerfen et al Neuron 2013) as this paper originally characterized the projection patterns of 5 of the 9 lines used by the Allen Institute (Sepw1_NP39, Tlx3_PL56, Rbp4_KL100, Sim1_KJ18, Ntsr1_GN220). Also, these lines are used in multiple experiments and figures in the paper and identifying them by their gene alone is inadequate as there are multiple GENSAT BAC-Cre lines for a given gene, often with very different patterns of expression. The proper identification of these lines is provided in Supplemental Table 2 (though the reference listed (Gong et al, 2007) does not include these lines as they are included in the Gerfen et al. 2013 paper)

Author Rebuttals to First Revision:

Referee #2:

Remarks to the Author:

Review of Revision of: Cellular Anatomy of the Mouse Primary Motor Cortex

The extensive revisions made to the original submission of the Cellular Anatomy of the Mouse Primary Motor Cortex for the most part adequately address my concerns. In some cases there was additional clarification of descriptions of the findings and addition of references to prior work. In other cases the authors have justified how they have presented their findings and conclusions. While this reviewer (and presumably others) may not agree with all of their opinions, they have articulated their opinions clearly enough that will add to

the academic discourse of the organization of the cortex.

An example is their “demarcation of 11 cortical sublayers”, which by being put into the Abstract becomes one of the major conclusions of the study. As another reviewer points out the 6 layer cortical model is used as a “straw man” as it has been known and described often that the 6 layer model is inadequate. This study suggests that the demarcation of 11 sublayers refines the classic six layer cortical organization. A problem with this construct is that defining “layers” implies boundaries and it is clear from many prior studies and data in this study that there are not “sharp borders” between the identified sublayers. An alternative description of the organization is that this and many other prior studies

- have identified many distinct subtypes of cortical projection neurons based on their projection patterns
- the spatial distribution of major classes of these subtypes conform generally with the classic 6 layer cortex (layer 2/3: intracortical projections, layer 5: subcortical projections and layer 6: thalamic projections)
- Subtypes of these major classes display spatial distribution patterns within identifiable zones of the 6 major layers (zones having more indistinct boundaries)

The problem with defining “sublayers” is that it perpetuates, doesn’t refine, the problem with the concept of the cortex being organized in layers with distinct boundaries. The “intermingling“ of some newly identified cortical projection subtypes that are classified as being in different of the newly defined 11 sublayers may actually be due to there being a functional organization that does not conform to the 11 sublayer concept. There are also certainly cortical projection subtypes that have not been identified in this study. Another issue is that these 11 sublayers are identified in primary motor cortex, there are likely other projection subtypes in other cortical areas. These are the types of arguments that should be instructive in discussing the present study in terms of the organizing principles of the cortex. The data presented in this study, being very detailed, provides a basis for such discussion.

Response: We thank Reviewer's thoughtful comments and suggestions. As suggested, we made the following statement:

(page 8, line 3) ... These experiments revealed a refined laminar organization, suggesting 26 PN subtypes (**Fig. 1f**) spanning 11 newly delineated layers and sublayers (1, 2, 3, 4, 5a, 5b-superficial, 5b-middle, 5b-deep, 6a-superficial, 6a-deep, and 6b) (**Fig. 1f**). Some types of neurons display preferential sublaminar patterns, but other types occur in a smoother gradient across sublayers.

Specific comment:

In the Methods Section for the Animal Subjects used for Data Projection of the Allen Institute: Vrial Tracer Experiments. (line 1630) The sentence (line1642-1643): “Transgene expression patterns in many Cre driver lines used in this study were previously described 18.19” Should also include reference 33 (Gerfen et al Neuron 2013) as this paper originally

characterized the projection patterns of 5 of the 9 lines used by the Allen Institute (Sepw1_NP39, Tlx3_PL56, Rbp4_KL100, Sim1_KJ18, Ntsr1_GN220). Also, these lines are used in multiple experiments and figures in the paper and identifying them by their gene alone is inadequate as there are multiple GENSAT BAC-Cre lines for a given gene, often with very different patterns of expression. The proper identification of these lines is provided in Supplemental Table 2 (though the reference listed (Gong et al, 2007) does not include these lines as they are included in the Gerfen et al. 2013 paper)

Response: We cited this reference (Gerfen et al., Neuron 2013) in text and also in Extended Table 2.